# Channelrhodopsin variants for high-rate optogenetic neurostimulation at low light intensities

Lennart Roos [1,2,3,4,14], Aida Garrido-Charles [1,2,5,6,14], Niels Albrecht [1,2,3,15,15], Anna Vavakou[1,2,3,15,15], Alexey Alekseev[1,2,3,5], Martina Bleyer [7], Anupriya Thirumalai [1,2,3,8], Artur Mittring [1,2,8], Theocharis Alvanos [1,2,9,10], Antoine T Huet [1,2,8], Ernst Bamberg[11], Kathrin Kusch [1,3,8,12,13], Bettina J Wolf [1,2,3], Tobias Moser [1,2,3,9,10,13]✉ & Thomas Mager [1,2,3,5,13]✉

## Abstract

**Optogenetics allows versatile control of excitable cell networks, which advances basic science research and drives the development of future medical applications. Fast-closing channelrhodopsins (ChRs) are required for high temporal fidelity of neurostimulation, but their short channel open times require sufficient plasma membrane expression and high light intensity, challenging clinical translation. Here, we addressed the need of high-rate neurostimulation by engineering optimized blue-light-sensitive ChR variants. In particular, we report on the ChR2 variant f-ChR2 TC enabling high frequency stimulation at low light requirements, due to its good plasma membrane targeted expression and balanced closing kinetics. Upon Adeno-associated virus (AAV) mediated f-ChR2 TC expression in spiral ganglion neurons of the inner ear in mice, f-ChR2 TC accordingly enabled optogenetic stimulation of the auditory nerve with sizeable responses beyond 300 Hz and low pulse energy thresholds. Translating the approach to the larger cochlea of gerbils, we tested the utility of f-ChR2 TC for evaluating multichannel optical cochlear implants with blue light emitting diodes and found light-efficient stimulation of the auditory pathway by single LEDs at rates ≥100 Hz.**

**Keywords** Cochlear Implant; Deafness; Gene Therapy; Optogenetics; Synapse
**Subject Category** Neuroscience

## Introduction

With a global prevalence of 5.6% across all age groups, disabling hearing loss (HL) is the most common human sensory deficit (WHO 2021). HL is mainly caused by dysfunction of the cochlea (Moser et al, 2013; Eggermont 2017). Hearing aids and cochlear implants (CIs) are the key means of rehabilitation, but novel treatment options such as gene and optogenetic therapy are emerging (Dieter et al, 2020a; Kleinlogel et al, 2020; Wolf et al, 2022; Landegger et al, 2024; Moser et al, 2024). CIs directly stimulate the spiral ganglion neurons (SGNs), bypassing the defective sensory organ (Hochmair and Hochmair-Desoyer 1981; Wilson 2015; Lenarz 2017; Zeng 2017). CIs partially restore hearing and provide open speech perception for patients with severe to profound HL. However, CI users face limitations in daily life, such as unnatural auditory perception and difficulties in speech comprehension in noisy environments (Hunniford et al, 2023). This is attributed to the widespread of electric current from each electrode contact of the implanted CI in the saline-filled and tonotopically organized cochlea. Thus, large populations of SGNs are activated and transfer of frequency and intensity information is thereby limited (Shannon 1983; Kral et al, 1998; Miller et al, 2006). As light can be confined in space better than electrical current, stimulation by optical CIs (oCI) promises improved frequency coding and, consequently, hearing restoration for CI users (Izzo et al, 2006; Hernandez et al, 2014). Optogenetic stimulation of SGNs utilizes channelrhodopsins (ChRs), light-gated ion channels (Hernandez et al, 2014; Mager et al, 2018; Wrobel et al, 2018). Indeed, optogenetic SGN stimulation showed improved spectral selectivity compared to eCI stimulation (Dieter et al, 2019; Dieter et al, 2020b; Keppeler et al, 2020).

One concern regarding future clinical translation of optogenetic hearing restoration was that improved sound frequency coding might be traded for poorer temporal coding, given that ChRs

[1]Institute for Auditory Neuroscience and InnerEarLab, University Medical Center Göttingen, 37075 Göttingen, Germany. [2]Cluster of Excellence "Multiscale Bioimaging: from Molecular Machines to Networks of Excitable Cells" (MBExC), University of Göttingen, 37073 Göttingen, Germany. [3]Else Kröner Fresenius Center for Optogenetic Therapies, University Medical Center Göttingen, 37075 Göttingen, Germany. [4]Department of Otolaryngology, University Medical Center Göttingen, 37075 Göttingen, Germany. [5]Advanced Optogenes Group, Institute for Auditory Neuroscience, University Medical Center Göttingen, 37075 Göttingen, Germany. [6]Institute for Cardiovascular Physiology, University Medical Center Göttingen, Göttingen, Germany. [7]Laboratory Animal Science Unit, Pathology, German Primate Center, Leibniz Institute for Primate Research, 37077 Goettingen, Germany. [8]Auditory Circuit lab, Institute for Auditory Neuroscience and InnerEarLab, University Medical Center Göttingen, 37075 Göttingen, Germany. [9]Collaborative Research Center SFB 1286 "Quantitative Synaptology", University of Göttingen, 37073 Göttingen, Germany. [10]Auditory Neuroscience and Synaptic Nanophysiology Group, Max-Planck-Institute for Multidisciplinary Sciences, 37075 Göttingen, Germany. [11]Max Plank Institute for Biophysics, Frankfurt am Main, Germany. [12]Functional Auditory Genomics Group, Auditory Neuroscience and Optogenetics laboratory, German Primate Center, 37077 Göttingen, Germany. [13]Collaborative Research Center SFB 1690 "Disease mechanisms and functional restoration of sensory and motor systems", University of Göttingen, 37073 Göttingen, Germany. [14]These authors contributed as first authors: Lennart Roos, Aida Garrido-Charles. [15]These authors contributed as second authors: Niels Albrecht, Anna Vavakou. ✉E-mail: tobias.moser@mpinat.mpg.de; thomas.mager@med.uni-goettingen.de

typically close within ms after light off. Indeed, ChR closing kinetics is a crucial factor for the temporal fidelity of neuronal photostimulation (Gunaydin et al, 2010; Klapoetke et al, 2014; Keppeler et al, 2018; Mager et al, 2018). Physiological sound encoding employs SGN firing of up to a few hundred spikes per second with high temporal fidelity. Current sound coding strategies of eCIs employ high stimulation rates (~1 kHz) to achieve more stochastic firing due to partial SGN refractoriness. The firing rates achieved with strong electrical stimulation at these rates approach ~250 spikes per second (Miller et al, 2008), similar to maximal steady state SGN firing rates during acoustic stimulation (Kiang et al, 1965; Rose et al, 1967; Sachs and Abbas 1974). Yet, speech intelligibility has also been achieved at much lower stimulation rates (Friesen et al, 2005; Shannon et al, 2011).

Importantly, optogenetic stimulation mediated by ChRs with fast channel closing kinetics has enabled near-physiological SGN firing rates (100-300 spikes per second, Keppeler et al, 2018; Mager et al, 2018; Bali et al, 2021). These studies utilized the blue-light-activated, fastest native ChR, Chronos (Klapoetke et al, 2014), that was optimized for membrane targeting (Chronos ES/TS, Keppeler et al, 2018) and the red-light-activated fast and very fast Chrimson variants f- and vf-Chrimson (Mager et al, 2018; Bali et al, 2021). Mutating helix F, the closing kinetics of Chrimson (~24.6 ms, Klapoetke et al, 2014; Mager et al, 2018), were accelerated by a factor of ~4 in f-Chrimson. Helix F moves during open to closed transition (Sattig et al, 2013; Müller et al, 2015) and accelerated transition likely results from helix F movement-controlled protonation reactions governing the cycle time of ChRs (Mager et al, 2018).

However, charge transfer by ChRs and in consequence the efficiency of neuronal photostimulation is limited by the channel open time (Klapper et al, 2016). Thus, the light intensity needed to optogenetically evoke action potentials is higher for fast-closing ChRs than for ChRs with longer open times (Berndt et al, 2011; Mager et al, 2018). The limited charge transfer during short channel open times can partially be compensated by a stronger plasma membrane targeted expression (Keppeler et al, 2018; Bali et al, 2021). Here, in an effort to further improve the temporal fidelity for high-frequency optogenetic SGN stimulation, we pursued two strategies: (i) acceleration of ChR closing kinetics beyond that of Chronos and (ii) generation of ChR variants, which balance fast closing kinetics and high neural light sensitivity. Toward the first strategy, we generated Chronos F236Y (fast Chronos: f-Chronos), which, to our knowledge, is the fastest closing ChR to date. In the second approach, we engineered two ChR variants with balanced kinetics, namely the variant Chronos L149C (Chronos LC), which also shows minimal photocurrent desensitization, and the plasma membrane targeting optimized variant ChR2 F219Y/T159C (f-ChR2 TC). The comparative assessment in the rodent auditory pathway showed that, of the novel ChR variants, f-ChR2 TC is most suitable for blue light high-rate neurostimulation with low light requirements. We show that f-ChR2 TC enables efficient stimulation of the auditory pathway by LED-based oCIs in Mongolian gerbils.

# Results

## Generating blue light-activated ChR variants with optimized properties

In order to further accelerate the channel closing kinetics of Chronos (Klapoetke et al, 2014), the fastest naturally occurring

ChR to our knowledge (Keppeler et al, 2018), we generated Chronos F236Y. This mutation targets the helix F position homologous to the F219Y mutation in ChR2, a spot known to accelerate channel closing in green algal ChRs (Mager et al, 2018). Indeed, as shown by whole-cell patch clamp experiments in transiently transfected NG cells, the ultrafast ChR Chronos F236Y (f-Chronos, $\tau_{off} = 1.7 \pm 0.1$ ms, $n = 4$ at room temperature (RT), $\tau_{off} = 0.8 \pm 0.1$ ms, $n = 4$ at 33 °C) was faster than Chronos ($\tau_{off} = 3.1 \pm 0.5$ ms, $n = 6$ at RT ($p = 0.0009$, unpaired $t$-test with Welch's correction), $\tau_{off} = 1.9 \pm 0.5$ ms, $n = 6$ at 33 °C ($p = 0.021$, unpaired $t$-test with Welch's correction); Fig. 1A,B; Tables EV1 and 2). Moreover, the channel closing kinetics of f-Chronos lacked the pronounced voltage-dependence observed in Chronos (Fig. 1C), which indicates that, in contrast to Chronos, the closing of f-Chronos is rate limited by an electroneutral step. The finding that the channel closing kinetics of f-Chronos becomes voltage-dependent when lowering the extracellular pH, suggests that the electrogenic step can be assigned to a protonation reaction governing open-to-closed transition and that the electroneutral step may represent a conformational change during open-to-closed transition. This finding supports the hypothesis that the helix F mutations accelerate helix movement that controls protonation reactions governing the cycle time of ChRs (Mager et al, 2018). In line with the ultrafast channel closing kinetics of f-Chronos, the quantification of photocurrent fluctuations in response to repetitive, short light pulses (pulse length: 1 ms at ~40 mW/mm²) at different frequencies (Fig. EV1) indicated comparatively high fluctuation amplitudes in the 125 to 500 Hz range (Fig. EV1C), which could benefit SGN stimulation at high rates. At high rates of stimulation, the extent of desensitization upon pulsed illumination ($I_{stat}/I_{peak} = 0.19 \pm 0.02$; $n = 3$) approached desensitization found at continued illumination that was slightly more pronounced than in Chronos ($I_{stat}/I_{peak} = 0.19 \pm 0.04$; $n = 14$ for f-Chronos compared to $I_{stat}/I_{peak} = 0.35 \pm 0.09$; $n = 10$ for Chronos; $p = 0.002$ by unpaired $t$-test with Welch's correction; Fig. 1E and Table EV2). Furthermore, we determined a significantly lower stationary photocurrent density for f-Chronos ($4.66 \pm 2.98$ pA/pF, $n = 13$) than for Chronos ($24.01 \pm 7.46$ pA/pF, $n = 14$; $p < 0.0001$, Mann–Whitney test; Fig. 1F; Table EV1).

Next, we turned to generating ChR variants with intermediate closing kinetics. ChR2 L132C (CatCh) enables neurostimulation at low light levels owing to larger photocurrents, reduced desensitization, and slowed channel closing (Kleinlogel et al, 2011). The introduction of the homologous mutation (Chronos L149C: Chronos LC) in Chronos and f-Chronos led to a slowing of the channel closing kinetics (Chronos LC: $\tau_{off} = 8.2 \pm 1.7$ ms, $n = 11$ vs. Chronos: $\tau_{off} = 3.7 \pm 0.7$ ms, $n = 10$, $p < 0.0001$ by unpaired $t$-test with Welch's correction; f-Chronos LC: $\tau_{off} = 3.6 \pm 0.9$ ms, $n = 8$ vs. f-Chronos: $\tau_{off} = 1.6 \pm 0.4$ ms, $n = 6$, $p = 0.0007$ by Mann–Whitney test) and to a reduction of photocurrent desensitization ($I_{stat}/I_{peak}$ for Chronos: $0.35 \pm 0.09$, $n = 10$ vs. Chronos LC: $0.62 \pm 0.08$, $n = 6$, $p < 0.0001$ by unpaired $t$-test; f-Chronos: $0.19 \pm 0.05$, $n = 14$ vs. f-Chronos LC: $0.29 \pm 0.03$, $n = 3$, $p = 0.0021$ by unpaired $t$-test; TableEV2).

ChR2 T159C (ChR2 TC) exhibits a twofold larger stationary photocurrent than wild-type ChR2 (Berndt et al, 2011). We combined the T159C mutation with two mutations known to accelerate channel closing, E123T (Gunaydin et al, 2010) and F219Y (Mager et al, 2018). The channel closing kinetics of ChR2

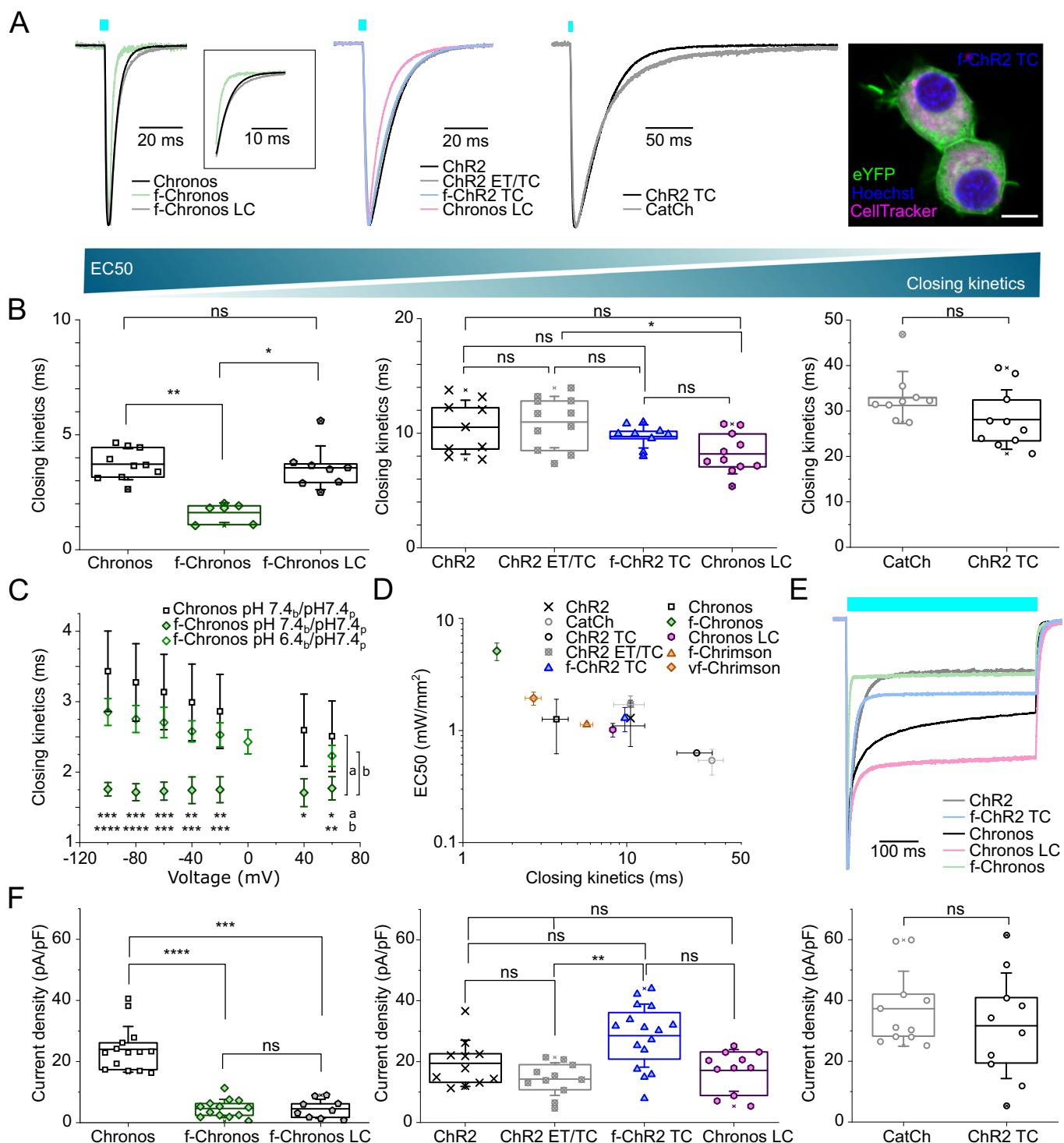

E123T/T159C (ChR2 ET/TC: $\tau_{off}$ = 11.0 ± 2.2, $n$ = 11; $p$ < 0.0001 by unpaired $t$-test with Welch's correction) and ChR2 T159C/F219Y (f-ChR2 TC: $\tau_{off}$ = 9.7 ± 1.0 ms, $n$ = 9; $p$ < 0.0001 by unpaired $t$-test with Welch's correction) were indeed faster than the channel closing kinetics of ChR2 TC ($\tau_{off}$ = 28.2 ± 6.5 ms, $n$ = 10). While the photocurrent density of f-ChR2 TC remained high (28.42 ± 10.36 pA/pF, $n$ = 16, $p$ = 0.6352 (ns) by unpaired $t$-test with Welch's

correction), the photocurrent density of ChR2 ET/TC (14.15 ± 5.37 pA/pF, $n$ = 11, $p$ = 0.013 by unpaired $t$-test with Welch's correction) was considerably smaller than the photocurrent density of ChR2 TC (31.39 ± 17.53 pA/pF, $n$ = 10; Table EV1). The C-terminal fusion of plasma membrane targeting sequences from the inward rectifying potassium channel Kir2.1 (TS and ES) had no considerable effect on the photocurrent density of f-ChR2 TC (Gradinaru

**Figure 1. Biophysical characterization of photocurrents elicited by blue light-activated ChRs.**

(A) Normalized photocurrents of ChR variants in response to a blue light pulse ($\lambda = 473$ nm, 3 ms, ~30 mW/mm$^2$) at a membrane potential of $-60$ mV. The whole-cell patch-clamp measurements were performed in NG cells heterologously expressing ChRs, 2 to 3 days after transient transfection. Representative confocal image of transfected NG cells expressing f-ChR2 TC-EYFP. EYFP fluorescence shown in green, nucleus stain (Hoechst, 1:2000) shown in blue, cytosol stain (CellTrackerTM, Invitrogen) shown in magenta. Scale bar 10 μm. The same image is shown in Appendix Fig. S1D. (B) Comparison of closing kinetics of natural and engineered blue-light activated ChRs. $\tau_{off}$ values were determined by monoexponential fits to photocurrents evoked by a 3 ms light pulse ($\lambda = 473$ nm, ~30 mW/mm$^2$), except for f-Chronos ($\lambda = 500$ nm, 7 ns, $10^{20}$ photons/m$^2$). Multiple group comparison obtained by Kruskal–Wallis test and post hoc Dunn's test: $p > 0.9999$; **$p = 0.0059$; *$p = 0.039$; $p = 0.1226$; *$p = 0.0188$; $p > 0.9999$; $p > 0.9999$; $p > 0.9999$; $p = 0.6584$; two-group comparison obtained by Mann–Whitney test: $p = 0.0947$ ($p > 0.05$ (ns)). Chronos: $n = 10$, f-Chronos: $n = 4$, f-Chronos LC: $n = 8$; ChR2: $n = 9$, ChR2 ET/TC: $n = 11$, f-ChR2 TC: $n = 9$, Chronos LC: $n = 11$; CatCh: $n = 9$, ChR2 TC: $n = 10$. Center lines represent mean values, minima and maxima are shown by crosses. Boxes show the 25$^{th}$ and 75$^{th}$ percentile and error bars depict SD. (C) Channel-closing kinetics at different membrane potentials were analyzed by ANOVA test with Bonferroni correction for Chronos vs f-Chronos (***$p = 0.0001$; ***$p = 0.0002$; ***$p = 0.0004$; **$p = 0.0012$; **$p = 0.0025$;*$p = 0.0116$ (unpaired $t$-test); *$p = 0.0251$); Chronos vs f-Chronos pH 6.4 ($p = 0.1421$; $p = 0.189$; $p = 0.324$; $p = 0.3859$; $p = 0.5984$; $p = 0.752$); and f-Chronos vs f-Chronos at pH 6.4 (**$p = 0.0079$; **$p = 0.0092$; *$p = 0.0127$; *$p = 0.0331$; *$p = 0.0421$; $p = 0.2888$) ($p > 0.05$ (ns)). Shown are the $\tau_{off}$ values at RT. Error bars depict SD. The $\tau_{off}$ values were determined by monoexponential fits to photocurrents, which were measured in response to blue light illumination ($\lambda = 500$ nm, 7 ns, $10^{20}$ photons/m$^2$). The whole-cell patch-clamp experiments were performed in NG cells, 2 to 3 days after transient transfection with Chronos (black squares, $n = 6$) and f-Chronos (green rhombus, $n = 4$) at the shown pH values ("b" stands for bath and "p" stands for pipette). (D) The half maximal activation value (EC$_{50}$) values obtained from stationary photocurrent amplitudes upon a 500-ms light pulse (473 nm) at a holding potential of $-60$ mV. ChR2: $n = 3$, CatCh: $n = 3$, ChR2 TC: $n = 1$, ChR2 ET/TC: $n = 3$, f-ChR2 TC: $n = 3$, Chronos: $n = 6$, f-Chronos: $n = 6$, Chronos LC: $n = 2$. Error bars depict SD. Data points for Chrimson variants replotted from (Mager et al, 2018; stimulated at 594 nm). (E) Normalized photocurrents of ChR variants in response to a blue light pulse ($\lambda = 473$ nm, 500 ms, ~30 mW/mm$^2$) at a membrane potential of $-60$ mV. (F) Comparison of stationary current densities of natural blue-light activated ChRs and gain of function mutants obtained by Kruskal–Wallis test and post hoc Dunn's test: ***$p = 1.47e-4$; ****$p = 3.43e-5$; $p > 0.9999$; $p > 0.9999$; $p > 0.9999$; $p = 0.1527$; $p > 0.9999$; **$p = 0.0012$; $p = 0.0629$; two-group comparison obtained by Mann–Whitney test: $p = 0.5116$ ($p > 0.05$ (ns)). Chronos: $n = 14$, f-Chronos: $n = 13$, f-Chronos LC: $n = 9$; ChR2: $n = 11$, ChR2 ET/TC: $n = 11$, f-ChR2 TC: $n = 16$, Chronos LC: $n = 11$; CatCh: $n = 11$, ChR2 TC: $n = 10$. Center lines represent mean values, minima and maxima are shown by crosses. Boxes show the 25$^{th}$ and 75$^{th}$ percentile and error bars depict the SD. Source data are available online for this figure.

et al, 2010; Keppeler et al, 2018, Appendix Fig. S1). In agreement with previous reports (Berndt et al, 2011; Mager et al, 2018), the faster ChR variants were less light sensitive (Fig. 1D). Moreover, analysis of photocurrent fluctuations in the high-frequency range (125 to 500 Hz) showed that the fluctuation amplitudes correlated with the speed of channel closing (Fig. EV1; Table EV3).

## Characterizing the utility of the new ChR variants in hippocampal neurons

Next, we characterized ultrafast f-Chronos, the gain-of-function mutants Chronos LC and f-ChR2 TC, as well as the previously described ChR2 mutant CatCh after adeno-associated viral vector (AAV2/9) mediated gene transfer in primary cultures of rat hippocampal neurons. The virus titer assuring robust transduction and proper neuronal expression of ChRs was found by monitoring the epifluorescence of the EYFP tag fused to the ChR constructs (Fig. 2A). The photocurrent densities and $\tau_{off}$ values determined in hippocampal neurons were similar to the values obtained in NG cells (Figs. 1 and 2B; Appendix Fig. S3; Appendix Table S1). Whereas Chronos LC, f-ChR2 TC and CatCh enabled reliable neuronal photostimulation by the short (1 ms) light pulses, spike probability for f-Chronos expressing neurons was low, with 20% of the neurons (3 out of 14) showing a spike probability higher than 80% (Fig. 2C). The low success rate in f-Chronos expressing neurons likely results from (i) suboptimal plasma membrane targeted expression, which is reflected in the comparatively low stationary photocurrent density values measured at saturating light intensities (Fig. 2D,E) and (ii) the limited charge transfer evoked by the short (1 ms) light pulses (Fig. EV1D) given the ultrafast channel closing kinetics (Fig. 1B). The investigation of high-rate neurostimulation is impeded by the limited and heterogeneous intrinsic maximal firing rate of rat hippocampal neurons which is typically 40 to 60 Hz (Gunaydin et al, 2010; Mager et al, 2018), well below that of fast spiking neurons such interneurons and SGNs (Mager

et al, 2018). We accordingly turned to the investigation of optogenetic SGN stimulation in the auditory pathway of mice.

## In vivo characterization in the auditory pathway

Next, we evaluated the utility of the newly engineered ChRs for high-rate neurostimulation in vivo in comparison to state-of-the art ChRs in mice. We chose the auditory pathway as a model system as it operates with high temporal fidelity and offers a promising avenue of clinical translation of optogenetic neurostimulation for improved hearing restoration by oCI.

First, we analyzed the new ChR variants by recording optically evoked auditory brainstem responses (oABR). Optogenetic modification of SGNs was achieved by postnatally injecting AAV2/9 carrying f-ChR2 TC, Chronos LC, and f-Chronos under the control of the human synapsin promoter into the scala tympani via the round window (at day $6 \pm 1$, Fig. 3A).

oABR measurements were performed at the age of $77 \pm 15$ days by surgically exposing the mouse cochlea, followed by the insertion of a laser-coupled optical fiber (200 μm diameter) via the round window (Fig. 3A). Blue light (473 nm) was then directly projected onto the modiolus harboring ChR-expressing SGNs. Upon optogenetic stimulation, we were able to record oABRs in all animals expressing f-ChR2 TC and Chronos LC (Fig. 3B,C). oABR typically displayed five waves, which likely reflected the synchronous activation of ChR-expressing SGNs (wave I) and signal propagation along the lower auditory pathway (subsequent waves; Figs. 3B, C and EV2). We did not detect oABRs with f-Chronos in $n = 7$ mice despite SGNs expressing f-Chronos (ABR-Recordings: Fig. 3D, immunohistochemistry: Fig. EV3).

We found that f-ChR2 TC outperformed Chronos LC in eliciting oABRs with lower threshold radiant flux and with larger oABR amplitudes (Figs. 3E–G and EV2C–F). Specifically, radiant energy thresholds for f-ChR2 TC were significantly lower (Fig. 3E: $1.44 \pm 0.798$ μJ, $n = 18$ vs. $5.27 \pm 2.24$ μJ, $n = 9$; $p = 0.0051$), Wave I

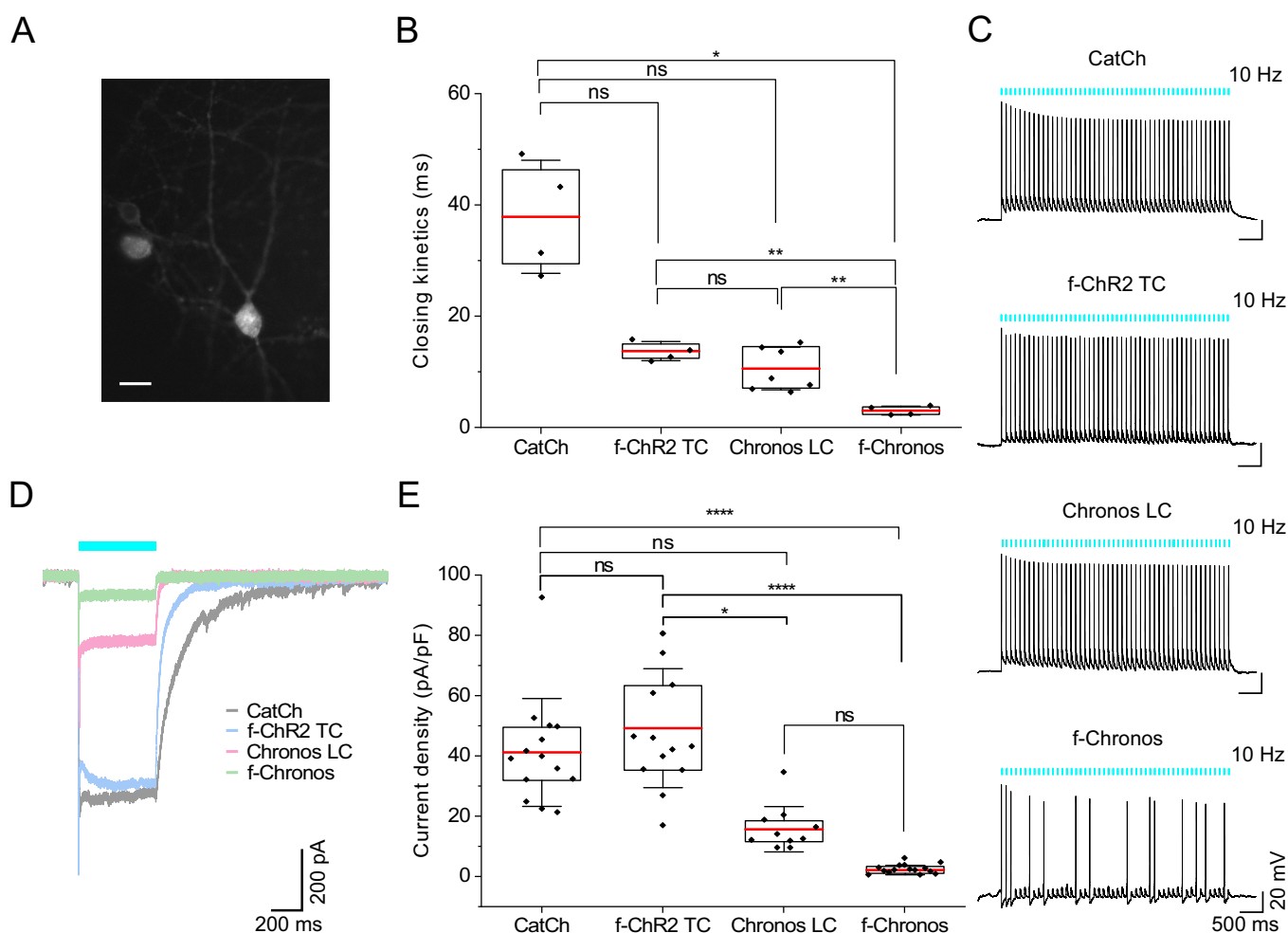

**Figure 2. Optogenetic stimulation of hippocampal neuron firings with optimized blue-light activated ChRs.**

(A) Representative EYFP fluorescence image of hippocampal neurons expressing f-Chronos-EYFP. Scale bar 20 μm. (B) Whole-cell patch clamp measurement in primary cultures of rat hippocampal neurons after transduction with AAV2/9 virus ($5 \times 10^9$ GC/mL) carrying CatCh, f-ChR2 TC, Chronos LC, or f-Chronos. Comparison of closing kinetics in hippocampal neurons of blue-light activated ChRs gain of function mutants obtained from monoexponential decay fits upon 1 ms light pulse (488 nm; 0.78 to 40 mW/mm²). Monoexponential decay fits of two f-Chronos neurons were obtained from a 9 ms light pulse, due to small photocurrents. Multiple group comparison obtained by Brown–Forsythe and Welch ANOVA test and post hoc Dunnett's T3 multiple comparisons test: *$p = 0.024$; $p = 0.052$; $p = 0.0672$; **$p = 0.0015$; $p = 0.03965$; **$p = 0.0079$ ($p > 0.05$ (ns)). CatCh: $n = 4$; f-ChR2 TC: $n = 4$; Chronos LC: $n = 7$; f-Chronos: $n = 4$. Center lines represent mean values, minima and maxima are shown by scattered data points. Boxes show the 25th and 75th percentile and error bars depict SD. (C) Exemplary current clamp traces of 1 ms light pulse stimulation at 10 Hz ($\lambda = 488$ nm, 0.78 to 17.18 mW/mm²; RT). (D) Exemplary photocurrent traces upon 500 ms light pulse stimulation ($\lambda = 488$ nm, 40 mW/mm²; RT). (E) Comparison of current densities in hippocampal neurons of blue-light activated ChRs gain of function mutants (500 ms; 488 nm; 20 to 40 mW/mm²). Multiple group comparison obtained by Kruskal–Wallis test and post hoc Dunn's test: ****$p = 2.29e{-}6$; $p = 0.0829$; $p > 0.9999$; ****$p = 4.29e{-}8$; *$p = 0.0117$; $p = 0.2013$ ($p > 0.05$ (ns)). CatCh: $n = 14$; f-ChR2 TC: $n = 14$; Chronos LC: $n = 10$; f-Chronos: $n = 15$. Center lines represent mean values, minima and maxima are shown by scattered data points. Boxes show the 25th and 75th percentile and error bars depict SD. Neurons included in the quantifications have an input resistance higher than 100 MΩ. Source data are available online for this figure.

(P1-N1) amplitudes were significantly higher (Fig. 3F: 14.95 ± 4.65 μV, $n = 18$ vs. 1.83 ± 0.911 μV, $n = 9$; $p = 0.0004$), and P1-N1 latencies were significantly shorter (Fig. 3G: 0.61 ± 0.09 ms, $n = 18$ vs. 0.82 ± 0.12 ms, $n = 9$; $p = 0.0076$) compared to Chronos LC. P1-N1 amplitudes of f-ChR2 TC-mediated oABRs grew substantially with radiant flux (Fig. 3H), promising a broad dynamic range of optogenetic sound encoding. We did not observe significant differences of blue-light activated f-ChR2 TC compared to red-light activated f-Chrimson, while f-ChR2 TC significantly outperformed Chronos for (I) threshold: $p = 0.0002$; (II) P1-N1 amplitude: $p = 0.001$, and (III) P1 latency: $p = 0.0002$ (unpaired Kruskal–Wallis test adjusted for multiplicity by Dunn's

correction; Fig. 3E–G, replotted from Keppeler et al, 2018 and Zerche et al, 2023).

To further investigate the temporal fidelity of the optogenetic SGN stimulation with our newly engineered blue-light sensitive ChRs, we recorded oABRs in response to blue light pulses at stimulation rates of 20 to 400 Hz for saturating radiant flux (~38 to 45.6 mW) for f-ChR2 TC (Fig. 3I,J, $n = 18$) and Chronos LC (Fig. EV2A, $n = 9$). As expected, oABR amplitudes declined with increasing stimulation rate, yet, oABRs were detectable at stimulation rates beyond 300 Hz. Again, f-ChR2 TC-mediated oABRs were compatible with those obtained with Chronos and

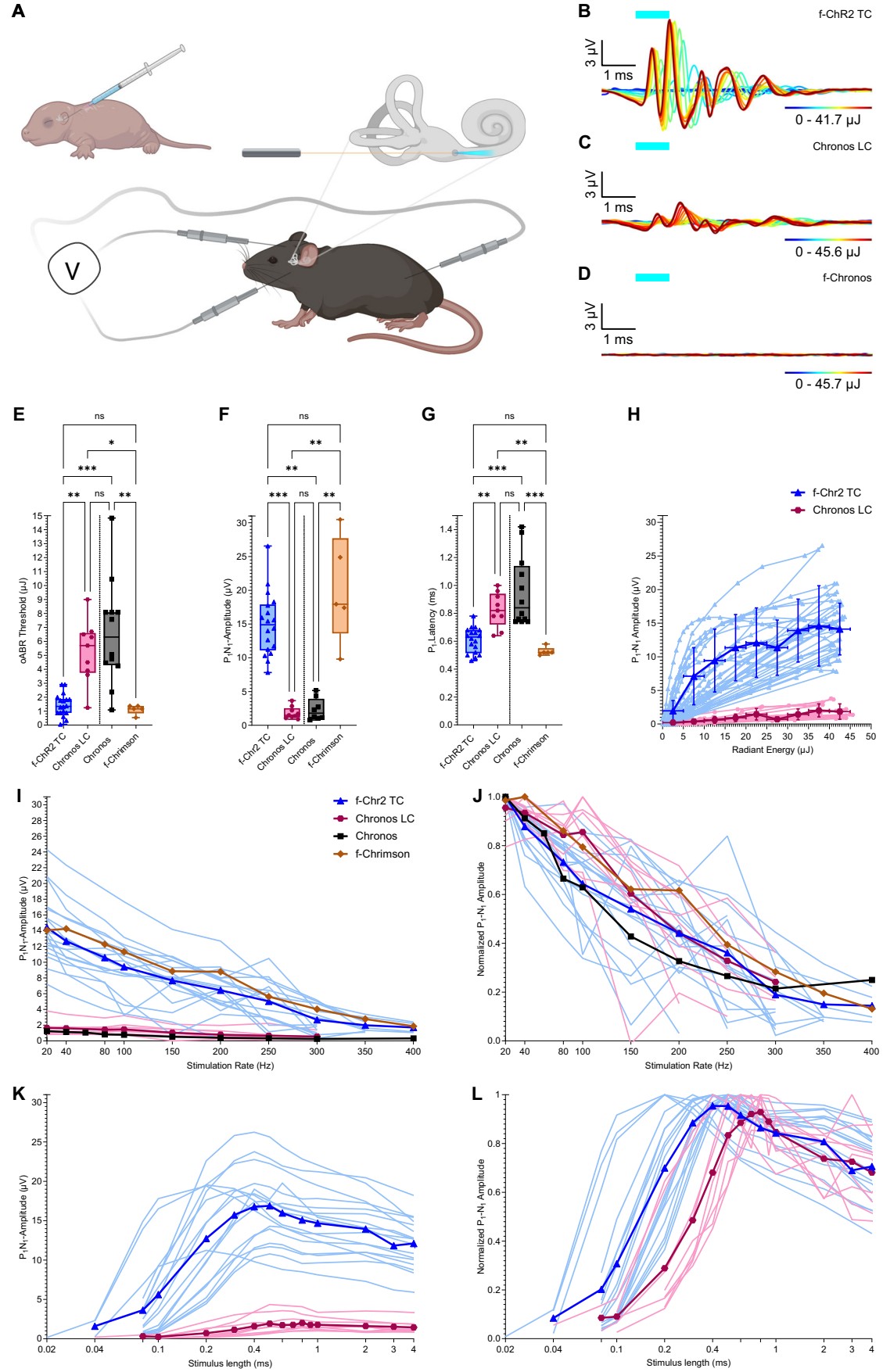

**Figure 3. Optogenetic stimulation of the mouse cochlea by f-ChR2 TC reveals low light requirements and good temporal fidelity.**

(A) Schematic representation of the in vivo method workflow with: (left) local administration of AAVs carrying the newly engineered blue-light sensitive ChR with optimized properties into the round window of the cochlea of neonatal mice. (right) insertion of a blue-light emitting laser-coupled fiber into the round window to probe for optical auditory brainstem recordings (oABR, bottom). (B–D) Exemplary oABRs driven with pulses of varying radiant flux (1 ms pulses at 10 Hz, color codes the radiant energy in μJ) of mice with SGN expressing f-ChR2 TC (ChR2(T159C/F219Y)-EYFP, B), Chronos LC (Chronos(L149C)-ES-EYFP-TS, C) and f-Chronos (Chronos(F236Y)-TS-EYFP-ES, D). (E–G) Radiant energy thresholds (E), P1-N1 amplitudes (F), and P1 latencies (G) of oABR using 1 ms, 10 Hz pulses at 473 nm light for newly engineered ChR, $n = 18$ mice for f-ChR2 TC, $n = 9$ mice for Chronos LC, shown in comparison with published data replotted for Chronos ($n = 12$ mice, obtained with 1 ms, 10 Hz pulses at 473 nm light, from Keppeler et al, 2018) and for f-Chrimson ($n = 5$ mice, obtained with 0.4 ms, 20 Hz pulses at 594 nm light, from Zerche et al, 2023). (F, G) Blue-light activated (473 nm, f-ChR2 TC, Chronos LC, and Chronos) oABRs were measured using a maximum laser-output calibrated in a radiant flux range of 38 to 45.6 mW, while for orange light (594 nm, f-Chrimson) a radiant flux of ~12 mW was applied. Data were analyzed as mean ± SD. Center lines represent median values. Boxes show the 25th and 75th percentile and error bars depict minima and maxima. ***$p = 0.0002$ (E); $p = 0.0004$ (F); $p = 0.0002$ (G: f-ChR2 TC vs. Chronos), $p = 0.0005$ (G: Chronos vs. f-Chrimson); **$p = 0.0051$ (E: f-ChR2 TC vs. Chronos LC), $p = 0.0073$ (E: Chronos vs. f-Chrimson), $p = 0.001$ (F: f-ChR2 TC vs. Chronos), $p = 0.0014$ (F: Chronos LC vs. f-Chrimson), $p = 0.0025$ (F: Chronos vs. f-Chrimson), p = 0.0076 (G: f-ChR2 TC vs. Chronos LC), $p = 0.0058$ (F: Chronos LC vs. f-Chrimson); *$p = 0.0323$ (E) were obtained by unpaired Kruskal–Wallis test each adjusted for multiplicity by Dunn's correction. (H) P1-N1 amplitudes as a function of radiant energy for all oABR measurements for f-ChR2 TC (blue) for $n = 18$ and Chronos LC (violet) for $n = 9$ at 1 ms, 473 nm light pulses of 10 Hz (bold: mean ± SD; faint: all measurements), binned per ChR for mean values in 5 μJ intervals (in bold). (I, J) P1-N1 amplitudes (I) and normalized P1-N1 amplitudes (J) of oABR at varying repetition rate using 1 ms pulses at ~38 to 45.6 mW (bold: mean; faint: all measurements) for $n = 18$ mice for f-ChR2 TC, $n = 9$ mice for Chronos LC, as well as for Chronos, $n = 10$ mice. For f-Chrimson, a radiant flux of ~12 mW with 0.4 ms pulses was applied, $n = 4$ mice. Stimulation rate applied in a range of 20 to 400 Hz is shown. Chronos and f-Chrimson data are replotted from Keppeler et al, 2018 and Zerche et al, 2023, respectively. (K, L) P1-N1 amplitudes (K) and peak-normalized P1-N1 amplitudes (L) of oABR for varying pulse durations using pulses at 10 Hz in a range of 38 to 45.6 mW (bold: mean; faint: all measurements) for $n = 17$ mice for f-ChR2 TC, $n = 9$ mice for Chronos LC. Stimulation length applied is shown in a range of 0.02 to 4 ms. Data were presented as mean ± SD. ****$p = 6.40e-7$ (K: P1-N1-amplitude) and ***$p = 0.0006$ (K: pulse duration) are obtained by unpaired two-tailed Mann–Whitney U-test analyzing the peak values. Panel (A) was created with BioRender.com. Source data are available online for this figure.

f-Chrimson (replotted from Keppeler et al, 2018 and Zerche et al, 2023). Finally, we probed oABRs at varying stimulus lengths using saturating radiant flux (~38 to 45.6 mW, at 10 Hz stimulation, Figs. 3K,L and EV2B). Both data arrays patterned a bell-shaped relationship, indicating the optimal stimulus duration (Fig. 3K,L). The optimal stimulus duration was significantly shorter for f-ChR2 TC (Fig. 3K: $0.42 \pm 0.13$ ms, $n$ 17 vs. $0.91 \pm 0.80$ ms, $n = 9$; $p = 0.0006$) compared to Chronos LC. Again, oABRs with f-ChR2 TC showed higher P1-N1 amplitudes (Fig. 3K: $17.49 \pm 4.57$ μV, $n = 17$ vs. $2.17 \pm 1.13$ μV, $n = 9$; $p < 0.0001$) than with Chronos LC.

## Analyzing the expression of engineered blue-light activated ChRs in SGNs

Next, we investigated expression of all three engineered ChR variants by confocal microscopy of immunolabelled mid-modiolar cryosections (16 μm, Figs. 4 and EV3; Appendix Fig. S4). We found robust f-ChR2 TC expression in SGNs throughout all turns of the left, injected cochleae (Fig. 4A). We assessed SGN density by targeting all SGN subtypes staining for parvalbumin in both cochleae per animal (Fig. EV3A,B), which typically showed ChR expression also in the non-injected ear, most likely owing to AAV spread via the cerebrospinal fluid (Kho et al, 2000), and in comparison, to cochleae of non-injected mice. We determined the fraction of SGNs expressing f-ChR2 TC by co-immunolabelling for EYFP (Fig. EV3C,D). We found a f-ChR2 TC expression rate of $83.48 \pm 5.573\%$ of the SGNs averaged for all turns of the injected cochleae (Fig. 4B; $n = 8$ injected cochleae). However, we also observed a significant reduction of SGN density for f-ChR2 TC compared to non-injected, non-treated control (Fig. EV3; $p = 0,0458$; $n = 9$ non-injected, non-treated cochleae, ordinary one-way ANOVA, corrected for multiple comparison by Bonferroni's post hoc test). Further, we observed a mean expression rate of $63.13 \pm 7.352\%$ for Chronos LC ($n = 6$ injected cochleae) and of $34.77 \pm 10.32\%$ for f-Chronos ($n = 5$ injected cochleae; Fig. 4B). For Chronos LC and f-Chronos, we also observed a significant

reduction of SGN density compared to the non-injected, non-treated cochleae (Fig. EV3; $p < 0.0001$, ordinary one-way ANOVA, corrected for multiple comparison by Bonferroni's post hoc test). SGN loss might have resulted from the intracochlear pressure injection at a young age. In addition, we cannot rule out potential cytotoxicity due to proteostatic stress (Stone et al, 2025) or immune response following AAV-mediated ChR expression. Further, histological and immunohistochemical investigation of cochlear paraffin sections can be found in Appendix Fig. S5. In brief, we observed neuropathological changes in SGNs but did not find evidence for a prevailing adaptive or innate immune response.

Next, we evaluated ChR membrane expression via line profile analysis assessing the distribution of EYFP fluorescence in individual SGN somata of the injected animals in the same cryosections (Fig. 4C–E). We observed a significantly higher ratio of membrane vs. intracellular EYFP fluorescence for f-ChR2 TC expressing SGNs to those expressing Chronos LC and f-Chronos (Fig. 4C; both $p < 0.0001$, Kruskal–Wallis test, post hoc corrected by Dunn's). The superior plasma membrane expression of f-ChR2 TC was evident also from inspection of peak-normalized fluorescence intensity (Fig. 4D; f-ChR2 TC: $N = 8$ injected cochleae, $n = 220$ cells), where a broader intracellular immunofluorescence shoulder was found for both Chronos variants (Fig. 4D; Chronos LC: $N = 6$ injected cochleae, $n = 385$ cells; f-Chronos: $N = 5$ injected cochleae, $n = 150$ cells). Differences in fluorescence distribution patterns between f-ChR2 TC and Chronos LC can be appreciated from representative confocal sections in Fig. 4E. Additionally, differences in EYFP-immunofluorescence intensity for these two ChRs are illustrated in Appendix Fig. S4.

## Optogenetically driven transmission at cochlear nucleus synapses

In order to further scrutinize the utility of f-ChR2 TC for optogenetic control of the auditory pathway, we turned to patch-clamp recordings from the cochlear nucleus. Here, we

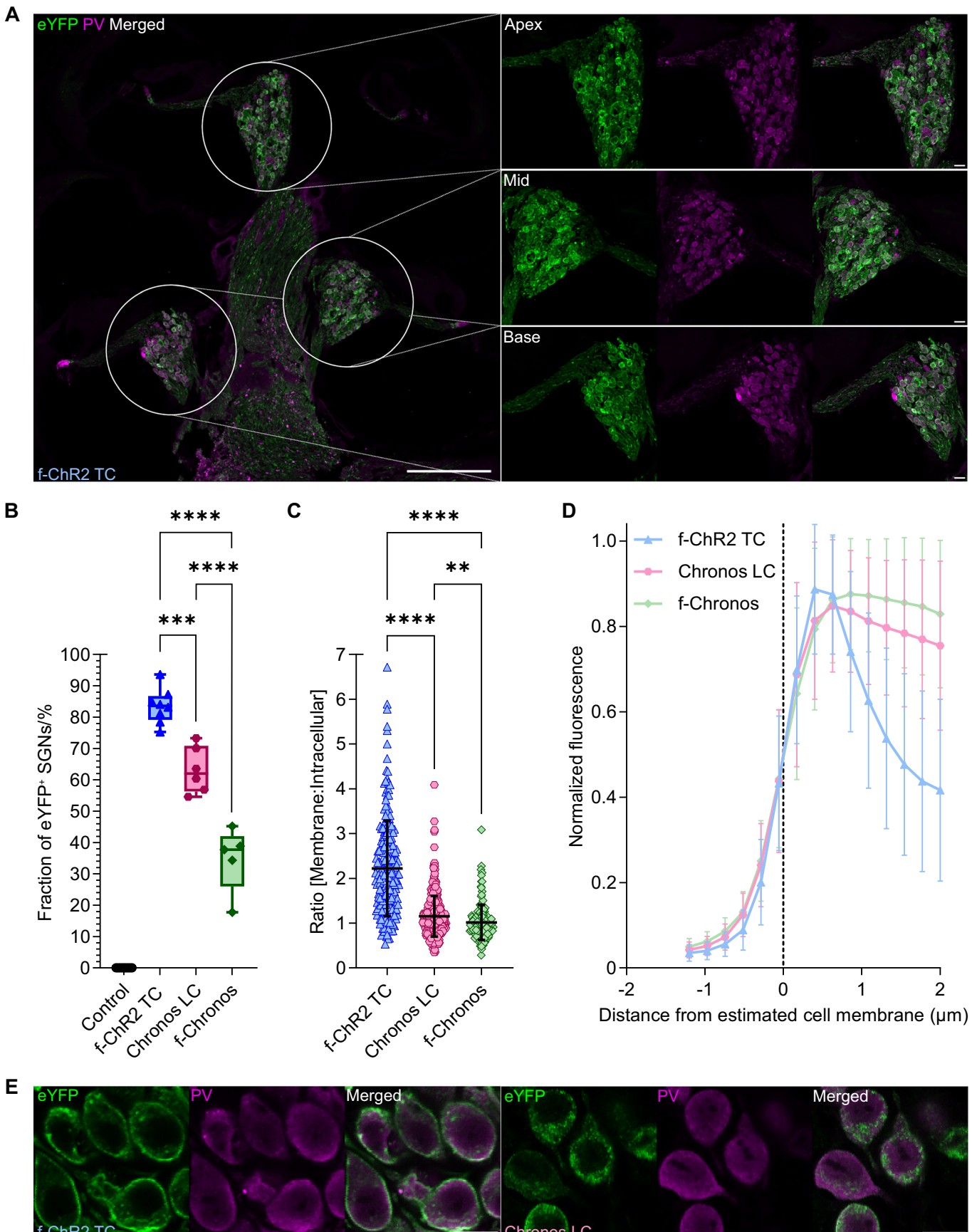

◀ **Figure 4. Immunohistochemical analysis of AAV-injected mouse cochleae expressing newly engineered blue-light sensitive ChRs with optimized properties.**

(A) Overview image (left) of a 20 x-scanned exemplary, immunolabelled mid-modiolar cochlea cryosection following postnatal injection of AAV2/9_hSyn_ChR2(T159C/F219Y)-EYFP: robust expression of f-ChR2 TC in SGNs across all cochlear turns (Apex, Mid, Base), zoom in (right): 40 x magnification. Staining: anti-parvalbumin as context marker, anti-GFP detecting ChR-EYFP, and merged channels, shown from left to right respectively. Scale bar at 20 x = 200 µm, at 40 x = 20 µm. (B) Box plot showing the ratio of EYFP-positive and parvalbumin-positive SGNs as ChR-expression rate (%). Data of all cochlear turns in the left cochlea for f-ChR2 TC (blue; $n = 8$), Chronos LC (violet; $n = 6$), and f-Chronos (green, negatively screened in oABRs; $n = 5$), are shown in comparison to non-injected left cochlea of littermates (black; $n = 9$): Center lines represent median values, boxes show the 25th and 75th percentile and error bars depict minima and maxima. ****$p = 1.45e-8$ (B: f-ChR2 TC vs. f-Chronos), $p = 3.83e-5$ (B: Chronos LC vs. f-Chronos); ***$p = 0.0004$ (B) by ordinary one-way ANOVA corrected for multiple comparison with Bonferroni's. (C-E) Line profile analysis of left cochleae used in (B) for anti-parvalbumin and anti-GFP immunofluorescence. Violin plots (C) and line graphs show (D) the ratio of the EYFP signal at the estimated membrane vs. the intracellular signal (C) and peak-normalized fluorescence intensity (D) per candidate. Plots show means ± SDs for f-ChR2 TC (blue; $n = 220$ cells of eight cochleae), Chronos LC (violet; $n = 385$ cells for six cochleae), and f-Chronos (green, negatively screened in oABRs; $n = 150$ cells for five cochleae). ****$p = 7.70e-41$ (C: f-ChR2 TC vs. Chronos LC), $p = 1.80e-41$ (C: f-ChR2 TC vs. f-Chronos); **$p = 0.0057$ (C) by Kruskal–Wallis test corrected for multiple comparison with Dunn's. (E) Fluorescence distribution of f-ChR2 TC (left) and Chronos LC (right) is shown at 40 x for anti-GFP, anti-parvalbumin and a composite image in a horizontal order. Scale bar = 10 µm. Source data are available online for this figure.

optogenetically drove presynaptic endbulbs of Held formed by SGNs using trains of light pulses and recorded optically evoked excitatory postsynaptic currents (oeEPSCs) by whole-cell patch clamp recordings from bushy cells in acute sagittal brainstem slices (for cell type identification refer to Methods, Appendix Figs. S6 and S7). These slices were obtained from postnatally AAV-injected (P5-P9) C57BL/6 wild-type mice after an expression period of 15 ± 0.4 days on average, as described before (Özçete and Moser 2021; Hain and Moser 2024).

Long light pulse stimulations (≥500 ms) were employed to test for potential direct transduction of bushy cells, which was revealed by a depolarizing photocurrent (Fig. 5A). Further, immunohistochemical labeling of VGluT1 (presynaptic marker), Homer1 (postsynaptic marker), and EYFP (for ChR expression) corroborated that some bushy cells also exhibited somatic ChR expression (Fig. 5A$_2$), while in other cases ChR expression was limited to the presynaptic SGN terminals (Fig. 5B$_2$). ChR-negative bushy cells exclusively showed big, sharp oeEPSC at the beginning of the light pulse (Fig. 5B).

The number of excitatory auditory nerve fibers converging onto each cell in the cochlear nucleus varies (Cao and Oertel, 2010; Mendoza Schulz et al, 2014). The convergence of inputs can be analyzed by examining the EPSC size of the postsynaptic cell upon electric or optogenetic recruitment of individual inputs, in an all-or-none manner. Given ChR expression differences and the variability of SGN membrane resistance, capacitance and spiking threshold, by reducing the light irradiance, fewer inputs are recruited, which is reflected by smaller oeEPSC amplitudes eventually arriving at monosynaptic input (single endbulb synapse).

Here, presynaptic inputs were optogenetically driven with 50 stimuli of 1 ms pulsed light trains at different irradiances (0.07 to 40 mW/mm$^2$) and at constant low frequency (10 Hz). We additionally applied 50 stimuli (1 ms) at different frequencies for saturating irradiances (40 mW/mm$^2$). Upon reduction of the applied irradiance, we observed a stepwise decrease of the amplitude for the first oeEPSC and an increase in synaptic delay (Fig. 5C$_1$). We observed a decrease in the EPSC probability with higher stimulation frequency (Fig. 5D) or lower irradiance (Fig. 5C$_2$), which was more pronounced at the end than at the beginning of the pulse train. Moreover, we found substantial heterogeneity of endbulb synaptic transmission to bushy cells: some cells showed oeEPSCs at 200 Hz (Fig. 5D$_{1-2}$), while others already failed to stably

respond at 50 Hz (Fig. 5D$_3$). Quantification of oeEPSC probability versus irradiance and frequency of stimulation shows that bushy cells that are more light-sensitive in terms of higher oeEPSC probability and shorter synaptic delay also follow higher frequencies of stimulation (Fig. 5E–G). We observed that f-ChR2 TC-mediated endbulb synaptic transmission across the full train of 50 light stimuli on average had an oeEPSC probability lower than 80% at 50 Hz of stimulation (Fig. 5G). Given the fast-closing kinetics of f-ChR2 TC at physiological temperature ($\tau_{off}$ ~5 ms, Table EV2), we reason that limited expression and photocurrent desensitization of f-ChR2 TC led to the observed heterogeneity of endbulb synaptic transmission to bushy cells and hindered most endbulb synapses to follow stimulation frequencies of more than 100 Hz. When analyzing the oeEPSC probability in bins of ten stimuli, we found the success rate at the beginning of the light stimulation (Stimuli 1–10) to provide an oeEPSC probability at 300 Hz of >80% in half of the bushy cells recorded (Fig. 5H,I). Quantification of the full data set indicates a wide range of optogenetically presynaptic activation patterns for bushy cells, as well as for stellate cells (Fig. EV4). A direct correlation between maximal frequency of stimulation and light sensitivity was found for both cell types (Fig. EV4C–F). The synaptic delay serves as an indirect measure of the strength of the optogenetic presynaptic depolarization: synaptic delay is shortened at higher irradiances employed (Fig. 5F). Further, endbulb synapses with lower light-sensitivity frequently display a prolonged synaptic delay compared to those of higher light sensitivity (Fig. 5E,F), which likely indicates lower ChR expression in these SGNs. However, low oeEPSC probability was not always correlated to long latencies (Appendix Fig. S8B). A quantification of the latencies at different frequencies (Appendix Fig. S8) showed a range of 1 to 2 ms and an increase throughout the pulse train (Appendix Fig. S8A)

Next, similarly to changing electrical shock strength for recruiting different numbers of transmitting endbulbs (Cao and Oertel 2010), we modified the irradiance in steps to vary the number of optogenetically triggered presynaptic terminals. For simplicity we selected six exemplary bushy cells with a high, a medium, and a low oeEPSC probability at different frequencies of stimulation (Appendix Fig. S9). Here, the reduction of irradiances, resulted in a decrease of amplitude for oeEPSC with the appearance of extra peaks or kinks for the first oeEPSC at different latencies, as an indicator of multiple inputs (Appendix Fig. S9). Apart from differences in oeEPSC amplitude and latency, we also quantified the oeEPSC charge per light pulse at

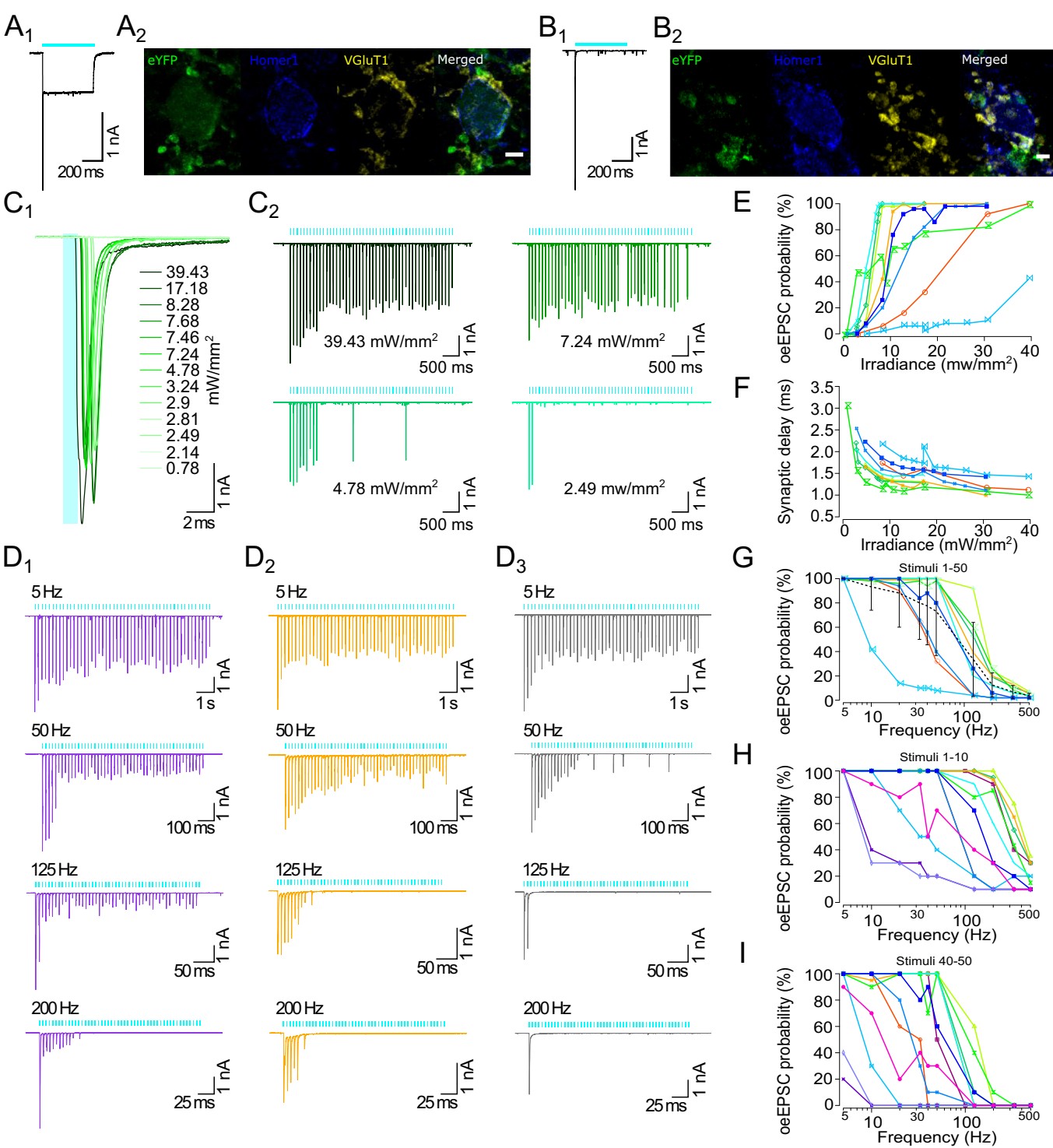

different irradiances applied, to observe the different steps corresponding to the different endbulb inputs. Here, the overlap of small inputs and continuous gradual increment of charge resulted in difficulties to clearly separate the number of inputs per bushy cell (e.g., cells C3.1 and C2.2, Appendix Fig. S10A,E respectively). The amplitude of the steps was not strictly related to the number of steps, but cells with a larger number of inputs had larger synaptic charge (Appendix Fig. S10E,F).

In addition, we used immunohistochemistry to verify that neonatal injections of AAV2/9 with an expression period of 15 days, used for the slice measurements, were also sufficient for robust ChR expression in SGNs. As shown in Appendix Fig. S11, f-ChR2 TC was expressed in SGNs across all cochlear turns (apex, middle and base), with no apparent cell loss shown by a Parvalbumin immunolabelling.

◄

**Figure 5. Optogenetic control of endbulb synapses formed by spiral ganglion neurons and bushy cells in the cochlear nucleus.**

($A_1$, $B_1$) Long light stimulation (500 ms, 488 nm, ~40 mW/mm²) indicated a stationary photocurrent in a patch-clamped bushy cell expressing f-ChR2 TC ($A_1$) or is exclusively receiving optogenetically driven presynaptic inputs ($B_1$), also confirmed by immunohistochemistry ($A_2$, $B_2$), respectively. ($A_2$, $B_2$), Maximal z-projections of confocal image stacks of sagittal slices fixed and immunolabelled for EYFP (green) for f-ChR2 TC localization, and context markers Homer1 (blue), a postsynaptic scaffold protein of excitatory synapses, and VGluT1 (yellow) as presynaptic vesicular glutamate transporter. Scale bar 5 µm. (**C**) Exemplary bushy cell recording showing the light intensity dependence of oeEPSC probability. 50 light pulses of 1 ms were delivered (10 Hz, λ = 488 nm) at different light intensities. ($C_1$) Overlap of oeEPSC1 at different light intensities shows the dependence of amplitude and synaptic delay on light intensity. ($C_2$) Full train of 50 light stimuli show failures occurring during the train upon reduction of irradiance. (**D**) Exemplary traces of three different bushy cells at different stimulation frequencies (pulse length: 1 ms, λ = 488 nm, ~40 mW/mm²). (**E**) oeEPSC probability at indicated irradiances upon photostimulation at a frequency of 10 Hz. (**F**) Dependence of synaptic delay on irradiance. (**G**) Dependence oeEPSC probability on frequency (λ = 488 nm, ~40 mW/mm²). The average oeEPSC probability is shown as a black dotted line. (**E–G**) Individual bushy cells are represented using different icons and colors (n = 9). Error bars show SD. (**H**) Dependence of oeEPSC probability on frequency for stimuli 1 to 10 (n = 12). (**I**) Dependence of oeEPSC probability on frequency for stimuli 40 to 50 (n = 12). Source data are available online for this figure.

## AAV-mediated expression of f-ChR2 TC enables efficient optogenetic stimulation of the auditory nerve by LED-based optical cochlear implants in Mongolian Gerbils

In order to assess the utility of f-ChR2 TC for translational studies on optogenetic hearing restoration, we turned to the Mongolian Gerbil and LED-based optical cochlear implants (oCIs). The gerbil offers a relatively large cochlea (~2.5x smaller than the human cochlea) and therefore has served as a translational animal model for demonstrating and characterizing optogenetic hearing restoration (Wrobel et al, 2018; Dieter et al, 2019; Dieter et al, 2020b; Huet et al, 2021; Michael et al, 2023; Thirumalai et al, 2025). In a preliminary set of experiments, we administered AAV-PHP.S carrying f-ChR2 TC under control of the human synapsin promoter to the gerbil cochlea at p7-9 (Methods). At ~3 months of age, a bullostomy was performed and oABRs were recorded in response to 1 ms light pulses delivered at 17 Hz from a 200 µm optical fiber (coupled to a 488 nm laser) placed into the round window niche. In three out of four gerbils, oABRs were readily evoked at low radiant flux threshold (0.30, 0.42, 0.45 mW) and with large amplitudes that saturated at 10 mW (Appendix Fig. S12). In the oABR-positive animals we proceeded to insertion of multichannel oCI based on blue Cree-LEDs (Keppeler et al, 2020) and a craniotomy for multielectrode array recordings of oCI-evoked multi-unit activity in the contralateral central nucleus of the inferior colliculus (IC, Fig. 6A, (Dieter et al, 2019)). The position of the electrodes along the tonotopic map of the IC was mapped by acoustic stimulation of the ipsilateral ear (Fig. 6B). Multi-unit activity elicited by individual LEDs was observed at sub-mW radiant flux by individual LEDs (Fig. 6C), similar to findings with ChReef, an advanced ChR variant activated by green light (Alekseev et al, 2025). Stimulation by LEDs placed at different tonotopic positions from the mid-cochlea (LED1) to the base (LED10) elicited spatially confined IC activity in the expected frequency ranges (exemplary spatial tuning curves in Fig. 6D–F). In keeping with the faster closing f-ChR2 TC (Fig. 1, compared to 60 ms for ChReef), synchronous IC firing (vector strength ≥0.5) was observed for stimulation rates up to 100 Hz (Fig. 6H,I, compared to up to 50 Hz for ChReef (Alekseev et al, 2025).

## Discussion

Since the discovery of ChRs, optogenetics has revolutionized life sciences (Nagel et al, 2002; Nagel et al, 2003; Boyden et al, 2005). Moreover, optogenetic therapies are emerging such as for hearing restoration (Dieter et al, 2020a; Huet et al, 2024), vision restoration (Busskamp et al, 2010; Sahel and Roska 2013; Kleinlogel et al, 2020; Sahel et al, 2021), and cardiac and other muscular disorders (Bruegmann et al, 2010; Bruegmann et al, 2015; Bruegmann et al, 2018; Vogt et al, 2021). Efficient optogenetic control of fast-spiking neurons such as SGNs to achieve near-physiological firing rates is challenged by limited charge transfer during the short channel open time in ChRs with fast channel closing kinetics (Keppeler et al, 2018). While fast channel closing kinetics is crucial for driving SGNs at high firing rates, the required high light intensity elevates the energy consumption of the oCI and bears the risk of phototoxicity, in particular when blue light is employed (Huet et al, 2024). Efforts to overcome the bottleneck of limited charge transfer have been made by balancing key ChR properties (Berndt et al, 2011; Mager et al, 2018; Bali et al, 2021). Short channel open times can be offset partially by enhanced membrane expression, e.g., by the use of trafficking sequences derived from the inward rectifying potassium channel Kir2.1 (Gradinaru et al, 2010; Keppeler et al, 2018; Bali et al, 2021). Nevertheless, massive ChR expression might result in proteostatic and, thus, cytotoxic stress. ChR variants with balanced channel closing kinetics, robust plasma membrane targeted expression, and low photocurrent desensitization are desirable for efficient and safe neurostimulation at high rates (Huet et al, 2024). Therefore, we aimed to optimize blue light-activated ChRs in this study, which would also be suitable for the later use with existing blue LED-based oCI prototype systems (Goßler et al, 2014; Dieter et al, 2020b; Jablonski et al, 2020; Keppeler et al, 2020).

Here, we report on optimized ChRs and their characterization in vitro and in vivo. We generated f-Chronos, the - to our knowledge - fastest ChR to date and corroborated the hypothesis that a previously described helix F mutation (Mager et al, 2018) speeds channel closing via the acceleration of protonation reactions that govern open-to-closed transition. This finding suggests that ChR variants in which channel closing is rate limited by protonation, apparent by the voltage-dependence of their closing kinetics, may generally be susceptible to helix F mutation-mediated acceleration. The L149C mutation (L132C mutation in ChR2; Kleinlogel et al, 2011), reduced photocurrent desensitization and slowed channel closing in Chronos (Chronos LC). Both f-Chronos and Chronos LC showed suboptimal plasma membrane expression, which could not be improved by C-terminal fusion of Kir2.1 plasma membrane trafficking sequences. Finally, the combination of the T159C mutation (Berndt et al, 2011) and the F219Y mutation

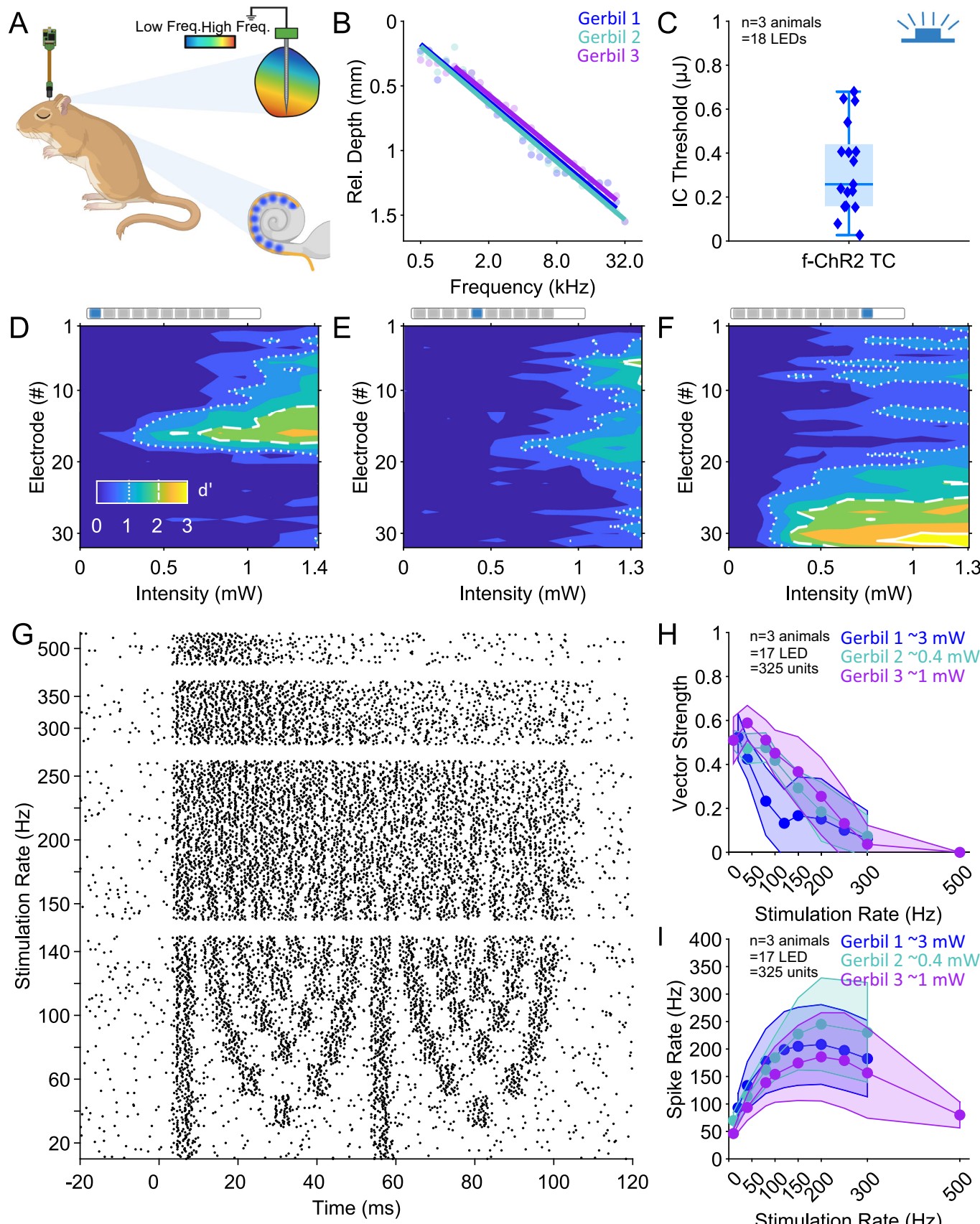

◀ **Figure 6. Characterizing the optogenetic activation of the auditory pathway by f-ChR2 TC and LED-based cochlear implants using inferior colliculus recordings in Mongolian gerbils.**

(A) Experimental design. Following placement of a 32-electrode array into the central nucleus of the inferior colliculus (IC) for recording of multi-unit activity, an LED-based optical cochlear implant (oCI) with up to ten channels was implanted into scala tympani via the round window. (B) Placement of the multielectrode array along the tonotopic axis in the IC. Depth of the best electrode for the presented sound frequencies relative to the most dorsal recording electrode. Tonotopic slopes were linearly fitted for each animal (indicated by different colors) after outliers were removed. $n = 75$ pure tones in three gerbils. (C) Optical IC thresholds for single pulses (1 ms) delivered from individual LEDs. Threshold is defined as the LED radiant energy at which multi-unit activity reaches d' ≥1, $n = 17$ LEDs in three gerbils. Boxes represent the 25th percentile, median, and 75th percentile; whiskers extend to the minimum and maximum values. (D–F) Exemplary spatial tuning curves in response to 1 ms light pulses delivered from LED1 (D), LED5 (E), and LED10 (F) placed at different tonotopic positions of the cochlea of gerbil 3 (around 3150 μm (LED1), 1750 μm (LED 5), and 0 μm (LED 10) from the round window). White contour lines represent d' of 1 (dotted), 2 (broken), and 3 (solid). (G) Raster plot of an exemplary multi-unit response to 100 ms pulse trains, assembled from 1 ms pulses at 1 mW intensity, emitted by a single LED. Repetition rates varied between 20 and 500 Hz. (H) Vector strength and (I), Spike Rate of active multiunits for individual animals (indicated by different colors) as a function of stimulation rate. 100 ms pulse trains assembled from 1 ms pulses were presented at around 3/0.4/1 mW (Gerbil 1/2/3) by individual LEDs. Dots indicate the average, and shaded areas indicate the standard deviation for active multiunits in individual animals. $n = 325$ multiunits by 17 LEDs in three gerbils. Source data are available online for this figure.

(Mager et al, 2018) resulted in a ChR2 variant (f-ChR2 TC) that combines balanced channel closing kinetics with good plasma membrane targeted expression. The experiments in primary cultures of rat hippocampal neurons confirmed the importance of high photocurrent ChRs with well-balanced kinetics for efficient neurostimulation. The ultrafast kinetics and the suboptimal plasma membrane expression of f-Chronos accordingly resulted in a very low success rate for neurostimulation. Despite suboptimal plasma membrane expression, Chronos LC enabled robust neurostimulation, comparable to f-ChR2 TC, in cultured neurons, which can likely be attributed to its balanced kinetics and reduced photocurrent desensitization.

In order to investigate high-rate stimulation, we turned to experiments in SGNs, fast spiking auditory neurons. Using optogenetically driven synaptic transmission (oeEPSC) from presynaptic endbulbs of Held terminals of SGNs to bushy cells in AVCN showed that f-ChR2 TC enabled transient high-rate neurostimulation. Limitations explaining the oeEPSC failure during sustained high-rate neurostimulation include limited expression levels of f-ChR2 TC in SGN terminals, as indicated by the correlation of oeEPSC probability and light sensitivity and the increased synaptic delay compared to electrically triggered EPSCs (Chanda and Xu-Friedman 2010). Decreasing peak photocurrents arising from f-ChR2 TC desensitization likely contributed to the suboptimal performance during sustained photostimulation. In addition, potential depolarization block resulting from prolonged SGN photodepolarization due to the limited closing kinetics of f-ChR2 TC may have contributed to suboptimal sustained photostimulation at high frequencies. Moreover, high structural and functional variability of the synapses, involving differences in release probability and the number of docked synaptic vesicles (SVs) per active zone, might have also impacted on oeEPSC probability (Oleskevich et al, 2000; Wichmann and Kuner 2022).

Our in vivo analysis of the optogenetic activation of the mouse auditory pathway identified f-ChR2 TC as the most promising among the three new blue-light activated ChR variants. Likely due to limited charge transfer and suboptimal plasma membrane expression, we could not achieve optogenetic activation of the auditory pathway in f-Chronos injected mice despite ~30% SGN expression, which is above the expression rate supporting oABRs for other ChRs (e.g., Wrobel et al, 2018; Huet et al, 2021). f-ChR2 TC outperformed Chronos LC in eliciting oABRs at lower light intensity and with larger maximal oABR amplitudes. f-ChR2 TC

drove SGN firing at ≥100 spikes per second with a favorable energy budget. With an average activation threshold of 1.44 μJ and detectable oABRs at stimulation rates ≥300 Hz in mice, f-ChR2 TC outperforms all so far characterized blue-light activated ChR variants and is comparable to the fast red-light activated ChR variant f-Chrimson (Hernandez et al, 2014; Keppeler et al, 2018; Mager et al, 2018; Wrobel et al, 2018; Michael et al, 2023; Mittring et al, 2023; Zerche et al, 2023). This makes f-ChR2 TC a promising ChR for preclinical characterization of optogenetic hearing restoration using LED-based oCI prototype systems in more translational animal models with larger cochleae, such as Mongolian gerbils and non-human primates.

Indeed, when transferred to Mongolian gerbils, a first set of experiments showed a good match of f-ChR2 TC-mediated SGN stimulation to the radiant fluxes provided by LED-based oCIs. oABR thresholds of gerbils following postnatally cochlear injection of AAV-PHP.S- f-ChR2 TC tended to be in the sub-μJ range. Following implantation of multichannel LED-based oCIs, individual LEDs readily elicited multi-unit activity in the IC with sub-μJ thresholds, confirming light-efficient SGN stimulation mediated by f-ChR2 TC. The trend to lower thresholds in this preliminary data set compared to the recordings in mice might reflect subtle differences in AAV-mediated ChR-expression (e.g., due to different vectors: AAV-PHP.S vs. AAV 2/9). However, while, preliminary, we consider finding sub-μJ thresholds in gerbils promising for using f-ChR2 TC in animal models with a larger cochlea. Moreover, the IC corroboration of neurostimulation at high rates with low light requirement underlines the potential of f-ChR2 TC also in comparison to the recently described ChRmine mutant ChReef. However, the 35-fold difference in the employed AAV titers should be taken into account when comparing the findings obtained with ChReef and f-ChR2 TC in the IC recordings from gerbils.

f-ChR2-TC is a promising candidate for preclinical studies of optogenetic hearing restoration with blue μLED oCI implementations (Goßler et al, 2014; Dieter et al, 2020b; Keppeler et al, 2020; Huet et al, 2024). Future studies can employ f-ChR2 TC for preclinical assessment in gerbils and non-human primates of the efficacy of optogenetic hearing restoration with μLED oCIs compared to that provided by eCIs. Previous LED work had used CatCh but optogenetic stimulation remained limited, both, in terms of matching light requirement to the range of radiant fluxes provided by individual LEDs (Dieter et al, 2020b, S. 202; Keppeler et al, 2020), and in coding of temporal information (Wrobel et al,

2018; Dieter et al, 2019; Michael et al, 2023). Due to their strong impact on transduction rate, vitality and ChR expression of SGNs, the further optimization of several factors such as the route of administration and the choice of suitable AAV serotype and titer remains an important objective for developing safe and reliable optogenetic control of SGN firing.

Moreover, high-frequency stimulation of SGNs using blue light still poses a risk of phototoxicity. A recent study has demonstrated the potential of the engineered green light-activated ChR variant ChReef, to mitigate this issue (Alekseev et al, 2025). ChReef enables sustained control of excitable cell activity at low light intensities, owing to its good plasma membrane targeting, minimal photocurrent desensitization, and comparatively high unitary conductance. Yet, the temporal fidelity of SGN activation is limited by ChReef's slow channel-closing kinetics. The generation and comprehensive characterization of both natural and engineered ChRs will further fuel the development of advanced, kinetically balanced variants that can be safely integrated into future blue- or green-μLED-based oCIs.

As an outlook into future applications, we consider f-ChR2 TC combined with a red-light activated ChR such as f-Chrimson a good candidate to expand the current dynamic range limitation of oCI coding of sound information (Huet et al, 2024) by SGN subtype specific activation via dual color optogenetics.

## Methods

### Reagents and tools table

| Reagent/resource | Reference or source | Identifier or catalog number |
|---|---|---|
| **Experimental models** | | |
| NG108-15 cells | ATCC, HB-12377TM, Manassas, USA | HB-12317 |
| Wistar rats (*R. norvergius*) | Charles River | 003Wistar |
| C57BL/6 (*M.musculus*) | Breeding facility of the University Medical Center Göttingen | |
| Mongolian gerbils (*Meriones unguiculatus*) | Breeding facility of the University Medical Center Göttingen | |
| **Recombinant DNA** | | |
| pcDNA3.1(-) | Invitrogen, Carlsbad, USA | |
| pHelper plasmid | TaKaRa, USA | |
| trans-plasmid with the AAV2/9 capsid | Penn Vectore Core, USA | |
| cis-plasmid with newly engineered blue-light sensitive ChR | This study | |
| **Antibodies** | | |
| chicken anti-GFP | Abcam, Cambridge, UK | ab13970 |
| Guinea pig anti-VGLUT1 | Synaptic Systems | Cat Nr. 135304 |
| Rabbit anti-Homer1 | Synaptic Systems | Cat Nr. 160002 |

| Reagent/resource | Reference or source | Identifier or catalog number |
|---|---|---|
| Guinea pig anti-parvalbumin | Synaptic Systems | Cat Nr. 195004 |
| Rabbit anti-calretinin | Swant | Cat Nr. 7697 |
| Mouse anti-ATP1A3 | Novus Biologicals | Cta. Nr. NB-300-540 |
| Goat anti-chicken 488 IgG (H + L) | Invitrogen | A-11039 |
| Goat anti-guinea pig 568 IgG (H + L) | Invitrogen | A-11075 |
| Goat anti rabbit 633 IgG (H + L) | Invitrogen | A-21070 |
| Goat anti rabbit 647 IgG (H + L) | Invitrogen | A21244 |
| Goat anti-guinea pig 647 IgG (H + L) | Invitrogen | A21450 |
| Goat anti-mouse 568 IgG (H + L) | Invitrogen | A10037 |
| Rabbit anti-CD3 | Dako Deutschland GmbH, Hamburg | A0452 |
| Rabbit anti-Iba-1 | GeneTex, Biozol Diagnostica Vertrieb GmbH, Eching, Germany | GTX101495 |
| **Oligonucleotides and other sequence-based reagents** | | |
| PCR primers | This study | Appendix Table S2 |
| Chronos-EYFP (*Stigeoclonium helveticum*) | Klapoetke et al, 2014 | accession number: KF992040 |
| ChR2-EYFP (*Chlamydomonas reinhardtii*) | Nagel et al, 2003 | accession number: AF461397 |
| **Chemicals, enzymes, and other reagents** | | |
| Polyethylene glycol 8000 | Thermo Fischer Scientific | 10065830 |
| NaCl | Sigma | S9888 |
| Tris | Sigma | T1503-1KG |
| MgCl$_2$ | Sigma | 63020-1 L |
| Salt-activated nuclease | Arcticzymes, USA | 70900-202 |
| Iodixanol | Optiprep, Axis Shield, Norway | 1022361 |
| Sterile phosphate-buffered saline (PBS) | Gibco, Thermo Fisher Scientific, Waltham, USA | 14190-094 |
| Pluronic F- 68 | Gibco, Germany | 24040-032 |
| Naica PCR reaction mix | Stilla, France | R10056 |
| Silver staining | Pierce, Germany | 24612 |
| Alexa 647 | Invitrogen | 11570266 |
| Eva Green | Biotum | #31000-T |
| Gel electrophoresis, 4 to 12% Tris–Glycine | Novex™, Thermo Fisher Scientific | XP04122Box |
| Dulbecco's Modified Eagle Medium (DMEM) | Sigma, St. Louis, USA | D5796-24x500ml |
| Fetal calf serum (FBS gold) | PAN Biotech | P30-3033 |
| Penicillin/streptomycin | Gibco, Germany | 15070063 |

| Reagent/resource | Reference or source | Identifier or catalog number |
| --- | --- | --- |
| Lipofectamine™ LTX Reagent | Invitrogen, Carlsbad, USA | 15338100 |
| EGTA | Carl Roth | 3054.3 |
| HEPES | Sigma-Aldrich | H3375 |
| $CaCl_2$ | Sigma-Aldrich | 21115 |
| $MgCl_2$ | Sigma-Aldrich | |
| $CaCl_2$ solution | Honeywell | 21114-1 L |
| $MgCl_2$ solution | Honeywell | 63020-1 L |
| Trypsin/EDTA (0,05%) | Invitrogen | cat. No. 25300-062 |
| Hanks' Balanced Salt solution (HBSS) | Sigma-Aldrich, St. Louis, USA | H9394-500 mL |
| Dulbecco's Modified Eagle Medium (DMEM) | Gibco, Thermo Fisher Scientific, Waltham, USA | 31966-021 |
| Glutamax | Gibco, Thermo Fisher Scientific, Waltham, USA | 35050061 |
| Fetal bovine serum | Gibco, Thermo Fisher Scientific, Waltham, USA | 16140071 |
| Penicillin/streptomycin (P/S) | Gibco, Thermo Fisher Scientific, Waltham, USA | 15140122 |
| Poly-D-lysine | Sigma | P1024 |
| Neurobasal A | Gibco, Thermo Fisher Scientific, Waltham, USA | 10888022 |
| B-27 supplement | Gibco, Thermo Fisher Scientific, Waltham, USA | 17504044 |
| Potassium gluconate | Sigma-Aldrich, St. Louis, USA | P1847 |
| KCl | Sigma-Aldrich, St. Louis, USA | P4504 |
| MgATP | Sigma-Aldrich, St. Louis, USA | A9187 |
| Na3GTP | Sigma-Aldrich, St. Louis, USA | G8877 |
| Glucose | Sigma-Aldrich, St. Louis, USA | G8270 |
| 1,2,3,4-tetrahydro-6-nitro-2,3-dioxo-benzo[f]quinoxaline-7-sulfonamide (NBQX) | Tocris | 0373 |
| D(−)-2-Amino-5-phosphonopentanoic acid (AP-5) | Tocris | 0106/1 |
| Sucrose | Sigma-Aldrich, St. Louis, USA | 84100 |
| $NaHCO_3$ | Sigma-Aldrich, St. Louis, USA | S5761 |
| $NaH_2PO_4.H_2O$ | Merck Millipore | 106346 |
| Na L-ascorbate | Sigma-Aldrich, St. Louis, USA | A4034 |
| Na pyruvate | Sigma-Aldrich, St. Louis, USA | 11360-070 |
| Myo-inositol | Sigma-Aldrich, St. Louis, USA | I5125 |
| Na L-lactate | Sigma-Aldrich, St. Louis, USA | BCBQ4938V |
| Cyanoacrylate glue | Loctite 401, Henkel | |
| $Na_2$ Phosphocreatine | | |
| QX-314 (N-(2,6-dimethylphenyl carbamoylmethyl) triethylammonium chloride | Alomone Labs, Jerusalem, Israel | Q-150 |
| Alexa-568 | Invitrogen | D22912 |

| Reagent/resource | Reference or source | Identifier or catalog number |
| --- | --- | --- |
| Strychnine hydrochloride | Sigma-Aldrich, St. Louis, USA | S8753 |
| Paraformaldehyde | Merck Millipore | 104005 |
| Formaldehyde solution (37%) | Carl Roth GmbH + Co. KG | CP10.2 |
| Goat Serum | Merck Millipore | S26-100ml |
| Triton X-100 | Merck Millipore | 648462 |
| Mowiol 4–88 | Carl Roth, Karlsruhe, Germany | 0718.2 |
| Isoflurane anesthesia | Baxter Healthcare Corporation, Deerfield, Illinois | 26675-46-7 |
| Xylocaine | Aspen Pharna Trading Limited, | PZN: 03839499 |
| Bupivan | Eugia Pharma Limited | PZN. 12675915 |
| Buprenovet | VetViva Richter GmbH | PZN: 18760711 |
| Rimadyl | Zoetis Deutschland GmbH | PZN: 11283981 |
| Metacam | Boehringer Ingelheim | PZN: 08890217 |
| Sterofundin HEG-5 | B. Braun Melsungen AG | PZN: 08609427 |
| Bepanthen Eye and Nose creme | Bayer Vital GmbH | PZN: 01578675 |
| Ethylenediaminetetraacetic acid | SERVA Elctrophoresis GmbH, Germany | 11280.02 |
| Cryomatrix embedding resin | Epredia, USA | 67-690-06 |
| OSTEOSOFT | Merck Millipore | 101728 |
| DAB Map Kit | Roche Diagnostics GmbH, Mannheim, Germany | |
| **Software** | | |
| Matlab | The MathWorks, Inc., Natick, MA, USA | |
| Microsoft Office 365 | Microsoft Corp., Redmond, WA, USA | |
| Igor Pro 6 | Wavemetrics, Portland, OR, USA | |
| Origin 9.0 | OriginLab, Inc., Northampton, MA, USA | |
| GraphPad Prism 10.2.1. | GraphPad Software Inc., La Jolla, CA, USA | |
| ImageJ | NIH, Bethesda, MD, USA | Schindelin et al., 2012 |
| Crystal Reader and Crystal Miner software | Stilla | |
| Patchmaster Next software (version 1.2, HEKA) | Harvard Bioscience Inc. | |
| Oxxius Lasers software (version 2.4) | Oxxius Coherent Inc | |
| ClampFit | Axon Instruments, Union City, USA | |
| Opolette 355 tunable laser system | Opotek Inc., Carlsbad, USA | |

| Reagent/resource | Reference or source | Identifier or catalog number |
|---|---|---|
| Andor SOLIS imaging software (version 4.32.30065.0). | Andor | |
| LASX_Office (1.4.3) | Leica, Hamburg, Germany | |
| Custom graphical interface | Napari | |
| Cheetah recording software | Neuralynx | |
| **Other** | | |
| Amicon filters | EMD | UFC910024 |
| AAV titration kit | TaKaRa/Clontech | 6233 |
| Naica Prism3 reader | Stilla | |
| Axopatch 200B amplifier | Axon Instruments, Union City, USA | |
| DigiData 1320 A interface | Axon Instruments, Union City, USA | |
| Pipette puller (Model P-1000) | Sutter Instruments, Novato, USA | |
| Diode-pumped solid-state lasers (λ = 473 nm) | Changchun New Industries Optoelectronics Tech. Co., Ltd., Changchun, China | |
| Fast computer-controlled shutter Uniblitz LS6ZM2 | VincentAssociates, Rochester, USA | |
| Opolette 355 tunable laser system | Opotek Inc., Carlsbad, USA | |
| Diode-pumped solid-state laser (LBX-488-40-CSB; 488 nm 40 mW,) | Oxxius Coherent Inc | |
| Upright Olympus BX51WI microscope with a x40/0.8 LUMPLFLN40XW objective | Olympus | |
| Andor zyla sCMOS camera | Andor | |
| mCherry HC Filter Set mirror (F36-508 HC-Set mCherry) | AHF - IDEX/Semrock | |
| EGFP HC Filter Set mirror (F36-528 HC-Set EGFP) | AHF - IDEX/Semrock | |
| EPC10 USB HEKA patch clamp amplifier | Harvard Bioscience Inc. | |
| FieldMaxII-TOP laser power meter | Coherent | |
| Temperature controller SC-20 and CL-200A | Warner Instruments | |
| Falcon cell strainer (100 μm) | Corning | 352360 |
| VT 1200 vibratome | Leica Microsystems, Wetzlar, Germany | |
| Borosilicate glass capillaries with filament (GB150F, 0.86 × 1.50 × 80 mm) | Science Products, 154 Hofheim, Germany | |
| Imaging spacer Grace Bio-Labs SecureSeal TM (GBL654008-100EA) | Sigma-Aldrich | |
| Optical fiber (200-μm diameter, 0.39 NA) | Thorlabs | |
| 473 nm laser (MLLFN-473-100) | Changchun New Industry Optoelectronics | |

| Reagent/resource | Reference or source | Identifier or catalog number |
|---|---|---|
| 488 nm laser (LBX-488-100) | Oxxius Coherent Inc | |
| Laser power meter (Solo-2; PM103USB; S140C). | Gentec-EO and Thorlabs | |
| National Instrument data acquisition cards (NI PCI-6229) | National Instruments | |
| Soundproof chamber | Industrial Acoustics | |
| Loudspeaker (Scanspeak Ultrasound; Vifa) | Avisoft Bioacoustics | |
| 0.25-inch microphone (4039; 46BF-1) | Brüel & Kjaer, GRAS | |
| Preamplifier (12AQ) | GRAS | |
| 32-channel linear silicone probe (A1x32-6mm-50-177-A32, 50-μm thickness) | NeuroNexus | |
| Digital Lynx 4S recording system | Neuralynx | |
| LED chips (C460TR2227-S2100) | Cree | |
| Confocal microscope SP8 | Leica, Hamburg, Germany | |
| Micromanipulator (SM-10 compact) | Luigs & Neumann | |
| LEICA CM3050 S Crysotat | Leica Biosystems, Nussloch GmbH 2025 | |
| Automated immunostaining system - Discovery XT | Roche Diagnostics GmbH, Mannheim, Germany) | |

## ChR variant generation

The pcDNA3.1(−) (Invitrogen, Carlsbad, USA) derivatives carrying the humanized DNA sequences of Chronos-EYFP (*Stigeoclonium helveticum* ChR, accession number: KF992040; Klapoetke et al, 2014), ChR2-EYFP (C-terminally truncated variant Chop2-315 of ChR2 from *Chlamydomonas reinhardtii*, accession number: AF461397; Nagel et al, 2003), ChR2(F219Y)-EYFP (Mager et al, 2018), and CatCh-EYFP (ChR2 L132C; Kleinlogel et al, 2011) were generated previously (MPI of Biophysics). The previously described ChR variants ChR2 T159C (Berndt et al, 2011) and ChR2 E123T/T159C (Berndt et al, 2011) were generated by site-directed mutagenesis using the primers shown in Appendix Table S3. Chronos F236Y, Chronos L149C, Chronos F236Y/L149C, and ChR2 F219Y/T159C were generated by site-directed mutagenesis employing the primers, which are also shown in Appendix Table S3. All ChR variants were C-terminally fused to EYFP. In addition, ChR2 T159C/F219Y, Chronos L149C and Chronos were fused to TS-EYFP-ES. The previously described targeting sequences of the inward rectifying potassium channel Kir2.1 (TS and ES) were employed for optimized plasma membrane expression (Gradinaru et al, 2010; Keppeler et al, 2018). ChR2(T159C/F219Y)-EYFP, Chronos(L149C)-TS-EYFP-ES and Chronos(L149C/F236Y)-TS-EYFP-ES were subcloned into a pAAV2 vector carrying the human synapsin promoter, the Woodchuck Hepatitis Virus

Posttranscriptional Regulatory Element (WPRE) and a polyadenylation site derived SV40.

## rAAV production and purification

Recombinant viral vector purification was carried out as previously reported, and its extensive description is available (Huet and Rankovic 2021). Briefly, triple transfection of HEK-293T cells was executed employing the pHelper plasmid (TaKaRa, USA), transplasmid with the AAV2/9 capsid and cis-plasmid with newly engineered blue-light sensitive ChR under the control of the human synapsin promoter (Challis et al, 2019). Cells were routinely screened for mycoplasma contamination. Viral particles were precipitated from culture supernatant with 40% polyethylene glycol 8000 (Acros Organics, Germany) in 500 mM NaCl and combined with cell pellets for processing. Pellets were suspended in 500 mM NaCl, 40 mM Tris, 2.5 mM $MgCl_2$, pH 8, and 100 Uml−1 of salt-activated nuclease (Arcticzymes, USA) at 37 °C for 90 min. Cleared lysates were purified using discontinuous iodixanol gradients (Optiprep, Axis Shield, Norway; 15, 25, 40, and 60%) at $350,000 \times g$ for 2.25 h. rAAV containing fractions were concentrated using Amicon filters (EMD, UFC910024) and formulated in sterile phosphate-buffered saline (PBS) supplemented with 0.001% Pluronic F-68 (Gibco, Germany). Viral vector titers were ascertained by the count of DNase I resistant vg using AAV titration kit (TaKaRa/Clontech) by qPCR (StepOne, Applied Biosystems) according the manufacturer's instructions or with adaptations to employment in crystal digital PCR (dPCR) system. Here, dilutions of isolated vector DNA were subjected to crystal dPCR using 5x Naica PCR reaction mix (Stilla, R10056) supplemented with primers provided in the AAV titration kit, 0,8 mg/µl Alexa 647 (Invitrogen, 11570266) and 1.5x Eva Green (Biotum, #31000-T). Formation of droplet crystals and PCR was performed in a Naica Geode (Stilla) using Naica Ruby or Sapphire Chips (Stilla) and ultimately examined in the Naica Prism3 reader (Stilla) equipped with Crystal Reader and Crystal Miner software (Stilla). The rAAVs titers and the quantification method used is indicated for each rAAV in the respective methods chapter. Purity of produced viral vectors was regularly verified by silver staining (Pierce, Germany) after gel electrophoresis (Novex™ 4 to 12% Tris–glycine, Thermo Fisher Scientific) according to the manufacturer's instructions (Appendix Fig. S13). Viral vector stocks were deposited at −80 °C until usage. In conclusion, final constructs used for the characterization in hippocampal neurons or optogene therapy in C57BL/6 mice included either CatCh, f-ChR2 TC, Chronos LC or f-Chronos, respectively, each under control of the human synapsin (neuronal targeting) combined with enhanced yellow fluorescent protein (EYFP), and with additional enhancement of trafficking signal (TS) and ER export signal (ES) for the Chronos variants. These constructs were packed into AAV2/9 capsids for injection into postnatal cochleae and subsequent optical stimulation of the auditory pathway. Work in gerbils focused on f-ChR2 TC produced and stored as above.

## In vitro experiments

### NG108-15 cell culture and transfection

NG108-15 cells (ATCC, HB-12377TM, Manassas, USA) were cultured at 37 °C and 5% $CO_2$ in DMEM (Sigma, St. Louis, USA)

supplemented with 10% fetal calf serum (Sigma, St. Louis, USA) and 1% penicillin/streptomycin (Sigma, St. Louis, USA). One day prior to transient transfections, the NG108-15 cells were seeded on 24-well plates. NG108-15 cells were transiently transfected with pcDNA3.1 (−) derivatives carrying specified ChRs and ChR mutants using Lipofectamine™ LTX Reagent (Invitrogen, Carlsbad, USA). Cells were tested for mycoplasma contamination using specific primers.

## Electrophysiological recordings on NG108-15 cells

Two to three days after transfection, NG108-15 cells were seeded on 12 mm diameter glass coverslips. Characterization of ChR mutants was performed using whole-cell patch-clamp under voltage clamp conditions using the Axopatch 200B amplifier (Axon Instruments, Union City, USA) and the DigiData 1320 A interface (Axon Instruments, Union City, USA). Patch pipettes with resistances of 2 to 5 MΩ were fabricated from thin-walled borosilicate glass on a horizontal puller (Model P-1000, Sutter Instruments, Novato, USA). The series resistance was <15 MΩ and the input resistance was >1 GΩ. The mean capacitance of the measured cells was $26.33 \pm 9.92$ pF ($n = 106$). If not stated differently the pipette solution contained 110 mM NaCl, 2 mM $MgCl_2$, 10 mM EGTA, 10 mM HEPES, pH 7.4 and the bath solution contained 140 mM NaCl, 2 mM $CaCl_2$, 2 mM $MgCl_2$, 10 mM HEPES, pH 7.4.

For determination and comparison of the off-kinetics and current densities, NG108-15 cells heterologously expressing the aforementioned ChRs were investigated at a membrane potential of -60 mV at RT (~24 °C), if not stated differently. Determination of closing kinetics was obtained from photocurrents elicited by a 3 ms light pulse and current densities from a 500-ms light pulse. Light pulses were delivered by focusing into a 400-µm optic fiber using diode-pumped solid-state lasers ($\lambda = 473$ nm) and a fast computer-controlled shutter (Uniblitz LS6ZM2, VincentAssociates, Rochester, USA). Closing kinetics at RT of f-Chronos were obtained from photocurrents elicited by a 7 ns light pulse ($\lambda = 500$ nm, $10^{20}$ photons/m$^2$) to avoid interference in the off-kinetics due to shutter opening/closing time (~700 µs) using the Opolette 355 tunable laser system (Opotek Inc, Carlsbad, USA). The current density ($J_{-60 mV}$) was determined by dividing the stationary current after a 500-ms light pulse with a saturating intensity of ~30 mW/mm$^2$ by the capacitance of the cell. The τoff value was determined by a monoexponential fit of the decaying photocurrent. In order to avoid an experimental bias, the NG108-15cells for the electrophysiological recordings were chosen independently of the brightness of their EYFP fluorescence. To investigate the dependence of the off-kinetics on the membrane potential, τoff values were determined at membrane potentials ranging from -100 to +60 mV in response to blue light illumination ($\lambda = 500$ nm, 7 ns, $10^{20}$ photons/m$^2$) using the Opolette 355 tunable laser system (Opotek Inc., Carlsbad, USA). The peak current recovery and the closing kinetics of selected ChRs were investigated at temperatures closer to physiological conditions (33 to 34 °C). The peak current amplitude was fully recovered after a period of 30 s in the dark (Appendix Fig. S2). Photocurrents were measured in response to 1 ms light-pulses by illumination of the focused field using a diode-pumped solid-state laser (LBX-488-40-CSB; 488 nm 40 mW, Oxxius Coherent Inc) connected through the back port of an

upright Olympus BX51WI microscope with a x40/0.8 LUMPLFLN40XW objective (Olympus) and a mCherry HC Filter Set mirror (F36-508 HC-Set mCherry, AHF - IDEX/Semrock). The Laser combiner box was connected to EPC10 USB HEKA patch clamp amplifier and shutter was controlled by voltage triggers through Patchmaster Next software (version 1.2, HEKA; Harvard Bioscience Inc.). Light power and wavelength were set by Oxxius Lasers software (version 2.4; Oxxius Coherent Inc). Maximal light power measured with the FieldMaxII-TOP laser power meter (Coherent) after the objective was ~40 mW/mm$^2$ for 488 nm. Temperature was controlled by CL-200A (Warner Instruments).

## Hippocampal neuron culture

Hippocampi were isolated from postnatal P1-P4 Wistar rats and treated with trypsin/EDTA (0.05%, Invitrogen, cat. No. 25300-062) for 20 min at 37 °C. The hippocampi were washed with dissection media (HBSS—10 mM Hepes) (Sigma-Aldrich, St. Louis, USA) and mechanically triturated with complete DMEM (DMEM + Glutamax supplemented with 10% fetal bovine serum and 1% P/S 100U/100 μg/mL). Cell suspension was filtered by a cell strainer (100 μm; Falcon) and live cells were counted by Trypan blue exclusion.

Approximately 50,000 cells were plated on 12 mm diameter glass cover slips coated with poly-D-lysine (Sigma) in 24-well plates in complete DMEM. After 3 to 4 h, the plating medium was replaced by Neuronal culture medium containing Neurobasal A (Gibco, Thermo Fisher Scientific, Waltham, USA) supplemented with 2% B-27 supplement, 1% Glutamax and 1% P/S (100 U/100 μg/mL) (Gibco, Thermo Fisher Scientific, Waltham, USA). Neuron cultures were fed every week by replacing 1/4 of the media with fresh Neuronal culture medium.

## Hippocampal neuron transduction

Briefly, $10^9$ to $5 \times 10^9$ genome copies/ml (GC/ml) of rAAV2/9 virus was added to each well 9 to 11 days after plating. Expression became visible 5 days post-transduction. The electrophysiological measurements were performed 13 to 21 days after transduction. No all-trans retinal was added to the culture medium or recording medium for any of the experiments described here.

## Electrophysiological recordings on hippocampal neurons

For whole-cell recordings in cultured hippocampal neurons, patch pipettes with resistances of 3 to 5 MΩ were filled with 129 mM potassium gluconate, 10 mM HEPES, 10 mM KCl, 4 mM MgATP and 0.3 mM Na$_3$GTP, titrated to pH 7.2. Extracellular solution contained 125 mM NaCl, 2 mM KCl, 2 mM CaCl$_2$, 1 mM MgCl$_2$, 30 mM glucose and 25 mM HEPES, titrated to pH 7.4. The series resistance was <20 MΩ, and the input resistance ranged from 0.1 to 1.14 GΩ. The mean capacitance of the measured cells was $22.22 \pm 9.98$ pF ($n = 62$). Neurons for the electrophysiological recordings were selected independently of their EYFP fluorescence. Recordings were conducted in the presence of the excitatory synaptic transmission blockers, 1,2,3,4-tetrahydro-6-nitro-2,3-dioxo-benzo[f]quinoxaline-7-sulfonamide (NBQX, 10 μM, Tocris) and D(−)-2-Amino-5-phosphonopentanoic acid (AP-5, 50 μM, Tocris). Electrophysiological signals were recorded using EPC10 USB HEKA amplifier (HEKA; Harvard Bioscience Inc.), filtered at 10 kHz by Patchmaster Next software (version 1.2, HEKA; Harvard Bioscience Inc.).

Light pulses were delivered by illumination of the focused field using a diode-pumped solid-state laser (LBX-488-40-CSB; 488 nm 40 mW, Oxxius Coherent Inc) connected through the back port of an upright Olympus BX51WI microscope with a x40/0.8 LUMPLFLN40XW objective (Olympus) and a mCherry HC Filter Set mirror (F36-508 HC-Set mCherry, AHF - IDEX/Semrock). The laser combiner box was connected to EPC10 USB HEKA patch clamp amplifier and the shutter was controlled by voltage triggers through Patchmaster Next software (version 1.2, HEKA; Harvard Bioscience Inc.). Light power and wavelength were set by Oxxius Lasers software (version 2.4; Oxxius Coherent Inc). Maximum light power measured with the FieldMaxII-TOP laser power meter (Coherent) after the objective was ~40 mW/mm$^2$ for 488 nm.

The current density ($J_{-70\,mV}$) was determined by dividing the stationary photocurrent in response to a 500 ms light pulse with a saturating intensity of ~20 to 40 mW/mm$^2$ and a wavelength of 488 nm by the capacitance of the cell.

In order to determine the lowest light intensity required to induce action potentials with a probability of 100%, 50 pulses ($\lambda = 488$ nm, pulse width = 1 ms, $\nu = 10$ Hz) of varying light intensities were applied. The spike probability was calculated by dividing the number of light-triggered spikes by the total number of light pulses.

## Ex vivo and in vivo experiments

### Animals

All animal experiments were carried out in accordance with relevant national and international guidelines (European Guideline for animal experiments 2010/63/EU, German Animal Welfare Act). The procedures have been approved by the responsible regional government office. Rodents were kept in a 12 h light/dark cycle with ad libitum access to food and water.

## Acute aVCN slice preparation and electrophysiology

Postnatally injected C57BL/6 wild-type mice of both sexes were studied after the onset of hearing (postnatal day 13) from day 20 to day 30 after birth.

### Slice preparation

Acute parasagittal slices (150 μm) from the anteroventral cochlear nucleus (aVCN) were obtained as described previously (Mendoza Schulz et al, 2014). Briefly, after sacrifice by decapitation, brains were dissected out and quickly immersed in ice-cold low Na$^+$ and low Ca$^{2+}$ cutting solution containing (in mM): 50 NaCl, 120 sucrose, 26 NaHCO$_3$, 1.25 NaH$_2$PO$_4$.H$_2$O, 2.5 KCl, 20 glucose, 0.2 CaCl$_2$, 6 MgCl$_2$, 0.7 Na L-ascorbate, 2 Na pyruvate, 3 myo-inositol, 3 Na L-lactate with pH adjusted to 7.4 and osmolarity of around 320 mOsm/l. After removal of the meninges from the ventral face of the brainstem, the two hemispheres were separated by a midsagittal cut, and the forebrain was removed at the pons-midbrain junction. The brain blocks containing the brain stem and cerebellum were then glued (cyanoacrylate glue; Loctite 401, Henkel) to the stage of a VT 1200 vibratome (Leica microsystems, Wetzlar, Germany) such that the medial side was glued on, the ventral side was facing the blade, and the lateral side was facing upwards, submerged in ice-cold cutting solution. For sectioning,

the blade was positioned at the height of the cerebellar flocculus, and sections were cut at a blade feed rate of 0.02 mm/s with an amplitude of 1 mm. Slices were incubated for 30 min in artificial cerebrospinal fluid (aCSF) maintained at 35 °C, and then kept at RT (22 to 24 °C) until recording. Composition of aCSF was identical to the cutting solution except (in mM): 125 NaCl, 13 glucose, 2 $CaCl_2$ and 1 $MgCl_2$. The pH of the solution was adjusted to 7.4 and the osmolarity was around 310 mOsm/l. All solutions were continuously aerated with carbogen (95% $O_2$, 5% $CO_2$).

### Electrophysiology

Patch-clamp recordings were made using an EPC10 USB Patch Clamp amplifier controlled by the Patchmaster Next software (version 1.2, HEKA; Harvard Bioscience Inc.). Sampling interval and filter settings were 25 µs and 7.3 kHz, respectively. Cells were visualized by differential interference contrast (DIC) microscopy through a x40/0.8 LUMPLFLN40XW water-immersion objective (Olympus) using an upright Olympus BX51WI microscope (Olympus). All experiments were conducted at a temperature of 33 to 35 °C, maintained by constant superfusion (flow rate 3 to 4 ml/min) of aCSF, heated by an inline solution heater (SC-20 with CL-200A controller; Warner Instruments, Hamden, CT, USA) and monitored by a thermistor placed between the inflow site and the slice, in the recording chamber.

Patch pipettes were pulled with a P-1000 micropipette puller (Sutter Instruments Co., Novato, CA, USA) from borosilicate glass capillaries with filament (GB150F, 0.86 × 1.50 × 80 mm; Science Products, 154 Hofheim, Germany). Open tip pipette resistance was 2 to 3.5 MΩ when filled with intracellular solution containing (in mM): 110 K-gluconate, 10 HEPES, 8 EGTA, 10 $Na_2$Phosphocreatine, 4 ATP-Mg, 0.3 GTP-Na, 4.5 $MgCl_2$, 10 NaCl, and 1 QX-314 (N-(2,6-dimethylphenyl carbamoylmethyl) triethylammonium chloride; Alomone Labs, Jerusalem, Israel) to block sodium channels, with a pH of 7.3 and an osmolarity of 320 mOsm/l. Additionally, 1 mM of fluorescent dye Alexa-568 (Invitrogen) was added to the recording pipette and cell structure was examined during experiments using a 532 nm laser with mCherry HC Filter Set mirror (F36-508 HC-Set mCherry, AHF - IDEX/Semrock). Cells were voltage-clamped at a holding potential of −70 mV, after correction for a liquid junction potential of 12 mV. Mean series resistance was around 8 MΩ. For the main set of recordings, bath solution (aCSF) was supplemented with 2 µM Strychnine hydrochloride, a glycine receptor antagonist.

EYFP fluorescence signal was checked after recordings using an EGFP HC Filter Set mirror (F36-528 HC-Set EGFP, AHF - IDEX/Semrock) with an Andor zyla sCMOS camera and Andor SOLIS imaging software (version 4.32.30065.0, Andor).

We differentiated between bushy and stellate cells based on their distinct biophysical properties (Cao and Oertel 2010). While the injection of depolarizing current in stellate cells triggers tonic firing (Appendix Fig. S6A–C), bushy cells show phasic firing (Appendix Fig. S6D–F). As our recording pipettes contained the $Na_V$ channel blocker "QX-314" to lower noise and improve voltage clamp, we checked the firing pattern right after the break, prior to full $Na_V$ block (Appendix Fig. S6$E_1$,F vs. $E_2$). Correct cell identification was further completed by criteria of morphology introducing an intracellular dye (Alexa Fluor 568, Appendix Fig. S7A,B; Wu and Oertel 1984), immunohistochemistry (Appendix Fig. S7C), and monitoring the decay kinetics of postsynaptic currents (Appendix Fig. S7D–J; Isaacson and Walmsley 1995; Lu et al,

2007; Chanda and Xu-Friedman 2010). All cells included in this study were classified as bushy cells due to the fast decay kinetics of their spontaneous EPSC and their oeEPSCs (Appendix Fig. S7D–F). The limit for bushy cell-like decay kinetics was set at ≤0.5 ms based on previous studies (Appendix Fig. S7E; Isaacson and Walmsley 1995; Gardner et al, 1999; Cao and Oertel 2010; Chanda and Xu-Friedman 2010; Butola et al, 2021). For ten out of fifteen cells that we classified as bushy cells, an additional slow component in oeEPSC decay was observed. For these cells, a biexponential fit was applied, to isolate the dominating fast component. Note that three cells were excluded from this study due to ambiguous kinetics and morphology.

Synaptic delay was calculated as the time between the start of stimulus (voltage output of the amplifier as dictated by the experiment protocol) and the time when the respective oeEPSC response reached 10% of its peak amplitude.

## Immunostaining and imaging of aVCN slices

After electrophysiological recordings, aVCN slices were fixed with 4% paraformaldehyde in phosphate-buffered saline for 45 min and kept at 4 °C in PBS until immunostaining was performed. After carefully placing the aVCN slices on glass slides, we incubated in Goat Serum Dilution Buffer (GSDB; 16% normal goat serum, 450 mM NaCl, 0. 3% Triton X-100, 20 mM phosphate buffer, pH 7.4) for 1 h, followed by incubation in primary antibodies diluted in GSDB overnight in a wet chamber at 4 °C. After washing 2 × 10 min with wash buffer (450 mM NaCl, 0.3% Triton X-100, 20 mM phosphate buffer) and 2 × 10 min with PBS, the slices were incubated with secondary antibodies diluted in GSDB, for 2 h, in a light-protected wet chamber at RT. The slices were then washed 2 × 10 min with wash buffer and 2 × 10 min with PBS. Finally, we stuck an imaging spacer Grace Bio-Labs SecureSeal ™ (GBL654008-100EA, Sigma-Aldrich) around the slice to avoid squeezing the tissue with the coverslip, and mounted with a drop of fluorescence mounting medium based on Mowiol 4–88 (Carl Roth, Karlsruhe, Germany) covered with a thin glass coverslip.

The following primary antibodies were used: chicken anti-GFP (catalog number: ab13970, dilution 1:500) (Abcam, Cambridge, United Kingdom), guinea pig anti-VGLUT1 (1:1000; Cat Nr 135304, SySy), and rabbit anti-Homer1 (1:500; Cat Nr. 160002, SySy). The following secondary AlexaFluor-labeled antibodies were used: goat anti-chicken 488 IgG (H + L), catalog number: A-11039, dilution 1:200 (Invitrogen); goat anti-guinea pig 568 IgG (H + L), catalog number A11075, dilution 1:200 (Invitrogen); goat anti rabbit 647 IgG (H + L), catalog number A21244, dilution 1:200 (Invitrogen); goat-anti guinea pig 647 IgG (H + L), catalog number A21450 dilution 1:200 (Invitrogen).

Confocal images were collected using a laser-scanning SP8 confocal microscope (Leica, Hamburg, Germany) equipped with 488 nm (Ar), 561 DPSS, and 633 nm (He-Ne) lasers and 40x/1.3 NA oil-immersion objective. Confocal z-stacks of 1 µm were summed to the maximal.

## Optogenetic stimulation of the auditory pathway

### Postnatal AAV injection into the cochlea

The same injection approach was performed for all animals used, which were further subjected to either: in vivo auditory brainstem recordings, in vivo inferior colliculus recordings or ex vivo acute

slice electrophysiology (Keppeler et al, 2018; Mager et al, 2018; Bali et al, 2021; Mittring et al, 2023; Zerche et al, 2023). AAV-round-window-injections were only conducted into the left cochlea of C57BL/6 wild-type mice of both sex at postnatal day 6 or wild-type Mongolian gerbils at postnatal day 8. The contralateral cochlea was used as a non-injected control. To summarize, mouse or gerbil pups were randomly selected for virus injections. Throughout the injection procedure, all animals were frequently monitored with regard to general isoflurane anesthesia (5% for anesthesia induction, 1 to 3% for maintenance), the absence of the hind-limb withdrawal reflex, the breathing rate as well as maintenance of physiological body temperature was controlled and application of eye creme was performed. Adjustments were made accordingly. Analgesia was provided by local xylocaine application as well as by subcutaneous buprenorphine (0.1 mg/kg) and carprofen (5 mg/kg) dosages. The tissue was carefully spread, and the cochleae were gently punctured using a borosilicate capillary pipette, which was kept in place to inject ~1 µl of the viral constructs of the following titers into the test subjects' cochleae:

Mice:

- AAV2/9_hSyn_ChR2(T159C/F219Y)-EYFP_WPRE_SV40pA at 1,35E + 13 GC/ml and at 3,77E + 13 GC/ml (qPCR)
- AAV2/9_hSyn_Chronos(L149C)-TS-EYFP-ES_WPRE_SV40pA at 7,4E + 13 GC/ml (qPCR)
- AAV2/9_hSyn_Chronos(F236Y)-TS-EYFP-ES_WPRE_SV40pA at 9,8E + 12 GC/ml (qPCR)

Gerbils:

- PHP.S_hSyn-hChR2(T159C/F219Y)-EYFP-WPRE_SV40pA at 1,51E + 12 GC/ml (cdPCR)

After injection of the constructs, the tissue was repositioned in the area of the procedure. The surgical lesion was then sutured. Carprofen (5 mg/kg) was administered for analgesia up until one day postsurgically.

## Laser-based optical stimulation of the auditory pathway in vivo

The injected left-sided cochleae were exposed surgically by performing a retroauricular incision followed by a bullostomy to inspect the round window and puncture its membrane. For optical stimulations an optical fiber (200 µm in diameter, 0.39 NA; Thorlabs) coupled to a 473 nm laser in mice (MLLFN-473-100; Changchun New Industry Optoelectronics) or 488 nm laser (LBX-488-100, Oxxius) in gerbils was positioned in the round window. Laser power was calibrated prior to each experiment using laser power meters (Solo-2; Gentec-EO and Thorlabs PM103USB, S140C).

## Optically evoked auditory brainstem responses

For electrophysiological experiments, data sets were obtained from 34 adult C57Bl/6 wild-type mice of either sex (12 male and 22 female) and 4 Mongolian gerbils (four male) that received virus injections (see: Postnatal AAV injection into the cochlea) prior to optical stimulations and measurements in vivo. Surgeries and measurements were then performed 76.8 ± 14.6 days after injections for mice and 79.5 ± 21.7 days after injections for gerbils with strict and frequent monitoring of the animals in isoflurane anesthesia (5% for anesthesia induction, 1 to 3% for maintenance): by controlling hind-limb withdrawal reflex, breathing, as well as maintaining physiological body temperature and applying eye cream. Adjustments were made accordingly. Analgesia was provided by subcutaneous injections of buprenorphine (0.1 mg/kg body weight) and carprofen (5 mg/kg body weight) for mice or meloxicam (5 mg/kg body weight) for gerbils. Stimuli were created and acquired using custom-written software (MATLAB, MathWorks) employing National Instruments data acquisition cards (NI PCI-6229; National Instruments) and a custom-built laser-controller. All measurements were conducted in a soundproof chamber (Industrial Acoustics). Acoustically evoked auditory brainstem responses (aABRs) as well as optically evoked ABRs (oABRs) were measured by placing subcutaneous needle electrodes behind the pinna, on the vertex, and tail-side of the anesthetized mice or gerbils. Prior to the oABR recordings, aABRs were measured using near-field acoustic stimulation, centered 30 cm in front of the animal's head with a loudspeaker (Scanspeak Ultrasound or Vifa; Avisoft Bioacoustics). Acoustic calibrations were performed with a 0.25-inch microphone (4039; Brüel & Kjaer or GRAS 46BF-1) and a corresponding amplifier. For all ABRs, the difference in far-field potential between the subcutaneously inserted needle electrodes at the vertex and mastoid was amplified using a custom-designed amplifier. The sampling rate was at 50 kHz for 20 ms, and the signal was filtered (300 to 3000 Hz), and averaged across 1000 stimulus presentations. The ABR thresholds were then determined as the lowest light or sound intensity for which an ABR waveform was reliably visible.

The time delay between the stimulus onset and the peak of the wave of interest was defined as the latency. Further, amplitudes were determined as the difference in response strength between each corresponding positive peak (P) and the negative (N).

## Recording of inferior colliculus responses

In three Mongolian gerbils, we recorded multi-unit activity from the tonotopically organized central nucleus of the inferior colliculus (IC), as previously described (Dieter et al, 2019). After head fixation and stereotactic alignment, a craniotomy was performed over the right hemisphere, ~2 mm lateral and ~0.5 mm caudal of lambda, contralateral to the injected ear. A 32-channel linear electrode (177 µm² electrode surface, 50 µm electrode spacing, Neuronexus) was placed over the exposed visual cortex and slowly advanced ~3300 µm into the brain using a micromanipulator (SM-10 compact, Luigs & Neumann). The correct placement along the tonotopic axis of the IC was probed using 100 ms pure tones between 0.5 and 32 kHz presented at 0 to 80 dB by a loudspeaker (Vifa, Avisoft Bioacoustic) (Fig. 6b). Acoustic calibrations were performed using a 0.25-inch microphone (GRAS 46BF-1, amplified by GRAS 12AQ). Accordingly, the recording site was adjusted by further advancing or retracting the silicon probe. Activity of multi-neuronal clusters was amplified, filtered (0.1/50 to 8000 Hz) and recorded at a sampling rate of 32 kHz by a Digital Lynx 4S recording system (Neuralynx). Stimulation was provided by a laser fiber coupled to a 488 nm laser (LBX-488-100, Oxxius) or LED-based oCIs.

## Stimulation with LED-based multichannel optical cochlear implants

Multichannel oCIs were assembled from up to 10 LED chips (C460TR2227-S2100, Cree, 220 × 270 µm, 300–400 µm pitch) emitting at 460 nm, integrated on polyimide substrates and encapsulated in silicone by Eric Klein and Patrick Ruther as described before (Keppeler et al, 2020). The cochlea was accessed over the bullostomy used for laser-based stimulation, and oCIs were carefully implanted through a base turn cochleostomy or the round window. Single LEDs were operated to present 100 ms pulse trains assembled from 1 ms pulses at rates between 10 and 500 Hz and intensities up to 6 mW using custom-built stimulation hardware.

All oCI-channels were calibrated separately using an optical power meter (Thorlabs PM103USB, S140C).

## Analysis of IC recordings

Recordings from the inferior colliculus were analyzed using custom-written MATLAB scripts previously described (Dieter et al, 2019; Keppeler et al, 2020; Michael et al, 2023). Raw data traces were band-pass filtered between 0.6 and 6 kHz (fourth-order Butterworth filter). For each time-point, a global mean was calculated over all electrodes and subtracted from each electrode's filtered trace to minimize the influence of any stimulation artefacts (McInturff et al, 2022). Afterwards, three times the mean absolute deviation (estimated as mean/0.675 for data 100 to 2 ms before stimulation) was defined as the threshold for detection of firing activity, threshold crossings were collected as spike times. To quantify response strengths of multiunits, i.e., response strengths from individual recording electrodes, spike rates were calculated in time windows relative to the beginning of the stimulus (2 to 25 ms for single LED pulses or 4 to 125 ms for 100 ms pure tones). Spike rates for 30 repetitions of increasing stimulus intensities were used to calculate a cumulative d'-value (Macmillan and Creelman 2004; Middlebrooks and Snyder 2007). These spike rates were first used to calculate empirical receiver operating curves (ROC) and the corresponding areas under the curve (AUC). D'-values were derived by multiplying the AUC transformed by the inverse cumulative normal distribution function by the square root of two and then summed over increasing stimulus conditions to reach a cumulative d'-value.

The first presented stimulus intensity to reach a cumulative d'-value above 1 was defined as the threshold. For acoustic stimulation, the best electrode was defined as the recording electrode (or the mean of electrodes) reaching the threshold at the lowest stimulus intensity. We estimated tonotopic slopes by linearly fitting the best electrodes while iteratively removing outliers, defined as points farther than ±1.5 standard deviations from the fitted line, in two rounds.

Temporal response properties were characterized in responsive units, i.e., units reaching a d' of 1 compared to a no stimulus condition, when stimulated with a single LED pulse at the intensity used for pulse trains. Spike rates for 100 ms pulse trains were derived by dividing the number of spikes in a time window from 0 to 110 ms relative to the stimulus start divided by the duration of this time window. As a metric for synchronization of spike timing

to the stimulus we derived the vector strength as

$$VS = \frac{\sqrt{\left(\left(\sum_{i=1}^{n}\cos(\theta_i)\right)^2 + \left(\sum_{i=1}^{n}\sin(\theta_i)\right)^2\right)}}{n}$$

with $\theta$ being the phase of a detected spike in each stimulus cycle (i.e. time from the beginning of a pulse to the subsequent pulse; Goldberg and Brown 1969). Here, we only used spikes occurring in a time window from 50 to 110 ms for the calculation of vector strength to avoid overestimation due to the onset response. The Rayleigh test was applied to assess the significance of the vector strength. Values with $L < 13.8$ (corresponding to $p > 0.001$) were considered non-significant and set to zero.

## Immunolabelling and confocal imaging of cochlear cryosections

Following the surgeries and the optogenetic measurements, cochleae from both sides were extracted from the temporal bone and prepared as described previously (Keppeler et al, 2018; Mager et al, 2018; Bali et al, 2021; Zerche et al, 2023). Briefly, cochleae were fixed for 45 min in 4% formaldehyde and subsequently decalcified for 5 to 10 days in 0.12 M ethylenediaminetetraacetic acid. For mid-modiolar cryosections (16-µm thick), decalcified cochleae were dehydrated in 25% sucrose of phosphate-buffered saline solution for 24 h, cryopreserved and then sectioned. Selected sections were immunolabelled using a goat-staining buffer (16% normal goat serum, 450 mM NaCl, 0.6% Triton X-100, 20 mM phosphate buffer, pH 7.4) and the following antibodies: guinea pig anti-parvalbumin (1:300, 195004 Synaptic Systems), chicken anti-GFP (1:500, ab13970 Abcam), goat anti-chicken 488 IgG (1:200, A-11039 Thermo Fisher Scientific), and goat anti-guinea pig 568 IgG (1:200, A-1107 Thermo Fisher Scientific). Primary antibodies were incubated at 4 °C ON and secondary antibodies for one hour at RT. Finally, the labeled sections were mounted in Mowiol 4–88 (Carl Roth). Cochlear slices were then imaged with a confocal microscope SP8 (Leica) mounted with a 20 x and a 40 x objective using immersion oil at a Z-step size of 2 and 1 µm, respectively. Across all cochlear turns (apex, middle and base), images were taken covering the whole stack, focusing on the modiolus.

## Quantification of SGNs in immunolabelled cochlear cryosections

For image analysis and SGN quantification, a custom-written MATLAB script modified from Huet et al, 2021 allowing computer-assisted analysis of cochlear cryosections as previously employed in Alekseev et al, 2025 was used. In short, the locations for the SGN somata were manually detected in their respective modiolar area for all images using the parvalbumin channel as a reference. Next, individual masks for each somata were automatically segmented from every Z-stack using Otsu's threshold method if the criteria of size (area and diameter) as well as of circularity were met. In case the automatic method did not work, masks were determined manually. Subsequently, a Gaussian mixture model with typically 1 to 3 components was fitted to the distribution of the median somatic SGN brightness measured in the GFP channel. Hence, a

fluorescence threshold was set as the mean plus twice the standard deviation of the Gaussian distribution with the lowest mean. Only if exceeding the threshold, SGN somata were considered as optogenetically modified. In summary, this enabled the assessment of the total SGN density as well as the density of GFP-positive SGNs.

## Quantification of immunofluorescence distribution in immunolabelled cochlear cryosections

To assess the subcellular expression profiles of channelrhodopsins in spiral ganglion neurons (SGNs), we developed a custom Python program to plot fluorescence line profiles, allowing a manual verification process. First, we selected the central plane of the Z-stack and identified the centroids of the cells in the cytoplasmic calretinin channel using the mean-shift clustering algorithm with a bandwidth of 6.5 μm. These centroids were used to generate 2D square crops of the images, each containing one complete SGN.

Cell membrane masks were automatically generated based on the cytoplasmic calretinin channel fluorescence using a fixed threshold. Line profiles were then automatically positioned perpendicular to these masks, avoiding intersection with surrounding cells. Manual verification of the SGN masks and line profile positions was conducted using a custom graphical interface developed with Napari. We manually checked the Z-plane for a noticeable fluorescence drop indicating the position of the nucleus; cells lacking a discernible nucleus were excluded to prevent bias from line profiles in the nuclear region or parallel to the cell surface. Any line profiles intersecting visually identifiable nuclei were excluded from further analysis. Only cells and line profiles that passed manual verification were included in the further analysis. Cells with fewer than three confirmed line profiles were automatically excluded. Fluorescence data from the GFP channel (488 nm) were recorded along each line profile and averaged across all lines for each cell, creating an expression line profile for each cell. These expression line profiles from all verified cells were then averaged, and the standard deviation was calculated. The ratio of membrane to intracellular fluorescence was calculated for each selected cell by dividing the mean fluorescence measured between 0.4 and 0.6 μm by the mean fluorescence measured between 1.4 and 1.6 μm.

## Immunohistochemical quantification of cochlear paraffin sections

Left and right cochleae of animals injected with f-ChR2 TC ($n = 3$ each) and Chronos LC ($n = 2$ each) were fixed in 4% formaldehyde for 45 min and stored in PBS at 4 °C until processing. Here, decalcification was performed using the EDTA (ethylenediamine-tetraacetic acid)-based OSTEOSOFT® mild decalcifier solution for histology (Merck KGaA, Darmstadt, Germany) for 24 h. Decalcified samples were then embedded in paraffin and serial sectioned at ~4 μm. Every fifth section was stained with hematoxylin and eosin (HE) for histological analyses. Sections in between were mounted on superfrost slides and used for immunohistochemical staining with an anti-CD3 antibody labeling T-cells (A0452, Dako Deutschland GmbH, Hamburg, Germany, polyclonal rabbit anti-human CD3, 1:50) and an anti-Iba-1 antibody to visualize the macrophage/

microglia-specific calcium-binding protein Iba-1 (GeneTex, Biozol Diagnostica Vertrieb GmbH, Eching, Germany, rabbit polyclonal, 1:100). Immunohistochemical staining was performed with an automated immunostaining system (Discovery XT, Roche Diagnostics GmbH, Mannheim, Germany) using the SABC (Streptavidin-Biotin-Complex) method, DAB (diaminobenzidine tetrahydrochloride) for signal detection (DAB Map Kit, Roche Diagnostics GmbH, Mannheim, Germany) and hematoxylin for counterstaining. Mouse spleen and brain tissue served as positive controls. Pure antibody diluent was applied to the sections of the negative control.

HE sections were examined with a light microscope by a veterinary pathologist for pathological changes. As previously described (Gibson-Corley et al, 2013; Bali et al, 2022), five semiquantitative scores (none 0, minimal 1, mild 2, moderate 3, severe 4, see also Appendix Fig. S5B) were applied for estimation of the extent of neuronal density and interstitial vacuolation in apical, middle and basal turns of spiral ganglions from injected and contralateral cochleae. Immunohistochemical stains for CD3 and Iba-1 were also evaluated light microscopically. The presence of immune cells was descriptively documented.

## Data analysis

The data were processed and analyzed using Matlab software, also custom-made (The MathWorks, Inc., Natick, MA, USA), Microsoft Office 365 (Microsoft Corp., Redmond, WA, USA), Igor Pro 6 (Wavemetrics, Portland, OR, USA), Origin 9.0 (OriginLab, Inc., Northampton, MA, USA), GraphPad Prism 10.2.1. (GraphPadSoftware Inc., La Jolla, CA, USA), and ImageJ (NIH, Bethesda, MD, USA). Averages were expressed as mean ± SD, as specified.

For statistical comparison between two groups, data sets were tested for normal distribution (the D'Agostino & Pearson omnibus normality test or the Shapiro–Wilk test) and equality of variances ($F$-test) followed by two-tailed unpaired Student's $t$-test, or the unpaired two-tailed Mann–Whitney $U$-test when data were not normally distributed, and Welch correction was used when variance was unequal between samples. For evaluation of multiple groups, statistical significance was calculated by using ordinary one-way ANOVA test followed by Bonferroni's or Tukey's test for normally distributed data (equality of variances tested with the Brown–Forsythe test) or one-way Kruskal–Wallis test followed by Dunn's test for non-normally distributed data.

## Data availability

The code used for analysis is available via Zenodo at https://doi.org/10.5281/zenodo.15210800 and https://doi.org/10.5281/zenodo.17458418.

The source data of this paper are collected in the following database record: biostudies:S-SCDT-10_1038-S44321-025-00350-z.

## Peer review information

## The paper explained

### Problem

Optogenetics enables the control of excitable cell networks by light, thereby advancing both basic science research and future medical applications such as hearing restoration. Disabling hearing loss (HL) is common, and hearing can be partially restored to patients with severe HL or deafness by cochlear implants (CI). However, CI users face limitations in daily life, such as difficulties in speech comprehension in noisy environments. This is due to the broad spread of electric current from each electrode contact that then activates large sets of spiral ganglion neurons (SGNs) along the tonotopically organized cochlea, which limits frequency resolution. Future optical cochlear implants aim to overcome the limited frequency resolution by using optogenetic SGN stimulation, as light can be better confined in space. Ultrafast-closing channelrhodopsins (ChRs) that provide high temporal fidelity SGN stimulation, however, require high light intensities, which poses challenges for their use.

### Results

Here, we balance fast-closing kinetics and high neural light sensitivity by engineering light-sensitive variants of blue-light-shifted ChRs: Chronos (Chronos LC) and ChR2 (f-ChR2 TC) with fast closing kinetics. We characterize their utility for encoding time and intensity information in mouse SGNs in brainstem slices and in vivo. Comparative investigation of optogenetic stimulation in f-Chronos, Chronos LC and f-ChR2 TC identified f-ChR2 TC as a promising candidate ChR comparable to the previously described red-light activated f-Chrimson. f-ChR2 TC enabled SGN photostimulation at near-physiological firing rates with good temporal precision up to 300 Hz of stimulation and at a relatively low energy budget.

### Impact

Our work generated and characterized a novel ChR variant with well-balanced properties, combining fast closing kinetics and low activation energy. f-ChR2 TC is a novel optogenetic actuator with potential for fundamental research and preclinical studies on optogenetic therapies.

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

## Acknowledgements

The authors gratefully acknowledge the excellent technical support of Sandra Gerke, Sina Langer, Gerhard Hoch, Ludwig Ehrenreich and Dr. Christian Gossler, as well as excellent surgical training from PD Dr. Christian Wrobel and Dr. Burak Bali. Further, we thank PD Dr. Christian Wrobel and Dr. Daniel Keppeler for providing in vivo data for f-Chrimson and Chronos. The authors especially thank Christiane Senger-Freitag for expert help with cloning of the ChR constructs and postnatal injections of the animals, as well as Daniela Gerke for virus production and histological preparation of auditory samples, and further Patricia Räke-Kügler for excellent administrative support. Further, we thank Larissa Hummel and Nadine Schminke for the immunohistochemical preparation of cochlear paraffin sections and Esther Grewe for her support with the customized analysis of immunohistochemical data. Ultimately, the authors thank the Institute for Auditory Neuroscience for valuable scientific input and teamwork. The work was funded by the German Research Foundation through the Cluster of Excellence (EXC2067) Multiscale Bioimaging (TM, TMa) and the collaborative research centers 1286 (project B05 TM) and 1690 (projects B01 TM, TMa and A04 KK, TM) as well as by the Else Kröner Fresenius Foundation through the Else Kröner Fresenius Center for Optogenetic Therapies (KK, TM, TMa). Work in the Moser lab was further supported by the Ernst Jung Prize for Medicine and by Fondation Pour l'Audition (FPA RD-2020-10) and the Volkswagen-Stiftung from the "Niedersächsisches Vorab" (ZN3898 and ZN4000). LR was supported as a clinician scientist by EXC2067 and the Else Kröner Fresenius Center for Optogenetic Therapies. AG was supported by the Alexander von Humboldt Foundation. AV was supported by HORIZON TMA MSCA Postdoctoral Fellowship (OPTOCODE, grant 101107675) and NA by a scholarship of the Göttingen Promotionskolleg für Medizinstudierende, funded by the Jacob-Henle-Programm/Else-Kröner-Fresenius-Stiftung (Promotionskolleg für Epigenomik und Genomdynamik 2021_EKPK.04) and by the German Academic Scholarship Foundation.

## Author contributions

**Lennart Roos**: Data curation; Formal analysis; Investigation; Writing—original draft; Writing—review and editing. **Aida Garrido-Charles**: Data curation; Formal analysis; Investigation; Writing—original draft; Writing—review and editing. **Niels Albrecht**: Formal analysis; Investigation. **Anna Vavakou**: Formal analysis; Investigation. **Alexey Alekseev**: Formal analysis. **Martina Bleyer**: Formal analysis; Investigation. **Anupriya Thirumalai**: Formal analysis. **Artur Mittring**: Investigation. **Theocharis Alvanos**: Investigation. **Antoine T Huet**: Formal analysis; Investigation. **Ernst Bamberg**: Resources; Writing—review and editing. **Kathrin Kusch**: Investigation. **Bettina J Wolf**: Supervision; Investigation. **Tobias Moser**: Conceptualization; Supervision; Funding acquisition; Writing—original draft; Writing—review and editing. **Thomas Mager**: Conceptualization; Data curation; Formal analysis; Supervision; Investigation; Writing—original draft; Writing—review and editing.

Source data underlying figure panels in this paper may have individual authorship assigned. Where available, figure panel/source data authorship is listed in the following database record: biostudies:S-SCDT-10_1038-S44321-025-00350-z.

## Funding

## Disclosure and competing interests statement

TM is the co-founder of the OptoGenTech Company. The remaining authors declare no competing interests.

# Expanded View Figures

**Figure EV1. Photocurrent measurements of blue light-activated ChRs at physiological temperature.**

(A) Exemplary peak-normalized photocurrents from whole-cell patch-clamp measurements at ~34 °C in NG108-15 cells. ChRs were activated by 50 light pulses of 1 ms at 488 nm (~40 mW/mm²) at different frequencies (5, 50, 125, and 500 Hz). The lower panels are magnifications of the 500 Hz traces showing photocurrent fluctuations at the stationary state. (B) Dependence of the stationary/peak ratio on light pulse frequency. Error bars depict SD. (C) Quantification of photocurrent fluctuations normalized to the stationary current amplitude. In panels (B, C): CatCh: $n = 5$, f-Chronos: $n = 3$, Chronos LC: $n = 3$, f-ChR2 TC: $n = 4$. Error bars depict SD. Statistical comparisons can be found in Table EV3. (D) Exemplary peak normalized photocurrents measured in NG cells expressing f-Chronos, Chronos, f-ChR2 TC, or CatCh showing the relation between photocurrent decay kinetics and the transferred charge (area under the curve shown in yellow).

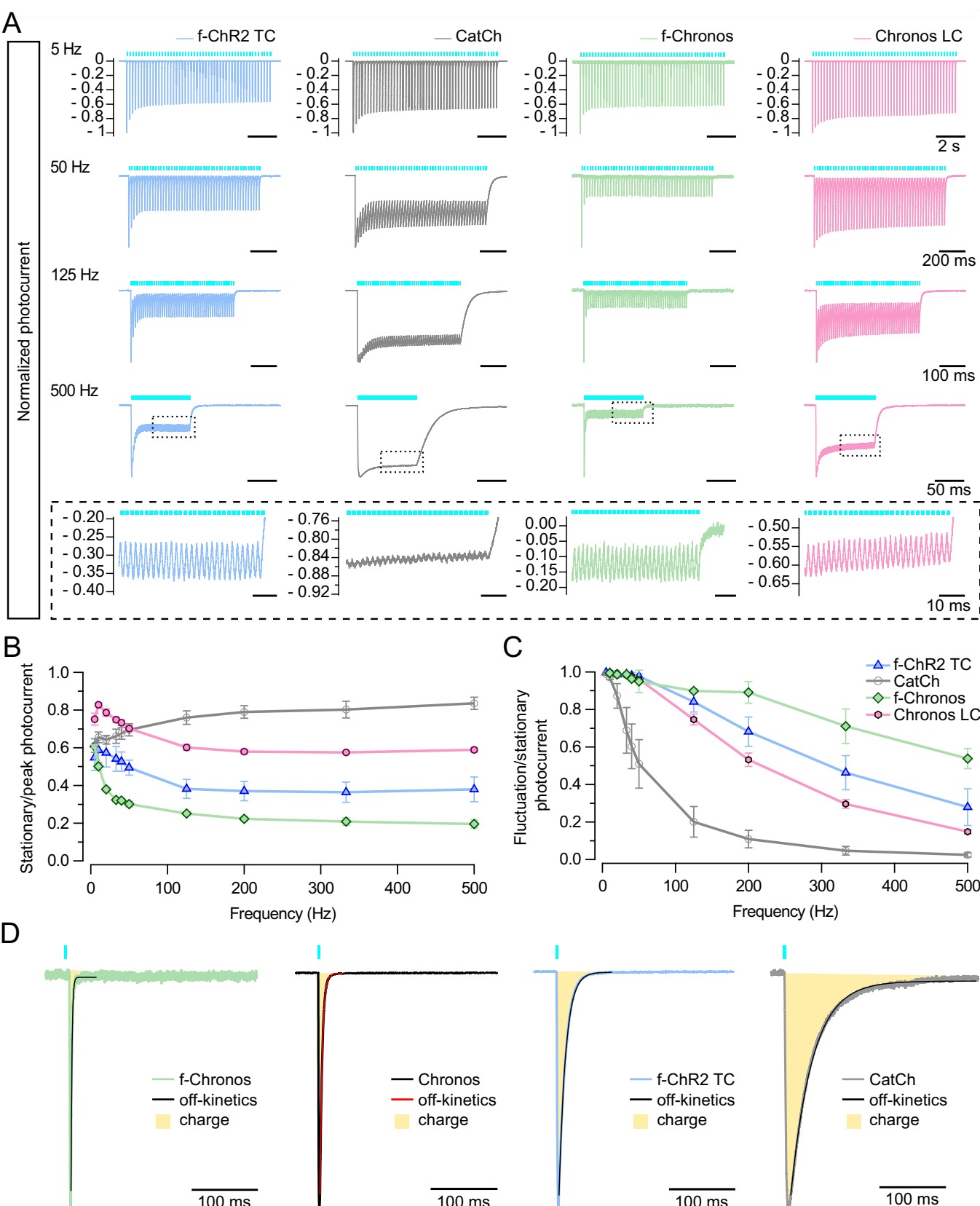

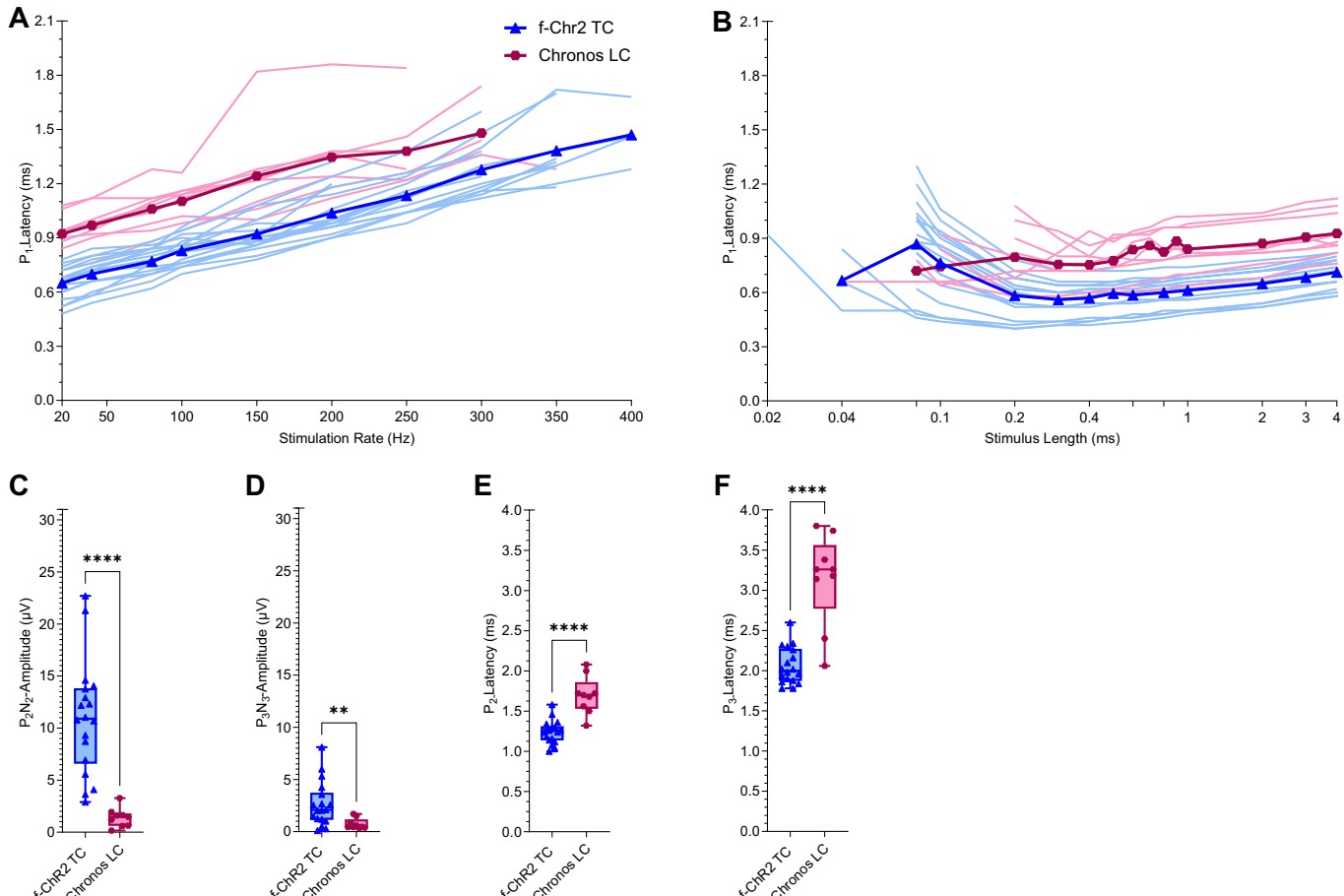

**Figure EV2.  Analysis of auditory pathway activity evoked by photostimulation of SGNs expressing optimized blue-light-sensitive ChRs.**

(A) P1 latency of oABRs at varying repetition rate using 1 ms pulses at ~38 to 45.6 mW (bold: mean; faint: all measurements), $n = 18$ mice for f-Chr2 TC, $n = 9$ mice for Chronos LC. (B) P1 latency of oABRs for varying pulse durations using ~38 to 45.6 mW pulses at 10 Hz (bold: mean; faint: all measurements) for $n = 17$ mice for f-Chr2 TC, $n = 9$ mice for Chronos LC. (C–F) P2-N2, P3-N3 amplitudes and P2, P3 latencies of oABRs for $n = 18$ mice for f-Chr2 TC, $n = 9$ mice for Chronos LC depicting activation of the auditory pathway using ~38 to 45.6 mW, 1 ms pulses at 10 Hz. Data were analyzed as mean ± SD. Center lines represent median values. Boxes show the 25th and 75th percentile and error bars depict minima and maxima. ****$p = 8.53e{-}7$ (C), $p = 1.09e{-}5$ (E), $p = 2.62e{-}5$ (F); **$p = 0.0075$ by two-tailed Mann–Whitney test.

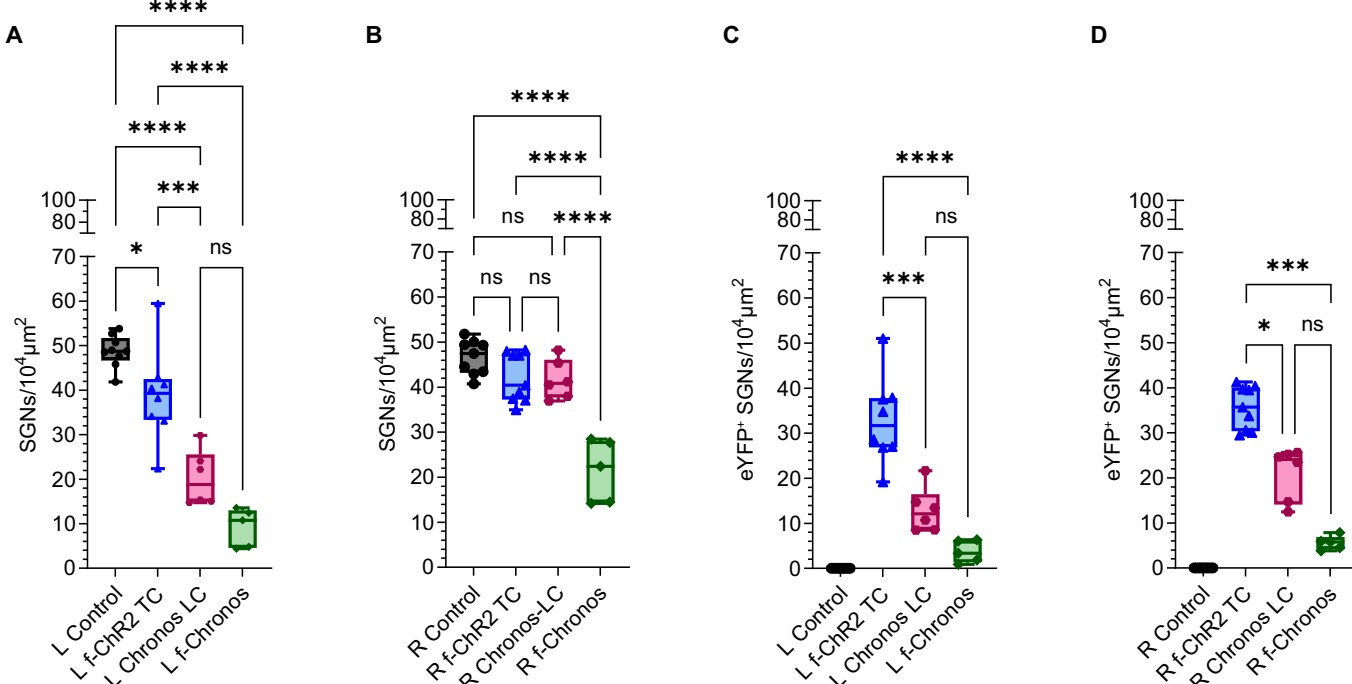

**Figure EV3. Immunohistochemical quantification of SGNs expressing optimized blue-light sensitive ChRs.**

(A–D) Box plots show statistics for the SGN density of left (injected, **A**) and right (non-injected, **B**) cochleae across all cochlear turns, as well as for the density of ChR-expressing (EYFP-positive) cells for the left (**C**) and the right (**D**) side. Quantification includes f-ChR2 TC (blue; $n = 8$ for the left and $n = 9$ for the right cochleae), Chronos LC (violet; $n = 6$ for left and right cochlea), f-Chronos (green; $n = 5$ for left and right cochlea), and non-treated wild-type cochleae (black; $n = 9$ cochlea for both sides each). Center lines represent median values. Boxes show the 25th and 75th percentile and error bars depict minima and maxima. ****$p = 2.68e-7$ (**A**: L Control vs L Chronos LC), $p = 1.74e-9$ (**A**: L Control vs. L f-Chronos), $p = 5.00e-7$ (**A**: f-ChR2 TC vs. L f-Chronos), $p = 1.68e-8$ (**B**: R Control vs. R f-Chronos), $p = 6.14e-7$ (**B**: R f-ChR2 TC vs. R f-Chronos), $p = 3.53e-6$ (**B**: R Chronos LC vs. R f-Chronos), $p = 6.06e-6$ (**C**: L f-ChR2 TC vs. L f-Chronos); ***$p = 0.0002$ (**A**), $p = 0.0003$ (**C**), $p = 0.0002$ (**D**); *$p = 0.0458$ (**A**), $p = 0.0485$ (**D**) by ordinary one-way ANOVA corrected with Bonferroni's (**A–C**) and Kruskal–Wallis test corrected for multiple comparison with Dunn's (**D**).

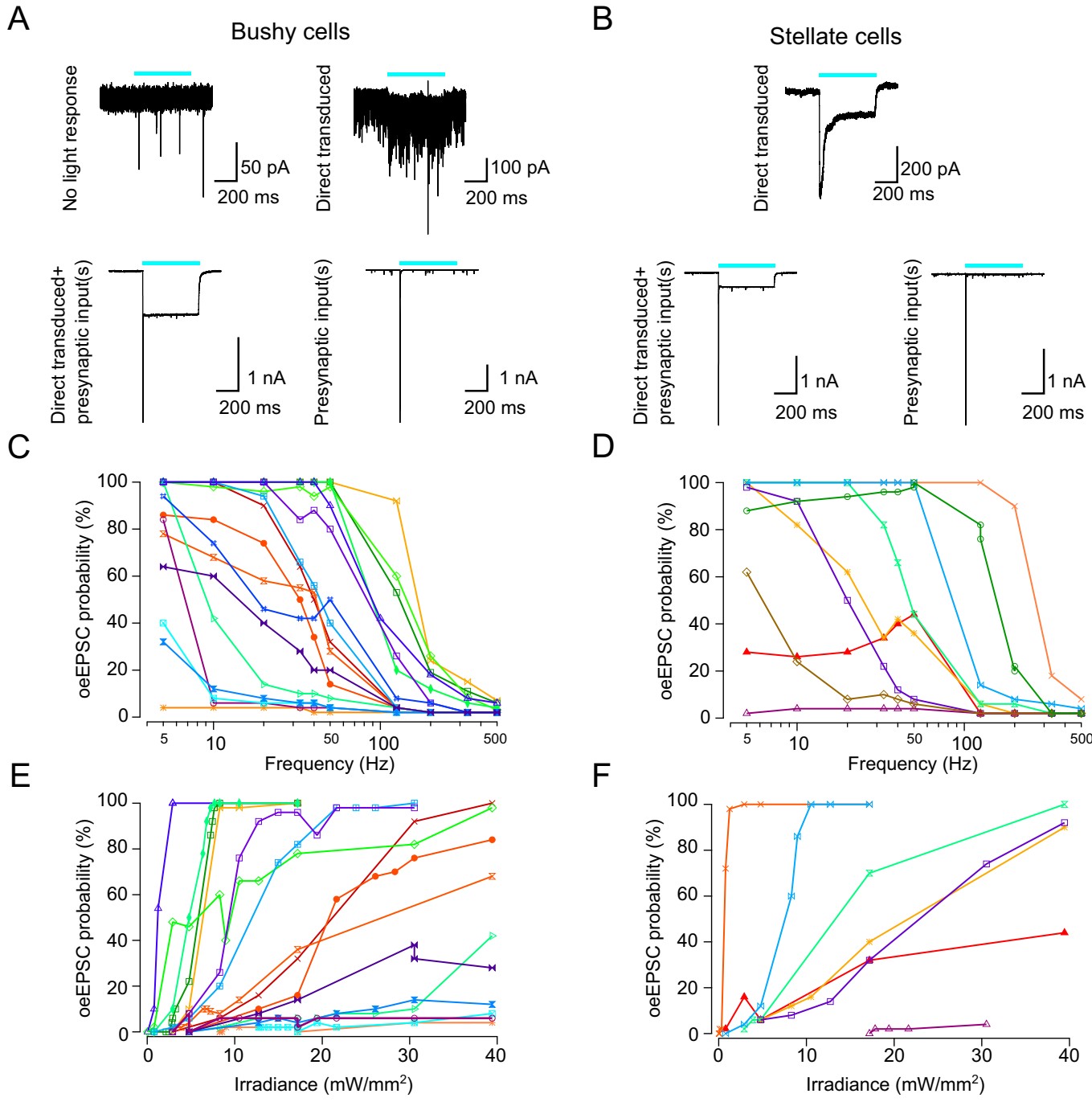

**Figure EV4. Variability of photoresponses in principal cells of the AVCN.**

(A, B), Recordings upon long light stimulation (500 ms, 488 nm, ~40 mW/mm²) of either bushy (A) or stellate (B) cells, indicating transduced principal cells (directly transduced) and non-transduced SGNs, a combination of directly transduced principal cells + transduced SGN presynaptic inputs), or none of them (no light response). Dependence of oeEPSC probability on the stimulation frequency ((C), for bushy cells; (D), for stellate cells) and the irradiance ((E), for bushy cells; (F), for stellate cells) in principal cells only receiving presynaptic input(s).

