## [Peer Review File · EMBO Molecular Medicine]

Channelrhodopsin variants for high-rate optogenetic neurostimulation at low light intensities

Thomas Mager, Tobias Moser, Lennart Roos, Aida Garrido-Charles, Alexey Alekseev, Anupriya Thirumalai, Artur Mittring, Theocharis Alvanos, Antoine Huet, Ernst Bamberg, Kathrin Kusch, Bettina Wolf, Niels Albrecht, Anna Vavakou, and Martina Bleyer

Corresponding authors: Thomas Mager (thomas.mager@med.uni-goettingen.de) , Tobias Moser (tmoser@gwdg.de)

Review Timeline:

Submission Date:	14th Mar 25
Editorial Decision:	21st Apr 25
Revision Received:	7th Sep 25
Editorial Decision:	10th Oct 25
Revision Received:	14th Nov 25
Accepted:	17th Nov 25

Editor: Jingyi Hou

Transaction Report:

23rd Apr 2025

Dear Tobias,

Thank you again for submitting your work to EMBO Molecular Medicine. We have now received feedback from the two referees who agreed to evaluate your manuscript. As you will see in the reports below, the referees are overall supportive but have raised several concerns that will need to be thoroughly addressed in a major revision of the manuscript.

The referees' recommendations are clear, so I won't repeat the points listed below. In particular, the concerns regarding the long-term stability and safety of f-ChR2 TC need to be addressed. Additionally, we kindly ask that you revise your manuscript to place greater emphasis on the biomedical aspects, improve overall clarity, and make the content more accessible to the general audience of EMBO Mol Med. All other issues raised by the referees need to be addressed as well.

We would welcome the submission of a revised version within three months for further consideration. As you may already know, our editorial policy allows in principle a single round of major revision, and it is therefore essential to provide responses to the referees' comments that are as complete as possible.

Please also contact us as soon as possible if similar work is published elsewhere. If other work is published, we may not be able to extend the revision period beyond three months.

I look forward to receiving your revised manuscript.

Kind regards,
Jingyi

Jingyi Hou
Senior Editor
EMBO Molecular Medicine

We require:

- 1) A .docx formatted version of the manuscript text (including legends for main figures, EV figures and tables). Please make sure that the changes are highlighted to be clearly visible.
- 2) Individual production quality figure files as .eps, .tif, .jpg (one file per figure). For guidance, download the 'Figure Guide PDF': (<https://www.embopress.org/page/journal/17574684/authorguide#figureformat>).
- 3) A .docx formatted letter INCLUDING the reviewers' reports and your detailed point-by-point responses to their comments. As part of the EMBO Press transparent editorial process, the point-by-point response is part of the Review Process File (RPF), which will be published alongside your paper.
- 4) A complete author checklist, which you can download from our author guidelines (<https://www.embopress.org/page/journal/17574684/authorguide#submissionofrevisions>). Please insert information in the checklist that is also reflected in the manuscript. The completed author checklist will also be part of the RPF.

6) It is mandatory to include a 'Data Availability' section after the Materials and Methods. Before submitting your revision, primary datasets produced in this study need to be deposited in an appropriate public database, and the accession numbers and database listed under 'Data Availability'. Please remember to provide a reviewer password if the datasets are not yet public (see <https://www.embopress.org/page/journal/17574684/authorguide#dataavailability>).

12) Author contributions: You will be asked to provide CRediT (Contributor Role Taxonomy) terms in the submission system. These replace a narrative author contribution section in the manuscript.

13) A Conflict of Interest statement should be provided in the main text.

14) Every published paper now includes a 'Synopsis' to further enhance discoverability. Synopses are displayed on the journal

webpage and are freely accessible to all readers. They include a short stand first (maximum of 300 characters, including space) as well as 2-5 one-sentences bullet points that summarizes the paper. Please write the bullet points to summarize the key NEW findings. They should be designed to be complementary to the abstract - i.e. not repeat the same text. We encourage inclusion of key acronyms and quantitative information (maximum of 30 words / bullet point). Please use the passive voice. Please attach these in a separate file or send them by email, we will incorporate them accordingly.

Please also suggest a visual abstract to illustrate your article as a PNG file 550 px wide x 300-600 px high.

15) All Materials and Methods need to be described in the main text using our 'Structured Methods' format. According to this format, the Methods section includes a Reagents and Tools Table (listing key reagents, experimental models, software and relevant equipment and including their sources and relevant identifiers) followed by a Methods and Protocols section describing the methods, ideally using a step-by-step protocol format. The aim is to facilitate adoption of the methodologies across labs.

Please download and fill our Reagents and Tools Table template (.docx), which you can find in our author guidelines: <https://www.embopress.org/page/journal/17574684/authorguide#structuredmethods>

When submitting your revised manuscript, please DO NOT include the Reagents and Tools Table in the Methods section of the manuscript but upload it as a separate file choosing the file type "Reagent Table".

**** Reviewer's comments ****

Referee #1 (Comments on Novelty/Model System for Author):

Dear Dr Hou, the paper and the data are great and very convincing. However, it is written for a more biophysical audience, so I would recommend streamlining it and focusing on the biomedical aspects. Please see my comments to the authors for details.

Referee #1 (Remarks for Author):

The paper "Channelrhodopsin variants for high-rate optogenetic neurostimulation at low light intensities" by Roos et al. presents a sophisticated biophysical characterisation of an advanced ChR2 derivative called f-ChR2 TC. It is being tested in various cell lines and model systems with a view to its application in optogenetic hearing restoration. The depth of data provided is impressive, but also overwhelming. It might be helpful to reduce the data and figures provided and tailor them to the main messages/findings of the manuscript. The provision of expanded figures and, on top of this, appendix figures is an overload and reduces clarity and readability. There is also a huge imbalance towards the biophysical aspects and more biomedical impact would be desirable for the readership of this journal. Otherwise, the manuscript is very well written. Scientifically, the question remains as to why the authors now focus on a blue-shifted optogenetic tool for hearing restoration, having previously presented red-shifted versions. Blue light can cause phototoxicity. The manuscript would benefit greatly if the authors provided a comprehensive rationale for why the described enhanced f-ChR2-TC is better for biomedical applications than, for example, f-ChRimson. In this regard, it would be great if the authors could discuss or provide data on whether the enhanced expression of f-ChR2 TC and its sensitivity to blue light does not cause photo- or cytotoxicity over a meaningful period of time. In particular, both the in vivo and in vitro experiments were performed a few days after AAV delivery. It is unclear whether f-ChR2 TC is stable, functional and safe over time.

Minor points:

Line 67: "spectral coding" is not introduced - what is meant by this?

Line 173 (Figure 1F - right panel, x-axis label): is f-ChR2 TC meant?

Line 210: What is "ultrafast f-Chronos"?

Line 214: It would be great to show representative fluorescence images in the main figures.

Line 346: What is known about the cytotoxicity of optogenes after AAV overexpression?

Line 351: GFP or EYFP expression?

Line 418: Could you correlate optogene expression levels with better performance in bushy cells or could this be a reason for the differences observed?

Line 463: Which 'ChR' versions are meant?

Referee #2 (Remarks for Author):

General comments :

This study strength lies in the engineering and comprehensive characterization of a novel blue-light-sensitive channelrhodopsin variant, f-ChR2 TC, which uniquely combines fast closing kinetics with good plasma membrane expression and low light requirements, enabling efficient and safe high-rate neurostimulation. Through in vitro and in vivo experiments in the auditory pathway, the study demonstrates that f-ChR2 TC outperforms other blue-light activated ChRs and compares favorably to advanced red-light options, highlighting its potential for basic research and preclinical optogenetic therapies, particularly for hearing restoration.

The study is well designed, from careful characterisation of photocurrent kinetics in non-neuronal cells to quantification of the optogenetic auditory brainstem response. Interpretation of the results are deepened by the measurement made at these two levels, and the role of the introduced mutations at the molecular level, is elegantly outlined by the functional results obtained at the level of the full organism. The study also delves into the synaptic response driven by optogenetic stimulation, an important step into the comprehension of critical parameters for transmission and integration of a restored percept.

Specific comments:

1-

The large number of variants makes it difficult to clearly understand the relationships between the different constructs. Including a summary figure, such as a table or a phylogenetic-style tree, could help illustrate and organize the various optimization strategies explored in this study. Additionally, the use of "ChRs" as a generic term for channelrhodopsins and "ChR2" for the specific Channelrhodopsin-2 can be confusing—especially since some variants are derived from ChR2 mutations, while others originate from different microorganisms (e.g., *S. helveticum* for Chronos, *C. reinhardtii* for ChR2, or *C. noctigama* for Chrimson).

2-

Line 217 "... the success rate was low for f-Chronos expressing neurons reaching a spike probability of 80 to 100 % in only 3 out of 14 neurons (Fig. 2)."

The statement should be clarified, as a spike probability of 80~100% in only 3 out of 14 shows great limitation of the construct. It should be clear what type of stim was done here (it is clarified later, but should be stated here). It also appears misleading to display f-Chronos in Fig 2A if there is such a decreased efficacy.

Line 219 "i) suboptimal plasma membrane targeted expression, which is reflected in the comparatively low stationary photocurrent density values measured at saturating light intensities (Fig. 2)"

In this regard, I presume, to the result in Fig 2B, then I'd recommend to display in Fig 2A the photocurrent quantified in Fig 2B rather than the spike trains (although, showing both is fine). As photocurrents are displayed in Fig 1E, I wonder if the recordings show similar kinetics in neurons and NG cells.

3-

Line 222 "ii) the limited charge transfer evoked by the short (1 ms) light pulses given the ultrafast channel closing kinetics (Fig. EV1, Fig. 1)."

For clarity, I suggest to separate the figure references "... by the short light pulses (Fig EV1) ... closing kinetics (Fig 1B)". It would also be easier to understand the sentence meaning if the charge transfer was quantified in Fig EV1. Alternatively, rewrite the sentence to clarify that charge transfer is illustrated by the normalized stationary current.

4-

Line 270 "oABRs mediated by f-ChR2 TC compared favorably to published results obtained with ultrafast ChR variants: Chronos and f-Chrimson (Fig. 3 E-G, replotted from Keppeler et al. 2018 and Zerche et al. 2023)."

The statement is a bit unfair, and leads to the notion that f-ChR2 TC outperforms the other constructs. While true for Chronos, f-Chrimson seems on par with f-ChR2 TC here. Adjust phrasing accordingly.

5- Major remark

My next point concerns the reported reduction in SGN density following injection. The authors should comment on the magnitude of this reduction, especially since the effect appears to vary between constructs. This variability weakens the argument that the observed SGN loss is solely due to injection pressure. To strengthen this hypothesis, it would be important to include a control group in which mice are injected with a saline solution or better yet an AAV with YFP alone. The authors could also discuss ways to adapt the injection protocol to reduce this unwanted effect.

Additionally, in Figure EV3, the rationale behind the left/right panel separation is unclear. If the right side corresponds to the injected cochlea and the left to the non-injected one, it is important to clarify what was injected in the control group, and why YFP-positive SGNs are visible on the right side in panel EV3-D.

It is also important for a potential therapeutic development to assess if the reduction is due to the pressure injection, a toxic overexpression of the transgene, or an immune response. As the reduction in the number of cells is important, it should be possible to localise the dead cells, or identify the lost volume?

It might be worth to relate the observation with what seems like a reduced PV expression in the f-ChR2 TC.

6-

I acknowledge the challenges involved in analyzing the subcellular localization of the optogene from cryosections, and the authors have made a commendable effort through careful quantification and appropriately cautious terminology in describing their findings. While the methods section outlines the procedure for the line analysis, I was unable to find a clear explanation of how the ratio presented in Figure 4-C was calculated. Furthermore, it would have been beneficial to perform membrane surface staining in at least a small subset of tissue to validate the quantification.

7-

line392 " injected (P5-P9) C57BL/6 wild-type mice after a standardized expression period of ~ 2 weeks"
it is weird to have a standard being approximatively 2 weeks, please rephrase or clarify.

8-

line 404 " Thus, by reducing the light irradiance, fewer inputs are recruited, which is reflected by smaller step sizes in the oeEPSC amplitude eventually arriving at monosynaptic input (single endbulb synapse)."
From my understanding, this method is not used here. If it is indeed the case, please consider rephrasing, as it may confuse the reader.

Line 409" Where positive," what is positive here? Typo?

9- Major point

To measure the maximal frequency driving synaptic inputs the authors use variable frequency and saturating light (@ 40mW/mm²). While it help normalize the variability in light sensitivity, isn't it counterproductive in term of maximal frequency responses? I would assume that saturating light might increase the oeEPSC probability reduction observed in the train of stimuli.

10- Major point

Line 416 "Quantification of oeEPSC probability versus irradiance and frequency of stimulation shows that bushy cells that are more light-sensitive in terms of higher oeEPSC probability and shorter synaptic delay also follow higher frequencies of stimulation (Fig. 5 E-G)."

This aspect is a crucial component of the synaptic findings and deserves a clearer representation. While the current data allow the reader to infer the conclusion by comparing individual cells, the current layout is suboptimal. Notably, the line colors in panels E/F and G are inconsistent. The authors should consider a more effective way to illustrate this relationship within a single panel. One possible solution would be to extract, for each cell, a single irradiance or frequency value corresponding to a 50% oEPSC probability, and plot these values against one another.

11-

Line 421 "Given the fast-closing kinetics of f-ChR2 TC at physiological temperature ($\tau_{off} \sim 5$ ms, Table EV2), we reason that limited expression and photocurrent desensitization of f-ChR2 TC as well as a potential depolarization block hinder endbulb synapses to follow stimulation frequencies of more than 100 Hz."

The rationale behind this hypothesis is not fully elaborated. While the limited expression aspect is relatively straightforward, the desensitization observed with f-ChR2 TC should be more thoroughly discussed. Additionally, the role of depolarization block remains unclear. Since the authors refer to results from cells receiving "optogenetic inputs" rather than directly expressing the opsin, does this imply that the depolarization block occurs at the presynaptic terminals? Typically, a depolarization block would be expected in cells with higher-than-average opsin expression, which might be reflected by a shorter synaptic delay. Is such a correlation observed in this study?

12-

Fig EV4: does the panels D and F relate to stellate cells and C and E to bushy cells? If so, it is not clearly indicated in the legend.

13-

Line 437 "However, low oeEPSC probability was not always predicted by long latency, as shown exemplarily for the endbulb synapses depicted in Appendix Fig. S5 B-C, and compared to other synapses with similar oeEPSC probability."

I might misunderstand the point, but the example referred to in the legend of Fig S5 B-C seems to have a low probability AND a long latency (green dots). It may be the phrasing that is unclear as the legend indicate:

"Note that low oeEPSC probability was not always predicted by long latency, as shown exemplarily for the endbulb synapses depicted in green dots, and compared to other synapses with similar oeEPSC probability (displayed by purple hourglass icon () and red square). »

The current phrasing suggests that it is specifically for the green dots that the oeEPSC probability fails to predict long latency.

14-

Fig S6 C legend: « Quantification of synaptic delay dependent on frequency of stimulation (~ 40 mW/mm², 1 ms; continuous line) and dependent on irradiance (10 Hz, 1 ms; dashed line). «

does not fit with the actual figure, i only have continuous line, and 1 X axis, rendering issue?

Discussion:

It would be valuable to further elaborate on strategies that could help minimize desensitization.

Since neurons can differ in their intrinsic properties-affecting, for instance, the maximal sustainable firing rate-the authors should also expand on how the results observed in hippocampal neurons relate to the later findings in SGNs or oABR recordings.

Line 551 : "Limitations explaining failure or sustained high rate..." should be "...OF sustained high rate"

Although, that sentence remain unclear even with the proposed modification

Methods:

Line 656: T_{off} is determined as a monoexponential fit of the current, but there is no quantification of the goodness of the fit?

Could a double exponential lead to better fit?

It is then mentioned line 796 that 10/15 cells have a biexponential fit, without much explanation.

Line 663: the authors mention peak recovery of ChRs, where is that data presented?

Line 858 : the sentence is uncomplete (no verb).

RESPONSE TO REVIEWER REPORT(S):

Referee: 1

Referee #1 (Comments on Novelty/Model System for Author):

Dear Dr Hou, the paper and the data are great and very convincing. However, it is written for a more biophysical audience, so I would recommend streamlining it and focusing on the biomedical aspects. Please see my comments to the authors for details.

We would like to thank the reviewer for the thorough review of our manuscript. We understand the Referee's assessment of the scope of our study. Thus, we have streamlined the manuscript towards its biomedical aspects and revised our manuscript substantially by including additional experiments in a new main figure highlighting translational transferability of our newly engineered channelrhodopsin. Here, we employed f-ChR2 TC in Mongolian gerbils, which serve as a translational rodent model in hearing research as they offer a larger cochlea and a more human-like low frequency hearing than mice and rats. These experiments demonstrated low light requirements for activating the auditory pathway by f-ChR2 TC mediated optogenetic stimulation of the auditory nerve. We then used LED-based multichannel optical cochlear implants and recorded the ensuing multi-unit activity in the inferior colliculus of these gerbils by multielectrode array recordings and confirmed sub- μ J light requirements and good temporal fidelity of coding.

Referee #1 (Remarks for Author):

The paper "Channelrhodopsin variants for high-rate optogenetic neurostimulation at low light intensities" by Roos et al. presents a sophisticated biophysical characterisation of an advanced ChR2 derivative called f-ChR2 TC. It is being tested in various cell lines and model systems with a view to its application in optogenetic hearing restoration. The depth of data provided is impressive, but also overwhelming. It might be helpful to reduce the data and figures provided and tailor them to the main messages/findings of the manuscript. The provision of expanded figures and, on top of this, appendix figures is an overload and reduces clarity and readability. There is also a huge imbalance towards the biophysical aspects and more biomedical impact would be desirable for the readership of this journal. Otherwise, the manuscript is very well written.

We would like to thank the reviewer for the appreciation of our work and the advice to further improve our manuscript. In response to the reviewer's comment, we have completely overhauled the manuscript taking the suggestions of the reviewer into account. In particular we have better aligned the balance of the manuscript toward the biomedical application in the auditory system, including in the Mongolian gerbil.

Scientifically, the question remains as to why the authors now focus on a blue-shifted optogenetic tool for hearing restoration, having previously presented red-shifted versions. Blue light can cause phototoxicity. The manuscript would benefit greatly if the authors provided a comprehensive rationale for why the described enhanced f-

ChR2-TC is better for biomedical applications than, for example, f-ChRimson. In this regard, it would be great if the authors could discuss or provide data on whether the enhanced expression of f-ChR2 TC and its sensitivity to blue light does not cause photo- or cytotoxicity over a meaningful period of time.

In particular, both the in vivo and in vitro experiments were performed a few days after AAV delivery. It is unclear whether f-ChR2 TC is stable, functional and safe over time.

We fully understand the reviewers concerns and appreciate the comment. While previous studies indeed described the benefits of red-shifted f-Chrimson, we now focused on a blue-shifted optogenetic tool for two reasons.

First, we aimed to engineer a well-balanced channelrhodopsin to combine robust current densities while maintaining fast closing kinetics to allow for optogenetic stimulation of the cochlea at low pulse energy thresholds and at near physiological temporal fidelity. Thus, we now have laid the basis for further preclinical assessment of optogenetic hearing restoration and its efficacy in gerbils and non-human primates employing μ LED oCIs and its comparisons to eCIs. To date, this is not possible using red-shifted ChRs as oCI prototypes emitting red-light are still in development. Second, a blue-light-shifted ChR with similar properties to f-Chrimson, such as f-ChR2 TC, will ultimately help to expand the current dynamic range limitation of oCIs by facilitating SGN subtype specific activation using two non-overlapping spectra of visible light when combining blue and red-light activated ChRs for dual color stimulation.

However, we agree with the reviewer that cyto- and phototoxicity upon expression of blue-light activated ChRs needs to be carefully evaluated and tested over a meaningful period of time and is highly warranted for any clinical translation. In this regard the lab has already published previous work on stable, longevous expression of ChRs (f-Chrimson) in mice that we discussed in our manuscript and now highlighted to a greater extend (Bali et al. 2022). Further investigation on potential phototoxic effects may be out of the scope of this manuscript. Nevertheless, development of ChRs such as f-ChR2 TC, with low energy thresholds for activation will ultimately contribute in reducing phototoxicity when using blue light.

Minor points:

Line 67: "spectral coding" is not introduced - what is meant by this?

We thank the reviewer for the observation and replaced "spectral coding" by "sound frequency coding" for clarification.

Line 173 (Figure 1F - right panel, x-axis label): is f-ChR2 TC meant?

We thank the reviewer for the question and apologize for the confusion. The right panel of Fig. 1F, highlights the comparison of the current densities of ChR2 TC and CatCh, which are two other ChR2 variants. The current densities of f-ChR2 TC are shown in the middle panel of Fig. 1F. Further, the panels in Figure 1B and 1F are designed to compare channel closing kinetics (B) and current densities (F) for: very fast channelrhodopsins (left panel) with closing kinetics faster than 10 ms; medium-

fast channelrhodopsins (middle panel) with closing kinetics around 10 ms, and slow channelrhodopsins (right) with closing kinetics slower than 10 ms.

Line 210: What is "ultrafast f-Chronos"?

We apologize for the confusion cause and have now clarified the main text: f-Chronos is the ultrafast ChR variant Chronos F236Y (see the revised sections below):

Lines 107-108

“Toward the first strategy, we generated Chronos F236Y (fast Chronos: f-Chronos), which, to our knowledge, is the fastest closing ChR to date.”

Lines 121-125

“Indeed, as shown by whole-cell patch clamp experiments in transiently transfected NG cells, the ultrafast ChR Chronos F236Y (f-Chronos, $\tau_{\text{off}} = 1.7 \pm 0.1$ ms, $n = 4$ at room temperature (RT), $\tau_{\text{off}} = 0.8 \pm 0.1$ ms, $n = 4$ at 33 °C) was considerably faster than Chronos ($\tau_{\text{off}} = 3.1 \pm 0.5$ ms, $n = 6$ at RT (***) $p = 0.0009$, unpaired t-test with Welch’s correction), $\tau_{\text{off}} = 1.9 \pm 0.5$ ms, $n = 6$ at 33 °C (** $p = 0.021$, unpaired t-test with Welch’s correction); **Fig. 1 A-B; Table EV1-2**”

Line 214: It would be great to show representative fluorescence images in the main figures.

We included the respective images.

Line 346: What is known about the cytotoxicity of optogenes after AAV overexpression?

Changes in neuronal biology up to toxicity and neural loss e.g. due to proteostatic stress following AAV-mediated overexpression of proteins present a risk also in optogenetic therapies (Kleinlogel *et al*, 2020; Miyashita *et al*, 2013; Stone *et al*, 2025). In response to this comment, we have performed additional histological analysis of AAV-injected left cochleae and compared them to the non-injected right cochlea, which also contained optogenetically modified SGNs. With the AAV doses used we found signs of neural pathology (vacuolization and neural loss) yet no evidence for innate or adaptive immune response (**Appendix Fig. S5**). We have now included these results in the manuscript and also discussed potential mechanisms of cytotoxicity.

Attached respective section of the manuscript:

Lines 351-357

“ SGN loss might be an aftermath of the intracochlear pressure injection at young age. In addition, we cannot rule out potential cytotoxicity due to protein overload based proteostatic stress (Stone *et al*. 2025) or immune response following AAV-mediated overexpression. Further, histological and immunohistochemical investigation of cochlear paraffin sections can be found in **Appendix Fig. S5**. In brief, we observed neuropathological changes in SGNs but did not find evidence for a prevailing adaptive or innate immune response.”

”
.

Line 351: GFP or EYFP expression?

We thank the reviewer for noticing the inconsistency. Our transgene is composed of the respective ChR and if needed enhancing sequences as well as the reporter EYFP. Thus, it is EYFP expression. GFP antibodies were used for immunolabelling of EYFP in immunohistochemistry.

Line 418: Could you correlate optogene expression levels with better performance in bushy cells or could this be a reason for the differences observed?

We agree to the reviewer that ChR expression is a likely reason for the differences observed. As described in the main text, we observed a correlation between synaptic delay and the light sensitivity as well as the maximal frequency. Spike latency and therefore synaptic delay depends on photocurrent size, which in turn depends on the expression. We modified the following sentence, which is following the description of the correlation:

Lines 433-437

“Given the fast-closing kinetics of f-ChR2 TC at physiological temperature ($T_{off} \sim 5$ ms, **Table EV2**), we reason that limited expression and photocurrent desensitization of f-ChR2 TC as well as a potential depolarization block led to the observed heterogeneity of endbulb synaptic transmission to bushy cells and hindered most endbulb synapses to follow stimulation frequencies of more than 100 Hz.”

Line 463: Which 'ChR' versions are meant?

We thank the reviewer and corrected the legend of Figure 5 in order to specify the ChR version.

Referee #2 (Remarks for Author):

General comments :

This study strength lies in the engineering and comprehensive characterization of a novel blue-light-sensitive channelrhodopsin variant, f-ChR2 TC, which uniquely combines fast closing kinetics with good plasma membrane expression and low light requirements, enabling efficient and safe high-rate neurostimulation. Through in vitro and in vivo experiments in the auditory pathway, the study demonstrates that f-ChR2 TC outperforms other blue-light activated ChRs and compares favorably to advanced red-light options, highlighting its potential for basic research and preclinical optogenetic therapies, particularly for hearing restoration.

The study is well designed, from careful characterisation of photocurrent kinetics in non-neuronal cells to quantification of the optogenetic auditory brainstem response. Interpretation of the results are deepened by the measurement made at these two levels, and the role of the introduced mutations at the molecular level, is elegantly outlined by the functional results obtained at the level of the full organism. The study also delves into the synaptic response driven by optogenetic stimulation, an important step into the comprehension of critical parameters for transmission and integration of a restored percept.

We would like to thank the reviewer for the appreciation of our work and the advice to further improve our manuscript. We have completely overhauled the manuscript taking the reviewer's recommendations into account.

Specific comments:

1-

The large number of variants makes it difficult to clearly understand the relationships between the different constructs. Including a summary figure, such as a table or a phylogenetic-style tree, could help illustrate and organize the various optimization strategies explored in this study. Additionally, the use of "ChRs" as a generic term for channelrhodopsins and "ChR2" for the specific Channelrhodopsin-2 can be confusing—especially since some variants are derived from ChR2 mutations, while others originate from different microorganisms (e.g., *S. helveticum* for Chronos, *C. reinhardtii* for ChR2, or *C. noctigama* for Chrimson).

We thank the reviewer for the comment and completely understand that this can cause confusion. Following the reviewer's comment, we included "**Appendix Table S2**" to summarize all the variants and optimization strategies used in this study.

2-

Line 217 "... the success rate was low for f-Chronos expressing neurons reaching a spike probability of 80 to 100 % in only 3 out of 14 neurons (Fig. 2)."

The statement should be clarified, as a spike probability of 80~100% in only 3 out of 14 show great limitation of the construct. It should be clear what type of stim was done here (it is clarified later, but should be stated here). It also appears misleading to display f-Chronos in the Fig2A if there is such a decreased efficacy.

Line 219 "i) suboptimal plasma membrane targeted expression, which is reflected in the comparatively low stationary photocurrent density values measured at saturating light intensities (Fig. 2)" this refer, I presume, to the result in Fig2B, then I'd recommend to display in Fig2A the photocurrent quantified in Fig2B rather than the spike trains (although, showing both is fine). As photocurrents are displayed in fig 1E, I wonder if the recordings show similar kinetics in neurons and NG cells.

We thank the reviewer for pointing this out and adapted the manuscript accordingly. We exchanged the f-Chronos trace in Fig. 2 A (now **Fig. 2 E**) for a more representative recording showing a reduced spike probability upon photostimulation by a train of 50 light pulses with a pulse length of 1ms. Moreover, we included **Fig. 2 B** showing representative photocurrent recordings (quantified in **Fig. 2 C**). The comparison of the channel closing kinetics in hippocampal neurons (**Fig. 2 D**) and NG cells (**Fig. 1 B**), which is provided in the new **Appendix Figure S2**, shows that the closing kinetics are similar in the different cell types.

Lines 215-225

"The photocurrent densities and τ_{off} values determined in hippocampal neurons were similar to the values obtained in NG cells (**Fig. 1, Fig. 2B, Appendix Fig. S3, Appendix Table S1**). Whereas Chronos LC, f-ChR2 TC and CatCh enabled reliable neuronal photostimulation by the short (1 ms) light pulses, spike probability for f-Chronos expressing neurons was low, with 20% of the neurons (3 out of 14) showing a spike probability higher than 80% (**Fig. 2C**). The low success rate in f-Chronos expressing neurons likely results from i) suboptimal plasma membrane targeted expression, which is reflected in the comparatively low stationary photocurrent density values measured at saturating light intensities (**Fig. 2D-E**) and ii) the limited charge transfer evoked by the short (1 ms) light pulses (**Fig. EV1D**) given the ultrafast channel closing kinetics (**Fig. 1B**)."

3-

Line 222 " ii) the limited charge transfer evoked by the short (1 ms) light pulses given the ultrafast channel closing kinetics (Fig. EV1, Fig. 1). "

For clarity, I suggest to separate the figure references "... by the short light pulses (Fig EV1) ... closing kinetics (fig 1B)". It would also be easier to understand the sentence meaning if the charge transfer was quantified in fig EV1. Alternatively rewrite the sentence to clarify that charge transfer is illustrated by the normalized stationary current.

We thank the reviewer for the comment, which helped to improve clarity. We referred to the following relation. In transient currents evoked by brief light pulses that are shorter than ChR photocycle time, which is approximated by τ_{off} , the transferred charge (Q) is proportional to the current decay kinetics (ChR on-kinetics is usually beyond the time resolution of whole-cell patch-clamp experiments). For clarity, we added **Fig. EV 1 D**.

4-

Line 270 "oABRs mediated by f-ChR2 TC compared favorably to published results

obtained with ultrafast ChR variants: Chronos and f-Chrimson (Fig. 3 E-G, replotted from Keppeler et al. 2018 and Zerche et al. 2023)." The statement is a bit unfair, and lead to the notion that f-ChR2 TC outperform the other constructs. While true for Chronos, f-Chrimson seems on par whit f-ChR2 TC here. Adjust phrasing accordingly.

We thank the reviewer for the comment and adjusted the respective phrasing accordingly:

Lines 277- 282:

"We did not observe significant differences of blue-light activated f-ChR2 TC compared to red-light activated f-Chrimson, while f-ChR2 TC significantly outperformed Chronos for I) threshold: $p = 0.0002$; II) P1-N1 amplitude: $p = 0.001$ and III) P1-N1 latency: $p = 0.0002$ (unpaired Kruskal-Wallis test adjusted for multiplicity by Dunn's correction, **Fig. 3 E-G**, replotted from Keppeler et al. 2018 and Zerche et al. 2023)."

5- Major remark

My next point concerns the reported reduction in SGN density following injection. The authors should comment on the magnitude of this reduction, especially since the effect appears to vary between constructs. This variability weakens the argument that the observed SGN loss is solely due to injection pressure. To strengthen this hypothesis, it would be important to include a control group in which mice are injected with a saline solution or better yet an AAV with YFP alone. The authors could also discuss way to adapt the injection protocol to reduce this unwanted effect.

We understand the reviewers concern and appreciate the comment. We agree that the suggested inclusion of control groups such as injections of saline solutions and/or AAVs carrying EYFP alone would help better understand the mechanisms of the observed SGN reduction. Indeed, some of our previous studies have addressed these points. For example, based on a small sample size, we did not find significantly different SGN densities between postnatal injections of AAV-f-Chrimson or saline (analyzed at 3-months of age (Bali *et al*, 2022)). Moreover, as in the present MS, we have typically compared results in the AAV-injected ear to those of the non-injected ear (which typically also shows ChR expression due to CSF spread of AAV, (e.g. Mager *et al*, 2018; Keppeler *et al*, 2018; Bali *et al*, 2021, 2022). There, we typically find less pronounced SGN loss, suggesting an effect of injection and potentially of viral dose. We note that all findings are confounded by the substantial age-dependent SGN loss in C57 black 6 mice (Someya *et al*, 2009) which make it difficult to assess a treatment-related adverse effect (Bali *et al*, 2022).

Evidence for toxic effects of AAV as well as of fluorescent proteins or tagged ChR has been presented for other neurons/cells (Liu *et al*, 1999; Klein *et al*, 2006; Miyashita *et al*, 2013; Maimon *et al*, 2018).

In response to the reviewer's comment, we have performed additional experiments (**Appendix Fig. S5**). Given the report of immunological reactions to optogenetically rendered peripheral mammal neurons by microbial ChRs (Maimon et al. 2018) we probed sections of treated cochleae for signs of induced (CD3 positive cells) and innate (Iba-1 positive cells) immune response. In short, we did not encounter cells and so either the response proceeded before or was not prevalent. In addition, we performed a histopathological scoring based on hematoxylin and eosin (HE) stains of cochlear sections and found lower scores for neuron density and increased interstitial vacuolation compared with the contralateral, non-injected side.

Moreover, we have revised the **discussion** accordingly i) pointing out more translational AAV administration approaches such as via slow micropump-catheter systems with a pressure vent in the cochlea that are now also used in clinical *OTOF* gene therapy trials, and ii) discussing the need for further investigations to decipher the mechanism(s) of SGN reduction which we consider beyond the scope of this study.

Additionally, in Figure EV3, the rationale behind the left/right panel separation is unclear. If the right side corresponds to the injected cochlea and the left to the non-injected one, it is important to clarify what was injected in the control group, and why YFP-positive SGNs are visible on the right side in panel EV3-D.

We thank the reviewer for the remark and gladly provide clarification. As described above we compared the left, injected cochlea with the contralateral non-injected right cochlea from the same animal. In all of our studies with postnatally injected animals we found ChR expression in the contralateral non-injected cochlea likely due to AAV spread via the cochlea aqueduct(s) to/from the CSF space. This has now also been discussed more extensively in the manuscript. The black labelled control group is a non-injected littermate control of the injected animals, in which we did not detect any GFP-signal (zero eYFP+ SGNs/ $10^4\mu\text{m}^2$).

It is also important for a potential therapeutical development to assess if the reduction is due to the pressure injection, a toxic overexpression of the transgene, or a immune response. As the reduction in the number of cells is important, it should be possible to localise the dead cells, or identify the lost volume? It might be worth to relate the observation with what seems like a reduced PV expression in the f-ChR2 TC.

We thank the reviewer for this comment and refer to our above response. In brief, we performed additional experiments including labeling for immune cells and histopathology on cochleae treated with AAV-f-ChR2 TC and AAV-Chronos LC that we now included as **Appendix Figure S5**.

6-

I acknowledge the challenges involved in analyzing the subcellular localization of the optogene from cryosections, and the authors have made a commendable effort

through careful quantification and appropriately cautious terminology in describing their findings. While the methods section outlines the procedure for the line analysis, I was unable to find a clear explanation of how the ratio presented in Figure 4-C was calculated. Furthermore, it would have been beneficial to perform membrane surface staining in at least a small subset of tissue to validate the quantification.

We thank the reviewer for appreciating our efforts. In response to the comment, we now present a clear explanation on how the fluorescence ratios based of the line profiles were calculated in the methods section:

Lines 1092-1094

“The ratio of membrane to intracellular fluorescence was calculated for each selected cell by dividing the mean fluorescence measured between 0.4 and 0.6 μm by the mean fluorescence measured between 1.4 and 1.6 μm .”

In addition, we further analyzed a small set of cryosections from a f-ChR2 TC transduced cochlea that we additionally immunolabeled for Na^+/K^+ ATPase (ATP1A3, which provides a nice plasma membrane staining) validating our quantification of the subcellular ChR distribution: the shapes of the line profiles of f-ChR2 TC and ATP1A3 (n = 10 cells) agree very well (Rebuttal Figure 1).

Rebuttal Figure 1

7-

line392 " injected (P5-P9) C57BL/6 wild-type mice after a standardized expression period of ~ 2 weeks" it is weird to have a standard being approximatively 2 weeks, please rephrase or clarify.

We thank the reviewer for the observation and rephrased accordingly:

Lines 401-403:

"These slices were obtained from postnatally AAV-injected (P5-P9) C57BL/6 wild-type mice after an expression period of 15 ± 0.4 days on average, as described before (Özçete and Moser 2021; Hain and Moser 2024)."

Line 466:

"... an expression period of 15 days, used for the slice measurements..."

8-

line 404 " Thus, by reducing the light irradiance, fewer inputs are recruited, which is reflected by smaller step sizes in the oeEPSC amplitude eventually arriving at monosynaptic input (single endbulb synapse)." From my understanding, this method is not used here. If it is indeed the case, please consider rephrasing, as it may confuse the reader.

We thank the reviewer for this observation and excuse the confusion. We revised the MS accordingly for clarity:

Lines 414-418:

"Given ChR expression differences and the variability of SGN membrane resistance, capacitance and spiking threshold, by reducing the light irradiance, fewer inputs are recruited, which is reflected by smaller oeEPSC amplitudes eventually arriving at monosynaptic input (single endbulb synapse)."

Line 409" Where positive," what is positive here? Typo?

We understand the potentially confusing phrasing and want to point out that not all the cells that are successfully patched have a light-triggered response. By using "where positive", we wanted to refer to the cells with light-driven EPSCs (optically-evoked EPSCs) at least at maximum irradiance. For clarity we decided to delete the statement "Where positive".

9- Major point

To measure the maximal frequency driving synaptic inputs the authors use variable frequency and saturating light (@ 40mW/mm²). While it help normalize the variability in light sensitivity, isn't it counterproductive in term of maximal frequency responses? I would assume that saturating light might increase the oeEPSC probability reduction observed in the train of stimuli.

We thank the reviewer and accordingly provide further analyses of oeEPSC recordings following illumination with 50 stimuli of both saturating and subsaturating light at varying frequencies (see Rebuttal Fig. 2; recordings from four distinct bushy cells). As shown, reducing the light intensity consistently led to a decrease rather than an increase in oeEPSC probability within the high-frequency range in all cases.

We propose that this observation stems from the diminished photocurrent at lower light intensities, which resulted in longer latencies and fewer threshold-passing

events. Furthermore, we note that both the initial and sustained photocurrents increase not decrease with rising light intensities (see Rebuttal Fig. 3; f-ChR2 TC photocurrents at different light levels).

Rebuttal Figure 2

Off kinetics: ~10 ms

Rebuttal Figure 3

10- Major point
 Line 416 "Quantification of oeEPSC probability versus irradiance and frequency of stimulation shows that bushy cells that are more light-sensitive in terms of higher oeEPSC probability and shorter synaptic delay also follow higher frequencies of stimulation (Fig. 5 E-G)."

This aspect is a crucial component of the synaptic findings and deserves a clearer representation. While the current data allow the reader to infer the conclusion by comparing individual cells, the current layout is suboptimal. Notably, the line colors in panels E/F and G are inconsistent. The authors should consider a more effective way to illustrate this relationship within a single panel. One possible solution would be to extract, for each cell, a single irradiance or frequency value corresponding to a 50% oeEPSC probability, and plot these values against one another.

We apologize for the misunderstanding with the colors in panels E-G, we displayed the same cells with a dimmer tone of the same color to highlight the average in black. We now corrected and displayed all panels with the same color code to keep consistency and clarity.

11-

Line 421 "Given the fast-closing kinetics of f-ChR2 TC at physiological temperature ($\tau_{\text{off}} \sim 5$ ms, Table EV2), we reason that limited expression and photocurrent desensitization of f-ChR2 TC as well as a potential depolarization block hinder endbulb synapses to follow stimulation frequencies of more than 100 Hz."

The rationale behind this hypothesis is not fully elaborated. While the limited expression aspect is relatively straightforward, the desensitization observed with f-ChR2 TC should be more thoroughly discussed. Additionally, the role of depolarization block remains unclear. Since the authors refer to results from cells receiving "optogenetic inputs" rather than directly expressing the opsin, does this imply that the depolarization block occurs at the presynaptic terminals? Typically, a depolarization block would be expected in cells with higher-than-average opsin expression, which might be reflected by a shorter synaptic delay. Is such a correlation observed in this study?

We thank the reviewer for this important comment. As shown in Fig. 5, oeEPSC failures occur more frequently toward the end of the light stimulus train, with a lower oeEPSC probability in pulses 40–50 compared to pulses 1–10. We also observed an increase in oeEPSC latency over the course of the stimulus train (Appendix Fig. S8). Furthermore, pulsed light stimulation at high frequencies (>100 Hz) induces a level of desensitization comparable to that seen with continuous light stimulation (Fig. EV1 vs. Fig. 1E). Based on these observations, we reason that the reduced photocurrent amplitude at the end of the pulse train might be insufficient to trigger presynaptic terminal firing, thereby failing to induce postsynaptic oeEPSCs.

Regarding depolarization block, we agree with the reviewer that, intuitively, one might expect it to occur in cells with higher-than average expression levels. In such cases, light stimulation could induce a saturating photocurrent that impairs sustained neuronal firing and correlates with shorter synaptic delay. However, our dataset does not show that correlation (Appendix Fig. S8B). In contrast, high-performing endbulbs with higher oeEPSC probability at elevated stimulation frequencies also exhibit shorter synaptic latencies likely indicating high ChR expression. Given the absence of direct evidence for depolarization block in our data, we have revised the sentence in the results part accordingly.

Lines 433-437:

„Given the fast-closing kinetics of f-ChR2 TC at physiological temperature ($\tau_{\text{off}} \sim 5$ ms, Table EV2), we reason that limited expression and photocurrent desensitization of f-ChR2 TC led to the observed heterogeneity of endbulb synaptic transmission to bushy cells and hindered most endbulb synapses to follow stimulation frequencies of more than 100 Hz.”

Our interpretation of the depolarization block was directed/referring to the fact that the decay of photocurrent is also limited by the closing kinetics of f-ChR2 TC at 33 – 34 °C = 4.1 ± 0.96 ms. We reasoned that a substantial remaining photocurrent could have induced a depolarization block in the endbulb of Held in some cases. We attach here an exemplary recording from a bushy cell that expressed f-ChR2 TC to illustrate the notion of the large stationary photocurrent (Rebuttal Fig. 4). Nonetheless, this statement remains somewhat speculative since we did not record directly from the endbulb. However, as we cannot rule out the occurrence of depolarization blocks, we added the following sentence to the discussion:

Lines 594-597:

“In addition, potential depolarization block resulting from prolonged spiral ganglion neuron (SGN) photodepolarization due to the limited closing kinetics of f-ChR2 may have contributed to suboptimal sustained photostimulation at high frequencies.”

Rebuttal Figure 4: Whole cell recording of bushy cell directly expressing f-ChR2 TC during pulsed stimulation (1 ms, 488 nm, ~ 40 mW/mm²) at different frequencies.

12-

Fig EV4: does the panels D and F relate to stellate cells and C and E to bushy cells? If so, it is not clearly indicated in the legend.

We thank the reviewer for pointing this out and accordingly adjusted the legend to provide more clarity.

13-

Line 437 "However, low oeEPSC probability was not always predicted by long latency, as shown exemplarily for the endbulb synapses depicted in Appendix Fig. S5 B-C, and compared to other synapses with similar oeEPSC probability."

I might misunderstand the point, but the example referred to in the legend of Fig S5 B-C seems to have a low probability AND a long latency (green dots). It may be the phrasing that is unclear as the legend indicate:

"Note that low oeEPSC probability was not always predicted by long latency, as shown exemplarily for the endbulb synapses depicted in green dots, and compared to other synapses with similar oeEPSC probability (displayed by purple hourglass icon () and red square). »
The current phrasing suggests that it is specifically for the green dots that the oeEPSC probability fails to predict long latency.

We agree with the reviewer that **Fig S5 B-C** were not clearly explained and did not convey the intended message. We prepared the new **Appendix Fig. S8** (renamed Fig S5). In Appendix **Fig. S8 B** the latency of the first oeEPSC is correlated to the oeEPSC probability determined for each single cell plotted in **Appendix Fig. S8 A**. Cells with similar latency can show very distinct oeEPSC probabilities upon light stimulation with a train of 50 stimuli (1 ms pulse length at 5 Hz). Nonetheless, we observe a general trend: endbulb synapses with shorter latencies tend to have higher oeEPSC probabilities.

We rephrased the text and figure legend accordingly:

Line 447-448:

"However, low oeEPSC probability was not always correlating to long latencies (**Appendix Fig. S8 B**)."

Line 1623-1626:

"Note that low oeEPSC probability was not always correlating to long latencies, as shown exemplarily for the endbulb synapses depicted in red square when compared to other synapses with similar latency (displayed by cyan rhombus, purple hourglass icon (⊗) and blue triangular star)."

14-

Fig S6 C legend: « Quantification of synaptic delay dependent on frequency of stimulation (~ 40 mW/mm², 1 ms; continuous line) and dependent on irradiance (10 Hz, 1 ms; dashed line). «

does not fit with the actual figure, i only have continuous line, and 1 X axis, rendering issue?

We are thankful to the reviewer and apologize for this mistake in the figure legend, which we corrected accordingly:

Lines 1635 - 1636:

"C, Quantification of synaptic delay dependent on frequency of stimulation (~ 40 mW/mm², 1 ms; continuous line)."

Discussion:

It would be valuable to further elaborate on strategies that could help minimize desensitization.

Since neurons can differ in their intrinsic properties-affecting, for instance, the maximal sustainable firing rate-the authors should also expand on how the results observed in hippocampal neurons relate to the later findings in SGNs or oABR recordings.

We thank the reviewer for the valuable comments. Regarding strategies to optimize ChR properties, such as minimizing photocurrent desensitization, we now provide Appendix Table S2 and refer to recently published work (Huet et al., 2024; Alekseev et al., 2025). We agree to the reviewer that neuronal subtype-specific differences, such as the maximal sustainable firing rate, need to be considered. In response to the reviewer's advice, we accordingly provide the following sections.

Lines 225-229:

"The investigation of high-rate neurostimulation is impeded by the limited and heterogeneous intrinsic maximal firing rate of rat hippocampal neurons which is typically 40 to 60 Hz (Gunaydin et al. 2010; Mager et al. 2018), well below that of fast spiking neurons such interneurons and SGNs (Mager et al. 2018). We accordingly turned to the investigation of optogenetic SGN stimulation in the auditory pathway of mice."

Lines 586-587:

"In order to investigate high-rate stimulation, we turned to experiments in SGNs, fast spiking auditory neurons.

Lines 560-562:

"ChR variants with balanced channel closing kinetics, robust plasma membrane targeted expression, and low photocurrent desensitization are desirable for efficient and safe neurostimulation at high rates (Huet et al. 2024).

Lines 643-650:

"A recent study has demonstrated the potential of the engineered green light-activated ChR variant ChReef, to mitigate this issue (Alekseev et al. 2025). ChReef

enables sustained control of excitable cell activity at low light intensities, owing to its good plasma membrane targeting, minimal photocurrent desensitization, and comparatively high unitary conductance. Yet, the temporal fidelity of SGN activation is limited by ChReef's slow channel closing kinetics. The generation and comprehensive characterization of both natural and engineered ChRs will further fuel the development of advanced, kinetically balanced variants that can be safely integrated into future blue- or green- μ LED-based oCIs."

Line 551 : "Limitations explaining failure or sustained high rate..." should be "...OF sustained high rate"

Although, that sentence remain unclear even with the proposed modification

We thank the reviewer for noticing this typo. We corrected the sentence and improved it for clarity.

Lines 589-593:

"Limitations explaining oeEPSC failure during sustained high-rate neurostimulation include limited expression levels of f-ChR2 TC in SGN terminals, as indicated by the correlation of oeEPSC probability and light sensitivity and the increased synaptic delay compared to electrically triggered EPSCs (Chanda and Xu-Friedman 2010)."

Methods:

Line 656: Toff is determined as a monoexponential fit of the current, but there is no quantification of the goodness of the fit? Could a double exponential lead to better fit?

It is then mentioned line 796 that 10/15 cells have a biexponential fit, without much explanation.

We want to thank the reviewer and provide more explanations. The quality of the fits was ensured by $R^2 > 0.98$ and residual analysis. The photocurrents were fitted monoexponentially. The EPSCs were fitted mono- or bi-exponentially, whereas either the time constants derived from the monoexponential fits or the faster time constants derived from the biexponential fits were used to classify principal cells as bushy cells ($T_{off} \leq 0.5$ ms) or stellate cells ($T_{off} > 0.5$ ms) as shown in previous studies (Isaacson and Walmsley 1995; Gardner et al. 1999; Cao and Oertel 2010; Chanda and Xu-Friedman 2010; Butola et al. 2021).

Line 663: the authors mention peak recovery of ChRs, where is that data presented? page 28 -line 745:

"The temperature dependence of the off-kinetics and peak recovery of ChRs and their mutants was investigated at temperatures closer to physiological conditions (33 to 34 °C)."

We thank the reviewer for the comment. The data showing the peak recovery is presented in the new **Appendix Fig. S2**.

Line 746-749:

“The peak current recovery and the closing kinetics of selected ChRs were investigated at temperatures closer to physiological conditions (33 to 34 °C). The peak current amplitude was fully recovered after a period of 30 s in the dark (**Appendix Fig. S2**).

Line 858 : the sentence is uncomplete (no verb).

We would like to thank the reviewer and have corrected the sentence accordingly.

11th Oct 2025

Dear Dr. Mager,

Thank you for the submission of your revised manuscript to EMBO Molecular Medicine. We have now received the enclosed report from the referees who were asked to re-assess it. As you will see, the referees are now supportive, and I am pleased to inform you that we will be able to accept your manuscript pending the following amendments:

1. The remaining minor issues raised by Ref #2.

On a more editorial level, please do the following:

1. Remove all figures from the main manuscript file. Keep the figure legends for both main and Extended View (EV) figures at the end of the manuscript file.

2. Delete the "Authors' Contributions" section from the manuscript file.

3. References: Use "et al." after listing the first 10 authors in each reference.

4. Funding

- Ensure the funding information in the submission system matches the manuscript text. The following funding details are currently missing in the submission system: Grant 1690, HORIZON TMA MSCA Postdoctoral Fellowship (OPTOCODE, grant 101107675), Scholarship from the Göttingen Promotionskolleg für Medizinstudierende, funded by the Jacob-Henle-Programm/Else-Kröner-Fresenius-Stiftung (Promotionskolleg für Epigenomik und Genomdynamik 2021_EKPK.04)
- Remove all funders listed in the "Comments" box. Enter each funder individually using the "More Funders" option in the submission system.

5. EV tables:

- Remove all EV tables and their legends from the manuscript file.
- Upload each EV table as a separate file using the "Expanded View Content" file type.
- Each file must contain the table itself and a separate sheet labeled "Legend" containing the corresponding table legend.

6. Appendix:

- Remove all appendix files, figures, and tables from the manuscript.
- Also delete individual appendix figures and tables from the submission system.
- Compile all appendix material into a single PDF file named "Appendix". The PDF must begin with a title page that includes a Table of Contents listing all items with their page numbers. Place the legend below each figure and table within the appendix.

7. Figures 6H-J are cited, but panel J is missing or not labeled in the figure. Please resolve this discrepancy.

8. During our routine image check, we noted the following:

- There is image reuse between Figure 1A and Appendix Figure S1D. This reuse should be explicitly stated in the figure legends.

- In Appendix Figures S5A and S5D, a side section appears to have been removed, which is visible under image filters(see attached). Please provide the original, unprocessed source data for these figures.

9. Data availability:

- Remove the subheadings "Data Availability Statement" and "Code Availability Statement" from the manuscript. Merge the relevant content into a single "Data Availability" section. Place the Data Availability section before the Acknowledgments section in the manuscript.

- The following statements must be removed: "The code used for analysis is available from the corresponding authors upon reasonable request." "The data that support the findings of this study is available from the corresponding authors upon reasonable request." Since the study does not generate large-scale datasets, formal data deposition is not required.

- However, the computer code used for analysis must be deposited in an appropriate public repository (e.g., GitHub, Zenodo). A direct access link to the repository must be provided in the Data Availability section.

10. Please address the following comments related to figure legends:

- Please note that the exact p values are not provided in the legends of figures 1F, 2E, 4B, C; EV2 C, E, F; EV3 A-C

- Please note that the box plots need to be defined in terms of minima, maxima, centre, bounds of box and whiskers, and percentile in the legends of figures 1B, F; 2B, E; 6C

- Please note that information related to n is missing in the legends of figures 3H, 5G, H

- Please note that the error bars are not defined in the legends of figures 1C, D; 3H, 5H, EV1 B, C

11. Please correct the order and headings of the manuscript sections to: Abstract / Keywords / The Paper Explained /Introduction / Results / Discussion / Methods / Data Availability /Acknowledgements / Disclosure and Competing Interests Statement / References / Main Figure Legends / Tables /Expanded View Figure Legends

We look forward to reading a new revised version of your manuscript as soon as possible.

Kind regards,
Jingyi

Jingyi Hou
Senior Editor
EMBO Molecular Medicine

*** Instructions to submit your revised manuscript ***

***** Reviewer's comments *****

Referee #1 (Comments on Novelty/Model System for Author):

The provided in vitro and in vivo data are consistent with the conclusions.

Referee #1 (Remarks for Author):

The authors have thoroughly revised the manuscript. All of my questions and concerns have been adequately addressed. I thank the authors for their efforts and congratulate them on their work.

Referee #2 (Comments on Novelty/Model System for Author):

The paper technical mastery is of the highest quality, experiment are well designed and executed and interpretations are thorough. it linked proteic mutation to physiological properties of channels and effect on auditory restoration in whole organism. The best candidate variant that arise from the present study englobe together all the necessary properties making it the new landmark for auditory restoration. the potential for translation to clinical is straight forward. albeit the necessity for preclinical non human primate trials, which seem outside of the scope of this study. The study englobe various model to test the gene candidate, with the gerbil model (establish previously as a model for audition restoration) demonstrating consistent results.

Referee #2 (Remarks for Author):

The authors have addressed all my comments thoroughly and with well-reasoned arguments.

The revisions made in response to my feedback, as well as that of the other reviewer, have greatly improved the manuscript's readability, making it more coherent and narrative-driven, with a stronger emphasis on the potential clinical implications.

I only noted an error in legend of figure 5 (lines 498-498):

" Note the relationship between light-sensitivity and synaptic delay (e.g., synapses labeled in light blue horizontal hourglass icon () vs green cell vertical hourglass icon ())"

as far as i can tell, the icons mentioned are not on the figure.

The remaining minor issues raised by Ref #2.

I only noted an error in legend of figure 5(lines 498-498):

" Note the relationship between light-sensitivity and synaptic delay (e.g., synapses labeled in light blue horizontal hourglass icon () vs green cell vertical hourglass icon ())"

as far as i can tell, the icons mentioned are not on the figure.

We magnified the icons in order to improve the visibility and thank the reviewer for the comment, which helped to enhance clarity.

17th Nov 2025

Dear Dr. Mager,

Thank you for sending us your revised manuscript. We are pleased to inform you that your manuscript is accepted for publication and is now being sent to our publisher to be included in the next available issue of EMBO Molecular Medicine.

Sincerely,
Jingyi

Jingyi Hou
Senior Editor
EMBO Molecular Medicine
